# Alternating Gradient Flows: A Theory of Feature Learning in Two-layer Neural Networks

**Daniel Kunin**[†]
Stanford University

**Giovanni Luca Marchetti**[†]
KTH

**Feng Chen**
Stanford University

**Dhruva Karkada**
UC Berkeley

**James B. Simon**
UC Berkeley and Imbue

**Michael R. DeWeese**
UC Berkeley

**Surya Ganguli**
Stanford University

**Nina Miolane**
UC Santa Barbara

## Abstract

What features neural networks learn, and how, remains an open question. In this paper, we introduce *Alternating Gradient Flows (AGF)*, an algorithmic framework that describes the dynamics of feature learning in two-layer networks trained from small initialization. Prior works have shown that gradient flow in this regime exhibits a staircase-like loss curve, alternating between plateaus where neurons slowly align to useful directions and sharp drops where neurons rapidly grow in norm. AGF approximates this behavior as an alternating two-step process: maximizing a utility function over dormant neurons and minimizing a cost function over active ones. AGF begins with all neurons dormant. At each iteration, a dormant neuron activates, triggering the acquisition of a feature and a drop in the loss. AGF quantifies the order, timing, and magnitude of these drops, matching experiments across several commonly studied architectures. We show that AGF unifies and extends existing saddle-to-saddle analyses in fully connected linear networks and attention-only linear transformers, where the learned features are singular modes and principal components, respectively. In diagonal linear networks, we prove AGF converges to gradient flow in the limit of vanishing initialization. Applying AGF to quadratic networks trained to perform modular addition, we give the first complete characterization of the training dynamics, revealing that networks learn Fourier features in decreasing order of coefficient magnitude. Altogether, AGF offers a promising step towards understanding feature learning in neural networks.

## 1 Introduction

The impressive performance of artificial neural networks is often attributed to their capacity to learn *features* from data. Yet, a precise understanding of *what* features they learn and *how* remains unclear.

A large body of recent work has sought to understand what features neural networks learn by reverse engineering the computational mechanisms implemented by trained neural networks [1–6]. Known as *mechanistic interpretability (MI)*, this approach involves decomposing trained networks into interpretable components to uncover their internal representations and algorithmic strategies [7]. MI has achieved notable successes in understanding the *emergent abilities* of large language models [8], including identifying induction heads that enable in-context learning [3] and revealing that small transformers trained on algebraic tasks use Fourier features [6, 9]. Despite these discoveries, mechanistic interpretability remains limited by its empirical nature, lacking a theoretical framework to formally define features and predict when and how they emerge [10].

---

[†] Correspondence to: `kunin@berkeley.edu` and `glma@kth.se`.

39th Conference on Neural Information Processing Systems (NeurIPS 2025).

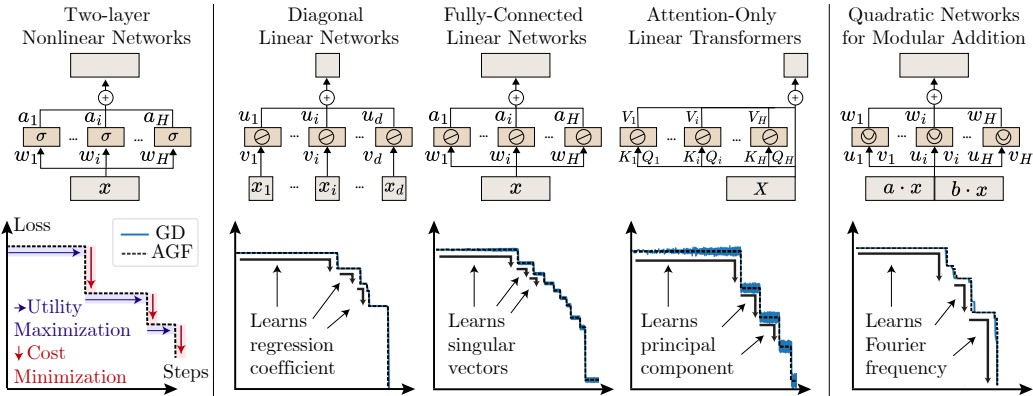

Figure 1: **A unified theory of feature learning in two-layer networks.** Left: Alternating Gradient Flows (AGF) models feature learning as a two-step process alternating between utility maximization (blue plateaus) and cost minimization (red drops), where each drop reflects learning a new feature (see Section 2). Middle: AGF unifies prior analyses of saddle-to-saddle dynamics (see Sections 3 and 4). Right: AGF enables new analysis of empirical phenomena (see Section 5).

A different line of work, rooted in *deep learning theory*, has sought to understand how features are learned by directly studying the dynamics of the network during training. Empirical studies suggest that neural networks learn simple functions first, progressively capturing more complex features during training [11–14]. Termed *incremental*, *stepwise*, or *saddle-to-saddle*, this learning process is marked by long plateaus of minimal loss change followed by rapid drops. It is conjectured to arise from networks, initialized at small-scale, jumping between saddle points of the loss landscape [15], with each drop corresponding to the acquisition of a new feature [16]. This saddle-to-saddle process has been explored across a range of simple settings, including diagonal linear networks [17, 18], fully connected linear networks [19, 15], tensor decomposition [20, 21], self-supervised learning [22, 23], shallow ReLU networks [24–30], attention-only transformers [31, 32], and multi-index models [33–39]. However, these analyses often rely on different simplifying assumptions about the architecture (e.g., linear activations), data (e.g., orthogonal inputs), or optimizer (e.g., layer-wise learning), making it difficult to establish a unified, scalable understanding of feature learning.

In this work, we introduce a theoretical framework that unifies existing analyses of saddle-to-saddle dynamics in two-layer networks under the vanishing initialization limit, precisely predicting the order, timing, and magnitude of loss drops, while extending beyond classical settings to explain empirically observed patterns of feature emergence. See Figure 1 for a visual overview of the paper. Our main contributions are:

1. We introduce *Alternating Gradient Flows (AGF)*, a two-step algorithm that approximates gradient flow in two-layer networks with small initialization by alternating between maximizing a utility over dormant neurons and minimizing a cost over active ones (Section 2).
2. We prove that AGF converges to the dynamics of gradient flow for diagonal linear networks in the vanishing initialization limit (Section 3).
3. We show how AGF unifies and generalizes existing theories of saddle-to-saddle dynamics across fully connected linear networks and attention-only linear transformers (Section 4).
4. We use AGF to predict the emergence of Fourier features in modular arithmetic tasks, providing the first theoretical account of both the order in which frequencies appear and the dynamics that drive it (Section 5).

Our work takes a step toward unifying deep learning theory and mechanistic interpretability, suggesting that what features networks learn, and how, can be understood through optimization dynamics. See Appendix A for further discussion of related theoretical approaches to studying feature learning and saddle-to-saddle dynamics in two-layer networks, including mean-field analysis [40–43] and teacher–student frameworks [44–51].

## 2 Deriving a Two-step Algorithm that Approximates Gradient Flow

**Setup and notations.** We consider a two-layer neural network with $H$ hidden *neurons* of the form $f(x; \Theta) = \sum_{i=1}^{H} a_i \sigma(w_i^\mathsf{T} g_i(x))$, where each hidden neuron $i$ has learnable input weights $w_i \in \mathbb{R}^d$ and output weights $a_i \in \mathbb{R}^c$, and processes a potentially neuron-specific input representation $g_i(x)$.

The activation $\sigma\colon \mathbb{R} \to \mathbb{R}$ is *origin-passing*, i.e., $\sigma(0) = 0$, as satisfied by common functions (e.g., linear, ReLU, square, tanh). Let $f_i(x;\theta_i) = a_i\sigma(w_i^{\mathsf{T}}g_i(x))$ denote the output of the $i^{\text{th}}$ neuron, where $\theta_i = (w_i, a_i)$. The parameters $\Theta = (\theta_1, \ldots, \theta_H) \in \mathbb{R}^{H \times (d+c)}$ for the entire network evolve under gradient flow (GF), $\dot{\Theta} = -\eta\nabla_\Theta\mathcal{L}(\Theta)$, to minimize the MSE loss $\mathcal{L}(\Theta) = \mathbb{E}_x\left[\frac{1}{2}\|y(x) - f(x;\Theta)\|^2\right]$, where $y\colon \mathbb{R}^d \to \mathbb{R}^c$ is the ground truth function and the expectation is taken over the input distribution. The learning rate $\eta > 0$ rescales time without affecting the trajectory. The parameter vectors for each neuron are initialized $a_i \sim \mathcal{N}(0, \frac{\alpha^2}{2c}\mathbf{I}_c)$, $w_i \sim \mathcal{N}(0, \frac{\alpha^2}{2d}\mathbf{I}_d)$. In the limit $\alpha \to \infty$, the gradient flow dynamics enter the *kernel regime* [52, 53]. When $\alpha = 1$, the dynamics correspond to a *mean-field* or *maximal update* parameterization [40, 54]. We study the *small-scale* initialization regime $\alpha \ll 1$, where the dynamics are conjectured to follow a saddle-to-saddle trajectory as $\alpha \to 0$ [15, 14].

## 2.1 The behavior of gradient flow at small-scale initialization

Bakhtin [55] showed that, under a suitable time rescaling, the trajectory of a smooth dynamical system initialized at a critical point and perturbed by vanishing noise converges to a piecewise constant process that jumps instantaneously between critical points. These results suggest that, in the vanishing initialization limit $\alpha \to 0$, gradient flow converges to a similar piecewise constant process, jumping between saddle points of the loss before reaching a minimum. To approximate this behavior, we first examine the structure of critical points in the loss landscape of the two-layer networks we study, and then analyze the dynamics of gradient flow near them. These dynamics reveals two distinct phases, separated by timescales, which in turn motivates the construction of AGF.

**Critical points are structured by partitions of neurons into dormant and active sets.** As $\alpha \to 0$, the initialization converges to the origin $\Theta = \mathbf{0}$, which is a critical point of the loss. The origin is one of potentially an exponential number of distinct (up to symmetry) critical points, that can be structured according to partitions of the $H$ hidden neurons. Formally, split neurons into two disjoint sets: *dormant neurons* $\mathcal{D} \subseteq [H]$ and *active neurons* $\mathcal{A} = [H]\backslash\mathcal{D}$, with parameters $\Theta_\mathcal{D} = (\theta_i)_{i\in\mathcal{D}}$ and $\Theta_\mathcal{A} = (\theta_i)_{i\in\mathcal{A}}$, respectively. Due to the two-layer structure of our model and the origin-passing activation function, setting the parameters of the dormant neurons to zero $\Theta_\mathcal{D} = \mathbf{0}$ restricts the loss to a function $\mathcal{L}(\Theta_\mathcal{A})$ of only the active neurons. Now any critical point $\Theta_\mathcal{A}^*$ of the restricted loss yields a critical point $\Theta = (\mathbf{0}, \Theta_\mathcal{A}^*)$ of the original loss. Thus the $2^H$ partitions of dormant and active neurons structure critical ponts of the loss landscape, with the origin corresponding to the special critical point with all neurons dormant.

**Dynamics near a saddle point.** Consider the dynamics near a saddle point defined by a dormant and active set, where we assume the active neurons are at a local minimum of their restricted loss $\mathcal{L}(\Theta_\mathcal{A})$. By construction, neurons in our two-layer network only interact during training through the *residual* $r(x;\Theta) = y(x) - f(x;\Theta) \in \mathbb{R}^c$, which quantifies the difference between the predicted and target values at each data point $x$. Near a critical point, the residual primarily depends on the active neurons, as the contribution from the dormant neurons is effectively zero. Consequently, dormant neurons evolve independently of each other, while active neurons remain collectively equilibrated. To characterize the dynamics of the dormant neurons, which will determine how the dynamics escape the saddle point, we define the *utility function*, $\mathcal{U}_i\colon \mathbb{R}^{d+c} \to \mathbb{R}$ for each $i \in \mathcal{D}$ as:

$$\mathcal{U}_i(\theta; r) = \mathbb{E}_x\left[\langle f_i(x;\theta), r(x)\rangle\right], \quad \text{where } r(x) = y(x) - f(x;\Theta_\mathcal{A}). \tag{1}$$

Intuitively, the utility $\mathcal{U}_i$ quantifies how correlated the $i^{\text{th}}$ dormant neuron is with the residual $r$, which itself is solely determined by the active neurons. Using this function we can approximate the directional and radial dynamics for the parameters of each dormant neuron as

$$\frac{d}{dt}\bar{\theta}_i = \eta\|\theta_i\|^{\kappa-2}\mathbf{P}_{\bar{\theta}_i}^\perp\nabla_\theta\mathcal{U}_i(\bar{\theta}_i; r), \qquad \frac{d}{dt}\|\theta_i\| = \eta\kappa\|\theta_i\|^{\kappa-1}\mathcal{U}_i(\bar{\theta}_i; r), \tag{2}$$

where $\bar{\theta}_i = \theta_i/\|\theta_i\|$ are the normalized parameters, $\mathbf{P}_{\bar{\theta}_i}^\perp = \left(\mathbf{I}_d - \bar{\theta}_i\bar{\theta}_i^{\mathsf{T}}\right)$ is an orthogonal projection matrix, and $\kappa \geq 2$ is the leading order of the Taylor expansion of the utility from the origin. Note that the directional dynamics scale as $\|\theta_i\|^{\kappa-2}$, while the radial dynamics scale as $\|\theta_i\|^{\kappa-1}$. Because $\|\theta_i\| \ll 1$ for dormant neurons, the directional dynamics evolves significantly faster than the radial dynamics. Consequently, the contribution of a dormant neuron to the residual is suppressed, keeping the residual effectively constant, and this approximation to the dynamics stays valid. This separation of timescales between directional and radial components has been utilized in several previous studies of gradient descent dynamics near the origin [56–61].

**Algorithm 1: AGF**

**Initialize:** $\mathcal{D} \leftarrow [H], \mathcal{A} \leftarrow \emptyset, \mathcal{S} \leftarrow 0 \in \mathbb{R}^H$
**while** $\nabla\mathcal{L}(\Theta_{\mathcal{D}}) \neq 0$ **do**

> **while** $\forall i \in \mathcal{D} \quad \mathcal{S}_i \leq c_i$ **do**
>> **for** $i \in \mathcal{D}$ **do**
>>> $\frac{d\bar{\theta}_i}{dt} = \eta_\alpha \|\theta_i\|^{\kappa-2} \mathbf{P}^\perp_{\bar{\theta}_i} \nabla\mathcal{U}(\bar{\theta}_i, r)$
>>> $\frac{d\mathcal{S}_i}{dt} = \eta_\alpha \kappa \mathcal{U}(\bar{\theta}_i, r)$
>
> Activate neuron: $\mathcal{D}, \mathcal{A} \leftarrow \mathcal{D} \setminus \{i_*\}, \mathcal{A} \cup \{i_*\}$
>
> **while** $\nabla\mathcal{L}(\Theta_{\mathcal{A}}) \neq 0$ **do**
>> **for** $j \in \mathcal{A}$ **do**
>>> $\frac{d\theta_j}{dt} = -\nabla_{\theta_j}\mathcal{L}(\Theta_{\mathcal{A}})$
>
> Remove collapsed neurons: $\mathcal{D}, \mathcal{A} \leftarrow \mathcal{D} \cup \mathcal{C}, \mathcal{A} \setminus \mathcal{C}$

**Conceptual Illustration: AGF**

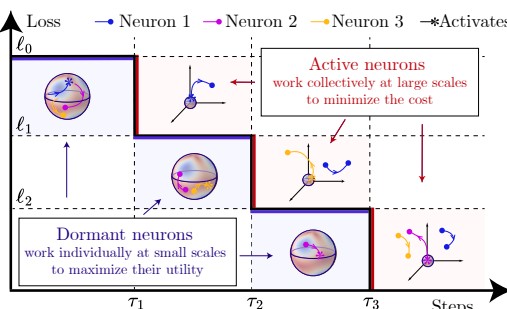

Figure 2: **Alternating Gradient Flows (AGF).** AGF alternates between utility maximization (blue) over dormant neurons and cost minimization (red) over active neurons, predicting saddle-to-saddle dynamics in training. Utility maximization: dormant neurons evolve independently driven by projected gradient flow to maximize their utility. We keep track of the accumulated utility for each dormant neuron to determine the first neuron to transition to active. Cost minimization: Active neurons work collectively driven by gradient flow to minimize the loss until convergence. After convergence, active neurons compute a new residual that determines the utility at the next iteration. It is possible in this step for active neurons to become dormant again (see Appendix B).

**Estimating the jump time from dormant to active.** The dynamical approximation in Equation (2) breaks down if a dormant neuron crosses the threshold $\|\theta_i\| = \mathcal{O}(1)$, becoming active by influencing the residual. We denote by $i_* \in \mathcal{D}$ the first dormant neuron to reach this threshold and by $\tau_{i_*}$ the *jump time* at which this dormant to active transition occurs. To determine $\tau_{i_*}$, we compute for each dormant neuron the earliest time $\tau_i$ when $\|\theta_i(\tau_i)\| > 1$, such that $i_* = \arg\min_{i \in \mathcal{D}} \tau_i$. This time $\tau_i$ is defined implicitly in terms of the *accumulated utility*, the path integral $\mathcal{S}_i(t) = \int_0^t \kappa \mathcal{U}_i(\bar{\theta}_i(s); r)\, ds$:

$$\tau_i = \inf\left\{ t > 0 \;\middle|\; \mathcal{S}_i(t) > \frac{c_i}{\eta} \right\}, \quad \text{where } c_i = \begin{cases} -\log\left(\|\theta_i(0)\|\right) & \text{if } \kappa = 2, \\ -\frac{1}{2-\kappa}\left(\|\theta_i(0)\|^{2-\kappa} - 1\right) & \text{if } \kappa > 2. \end{cases} \quad (3)$$

In the vanishing initialization limit as $\alpha \to 0$, the jump time defined in Equation (3) diverges as gradient flow exhibits extreme slowing near the critical point at the origin. Thus, to capture meaningful dynamics in the limit, we *accelerate* time by setting the learning rate $\eta = \mathbb{E}_{\Theta_0}[c_i]$ where $c_i$ is the threshold defined in Equation (3) and the expectation is taken over the initialization.

## 2.2 Alternating Gradient Flow (AGF) approximates GF at small-scale initialization

Based on the behaviors of gradient flow (GF) at small-scale initialization, we introduce *Alternating Gradient Flows (AGF)* (Algorithm 1) as an approximation of the dynamics. AGF characterizes phases of constant loss as periods of utility maximization and sudden loss drops as cost minimization steps, accurately predicting both the loss levels and the timing of loss drops during training. At initialization, all neurons are dormant. At each iteration, a neuron leaves the dormant set and enters the active set. The dormant neurons work individually to maximize their utility, and the active neurons collectively work to minimize a cost (Figure 2). Specifically, each iteration in AGF is divided into two steps.

**Step 1: Utility maximization over dormant neurons.** In this step, only dormant neurons update their parameters. They independently maximize (locally) their utility function over the unit sphere by following the projected gradient flow (left of Equation (2)). We keep track of the accumulated utility for each neuron to determine the first neuron to transition and the jump time using Equation (3). At the jump time, we record the norms and directions of the dormant neurons that have not activated, as they will serve as the initialization for the utility maximization phase of the next iteration.

**Step 2: Cost minimization over active neurons.** In this step, all active neurons interact to minimize (locally) the loss, by following the negative gradient flow of $\mathcal{L}(\Theta_{\mathcal{A}})$. For previously active neurons, the initialization is determined by the previous cost minimization step. For the newly activated neuron, the initialization comes from the utility maximization step. It is possible in this step for active neurons to become dormant again if the optimization trajectory brings an active neuron near the origin (see Appendix B). After convergence, we compute the new residual $r(x) = y(x) - f(x; \Theta_{\mathcal{A}})$, which defines a new utility for the remaining dormant neurons at the next iteration.

**Termination.** We repeat both steps, recording the corresponding sequence of jump times and loss levels, until there are either no remaining dormant neurons or we are at a local minimum of the loss. Through this process, we have generated a precise sequence of saddle points and jump times to describe how gradient flow leaves the origin through a saddle-to-saddle process.

While AGF and GF exhibit similar behaviors at small-scale initialization ($\alpha \ll 1$), a natural question is whether their trajectories converge to each other as $\alpha \to 0$. While a general proof remains open (see Section 6), in the next section we present an illustrative setting where convergence can be established.

## 3 A Setting Where AGF and GF Provably Converge to Each Other

Diagonal linear networks are simple yet insightful models for analyzing learning [62–65, 17, 66]. Central to their analysis is an interpretation of gradient flow in function space as mirror descent with an initialization-dependent potential that promotes sparsity when $\alpha$ is small. Using this perspective, Pesme and Flammarion [18] characterized the sequence of saddles and jump times for gradient flow in the limit $\alpha \to 0$. We show that, in this limit, AGF converges to the exact same sequence, establishing a setting where AGF provably converges to gradient flow (see Figure 3).

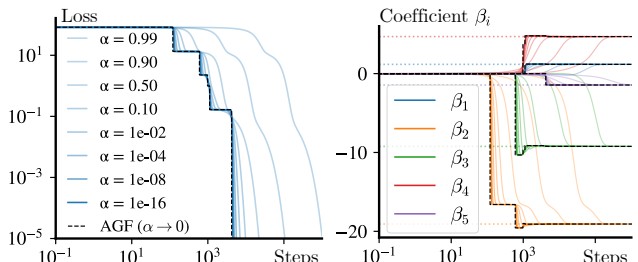

Figure 3: **AGF = GF as $\alpha \to 0$ in diagonal linear networks.** Training loss curves for a diagonal linear network under the setup described in Section 3 for various initialization values $\alpha$. As $\alpha \to 0$, the trajectory predicted by AGF and the empirics of gradient flow converge. To ensure a meaningful comparison between experiments we set $\eta = -\log(\alpha)$.

We consider a two-layer diagonal linear network trained via gradient flow to minimize the MSE loss $\mathcal{L}(\beta) = \frac{1}{2n}\|y - \beta^\mathsf{T} X\|^2$, where the regression coefficients $\beta \in \mathbb{R}^d$ are parameterized as the element-wise product $\beta = u \odot v$ with $u, v \in \mathbb{R}^d$, and $(X, y) \in \mathbb{R}^{d \times N} \times \mathbb{R}^N$ are the input, output data. This setup fits naturally within AGF, where the $i^{\text{th}}$ neuron corresponds to $\beta_i$ with parameters $\theta_i = (u_i, v_i)$, data representation $g_i(x) = x_i$, and $\kappa = 2$. Following Pesme and Flammarion [18], we initialize $\theta_i(0) = (\sqrt{2}\alpha, 0)$ such that $\beta_i(0) = 0$, and assume the inputs are in *general position*, a standard technical condition in Lasso literature [67, 18] to rule out unexpected linear dependencies.

**Utility maximization.** The utility function for the $i^{\text{th}}$ neuron is $\mathcal{U}_i = -u_i v_i \nabla_{\beta_i}\mathcal{L}(\beta_{\mathcal{A}})$, which is maximized on the unit sphere $u_i^2 + v_i^2 = 1$ when $\text{sgn}(u_i v_i) = -\text{sgn}(\nabla_{\beta_i}\mathcal{L}(\beta_{\mathcal{A}}))$ and $|u_i| = |v_i|$, yielding a maximal utility of $\bar{\mathcal{U}}_i^* = \frac{1}{2}|\nabla_{\beta_i}\mathcal{L}(\beta_{\mathcal{A}})|$. What makes this setting special is that not only can the maximal utility be computed in closed form, but every quantity involved in utility maximization—namely the accumulated utility, jump times, and directional dynamics—admits an exact analytical expression. The key insight is that the normalized utility $\bar{\mathcal{U}}_i(t)$ for each dormant neuron $i$ evolves according to a separable Riccati ODE, interpolating from its initial value $\bar{\mathcal{U}}_i(0)$ to its maximum $\bar{\mathcal{U}}_i^*$. As a result, we can derive an explicit formula for the normalized utility $\bar{\mathcal{U}}_i(t)$, whose integral yields the accumulated utility $\mathcal{S}_i(t)$, evolving as:

$$\mathcal{S}_i(\tau^{(k)} + t) = \frac{1}{2\eta_\alpha} \log \cosh\left(2\eta_\alpha\left(2\mathcal{U}_i^* t \pm \frac{1}{2\eta_\alpha}\cosh^{-1}\exp\left(2\eta_\alpha\mathcal{S}_i\left(\tau^{(k)}\right)\right)\right)\right), \quad (4)$$

where $\eta_\alpha = -\log(\sqrt{2}\alpha)$ is the learning rate and the unspecified sign is chosen based on $\text{sgn}(u_i v_i)$ and $\text{sgn}(\nabla_{\beta_i}\mathcal{L}(\beta_{\mathcal{A}}))$, as explained in Appendix C. This expression allows us to determine the next neuron to activate, as the first $i \in \mathcal{D}$ for which $\mathcal{S}_i = 1$, from which the jump time can be computed.

**Cost minimization.** Active neurons represent non-zero regression coefficients of $\beta$ (see Figure 3 right). During the cost minimization step, the active neurons work to collectively minimize the loss. When a neuron activates, it does so with a certain sign $\text{sgn}(u_i v_i) = -\text{sgn}(\nabla_{\beta_i}\mathcal{L}(\beta_{\mathcal{A}}))$. If, during the cost minimization phase, a neuron changes sign, then it can do so only by returning to dormancy first. This is due to the fact that throughout the gradient flow dynamics, the quantity $u_i^2 - v_i^2 = 2\alpha^2$ is conserved and thus, in order to flip the sign of the product $u_i v_i$, the parameters must pass through their initialization. As a result, the critical point reached during the cost minimization step is the

unique solution to the constrained optimization problem,

$$\beta_{\mathcal{A}} = \underset{\beta \in \mathbb{R}^d}{\arg\min} \mathcal{L}(\beta) \quad \text{subject to} \quad \begin{cases} \beta_i = 0 & \text{if } i \notin \mathcal{A}, \\ \beta_i \cdot \text{sgn}(u_i v_i) \geq 0 & \text{if } i \in \mathcal{A}. \end{cases} \tag{5}$$

All coordinates where $(\beta_{\mathcal{A}})_i = 0$ are dormant at the next step of AGF.

**AGF in action: sparse regression.** We now connect AGF to the algorithm proposed by Pesme and Flammarion [18], which captures the limiting behavior of gradient flow under vanishing initialization. Their algorithm tracks, for each coefficient of $\beta$, an integral of the gradient, $S_i(t) = -\int_0^t \nabla_{\beta_i} \mathcal{L}(\beta(\tilde{t}_\alpha(s))) \, ds$, with a time rescaling $\tilde{t}_\alpha(s) = -\log(\alpha)s$. They show that in the limit $\alpha \to 0$, this quantity is piecewise linear and remains bounded in $[-1, 1]$. Using these properties, each step of their algorithm determines a new coordinate to activate (the first for which $S_i(t) = \pm 1$), adds it to an active set, then solves a constrained optimization over this set, iterating until convergence. Despite differing formulations, this process is identical to AGF in the limit $\alpha \to 0$:

**Theorem 3.1.** *Let $(\beta_{AGF}, t_{AGF})$ and $(\beta_{PF}, t_{PF})$ be the sequences produced by AGF and Algorithm 1 of Pesme and Flammarion [18], respectively. Then, $(\beta_{AGF}, t_{AGF}) \to (\beta_{PF}, t_{PF})$ pointwise as $\alpha \to 0$.*

The key connection lies in the asymptotic identity $\log\cosh(x) \to |x|$ as $|x| \to \infty$, which implies that the accumulated utility $\mathcal{S}_i(t)$ in AGF converges to the absolute value of the integral $S_i(t)$. The unspecified sign in AGF corresponds to the sign of the boundary conditions in their algorithm. Thus, AGF converges to the same saddle-to-saddle trajectory as the algorithm of Pesme and Flammarion [18], and therefore to gradient flow. See Appendix C for a full derivation.

## 4 AGF Unifies Existing Analysis of Saddle-to-Saddle Dynamics

**Fully connected linear network.** Linear networks have long served as an analytically tractable setting for studying neural network learning dynamics [68–73]. Such linear networks exhibit highly nonlinear learning dynamics. Saxe et al. [70] demonstrated that gradient flow from a task-aligned initialization learns a sequential singular value decomposition of the input-output cross-covariance. This behavior persists in the vanishing initialization limit without task alignment [57, 74, 19, 20, 15], and is amplified by depth [75, 62]. Here, we show how AGF naturally recovers such greedy low-rank learning.

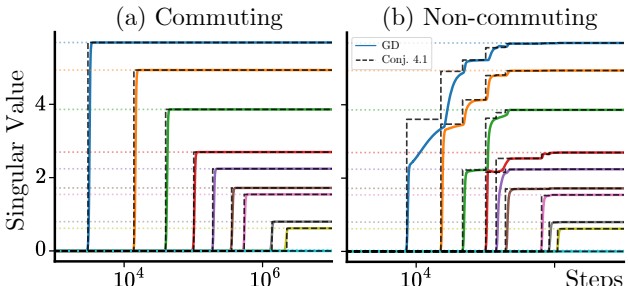

Figure 4: **Stepwise singular value decomposition.** Training a two-layer fully connected linear network on Gaussian inputs with a power-law covariance $\Sigma_{xx}$ and labels $y(x) = Bx$ generated from a random $B$. We show the dynamics of the singular values of the network's map $AW$ when $\Sigma_{xx}$ commutes with $\Sigma_{yx}^\mathsf{T}\Sigma_{yx}$ (a) and when it does not (b). Conjecture 4.1 (black dashed lines) predicts the dynamics well.

We consider a fully connected two-layer linear network, $f(x; \theta) = AWx$, with parameters $W \in \mathbb{R}^{H \times d}$ and $A \in \mathbb{R}^{c \times H}$. The network is trained to minimize the MSE loss with data generated from the linear map $y(x) = Bx$, where $B \in \mathbb{R}^{c \times d}$ is an unknown matrix. The inputs are drawn independently from a Gaussian $x \sim \mathcal{N}(0, \Sigma_{xx})$, where $\Sigma_{xx} \in \mathbb{R}^{d \times d}$ is the input covariance, and $\Sigma_{yx} = B\Sigma_{xx} \in \mathbb{R}^{c \times d}$ is the input-output cross-covariance, which we assume are full rank with distinct singular values. A subtlety in applying AGF to this setting is that the standard notion of a neuron used in Section 2 is misaligned with the geometry of the loss landscape. Due to the network's invariance under $(W, A) \mapsto (GW, AG^{-1})$ for any $G \in \mathrm{GL}_H(\mathbb{R})$, critical points form manifolds entangling hidden neurons, and conserved quantities under gradient flow couple their dynamics [76, 77]. To resolve this, we can reinterpret AGF in terms of evolving dormant and active *orthogonal basis vectors* instead of neurons. Each basis vector $(\tilde{a}_i, \tilde{w}_i) \in \mathbb{R}^{c+d}$ forms a rank-1 map $\tilde{a}_i\tilde{w}_i^\mathsf{T} \in \mathbb{R}^{c \times d}$ such that the function computed by the network is the sum over these rank-1 maps, $f(x; \Theta) = \sum_{i \in [\tilde{H}]} \tilde{a}_i\tilde{w}_i^\mathsf{T}x$, where $\tilde{H} = \min(c, H, d)$. From this perspective, at each iteration of AGF the dormant set loses one basis vector, while the active set gains one. See Appendix D for details.

**Utility maximization.** Let $\beta_{\mathcal{A}} = \sum_{i \in \mathcal{A}} \tilde{a}_i\tilde{w}_i^\mathsf{T}$ be the function computed by the active basis vectors and $m = |\mathcal{D}|$. The total utility over the dormant basis vectors is $\mathcal{U} = \sum_{i=1}^m \tilde{a}_i^\mathsf{T}\nabla_\beta\mathcal{L}(\beta_{\mathcal{A}})\tilde{w}_i$.

Maximizing this sum while maintaining orthonormality between the basis vectors yields a Rayleigh quotient problem, whose solution aligns the dormant basis $(\tilde{a}_i, \tilde{w}_i)_{i=1}^m$ with the top $m$ singular modes $(u_i, v_i)_{i=1}^m$ of $\nabla_\beta \mathcal{L}(\beta_\mathcal{A})$. Each aligned pair attains a maximum utility of $\mathcal{U}_i^* = \sigma_i/2$, where $\sigma_i$ is the corresponding singular value. The basis vector aligned with the top singular mode activates first, exiting the dormant set and joining the active one.

**Cost minimization.** The cost minimization step of AGF can be recast as a reduced rank regression problem, minimizing $\mathcal{L}(\beta)$ over $\beta \in \mathbb{R}^{c \times d}$ subject to $\text{rank}(\beta) = k$, where $k = |\mathcal{A}|$. As first shown in Izenman [78], the global minimum for this problem is an orthogonal projection of the OLS solution $\beta_\mathcal{A}^* = \mathbf{P}_{U_k} \Sigma_{yx} \Sigma_{xx}^{-1}$, where $\mathbf{P}_{U_k} = U_k U_k^\intercal$ is the projection onto $U_k \in \mathbb{R}^{c \times k}$, the top $k$ eigenvectors of $\Sigma_{yx} \Sigma_{xx}^{-1} \Sigma_{yx}^\intercal$. Using this solution, we can show that the next step of utility maximization will be computed with the matrix $\nabla_\beta \mathcal{L}(\beta_\mathcal{A}^*) = \mathbf{P}_{U_k}^\perp \Sigma_{yx}$ where $\mathbf{P}_{U_k}^\perp = \mathbf{I}_c - \mathbf{P}_{U_k}$.

**AGF in action: greedy low-rank learning.** In the vanishing initialization limit, AGF reduces to an iterative procedure that selects the top singular mode of $\nabla_\beta \mathcal{L}(\beta_\mathcal{A}^*)$, transfers this vector from the dormant to the active basis, then minimizes the loss with the active basis to update $\beta_\mathcal{A}^*$. This procedure is identical to the Greedy Low-Rank Learning (GLRL) algorithm by Li et al. [20] that characterizes the gradient flow dynamics of two-layer matrix factorization problems with infinitesimal initialization. Encouraged by this connection, we make the following conjecture:

**Conjecture 4.1.** *In the initialization limit $\alpha \to 0$, a two-layer fully connected linear network trained by gradient flow, with $\eta = -\log(\alpha)$, learns one rank at a time leading to the sequence*

$$f^{(k)}(x) = \sum_{i \leq k} \mathbf{P}_{u_i} \Sigma_{yx} \Sigma_{xx}^{-1} x, \qquad \ell^{(k)} = \tfrac{1}{2} \sum_{i > k} \mu_i, \qquad \tau^{(k)} = \sum_{i \leq k} \Delta\tau^{(i)}, \qquad (6)$$

*where $0 \leq k \leq \min(d, H, c)$, $(u_i, \mu_i)$ are the eigenvectors and eigenvalues of $\Sigma_{yx} \Sigma_{xx}^{-1} \Sigma_{yx}^\intercal$, $\sigma_i^{(k)}$ is the $i^\text{th}$ singular value of $\mathbf{P}_{U_k}^\perp \Sigma_{yx}$, and $\Delta\tau^{(i)} = \left(1 - \sum_{j=0}^{i-2} \sigma_{i-j}^{(j)} \Delta\tau^{(j+1)}\right)/\sigma_1^{(i-1)}$ with $\Delta\tau^{(0)} = 0$.*

When $\Sigma_{xx}$ commutes with $\Sigma_{yx}^\intercal \Sigma_{yx}$, Conjecture 4.1 recovers the sequence originally proposed by Gidel et al. [19]. Figure 4 empirically supports this conjecture. See Appendix D for details.

**Attention-only linear transformer.** Pretrained large language models can learn new tasks from only a few examples in context, without explicit fine-tuning [79]. To understand the emergence of this *in-context learning* ability, previous empirical [80–82] and theoretical works [83–88] have examined how transformers learn to perform linear regression in context. Notably, Zhang et al. [32] showed that an attention-only linear transformer learns to implement principal component regression sequentially, with each learned component corresponding to a drop in the loss. We show that AGF recovers their analysis (see Figure 5).

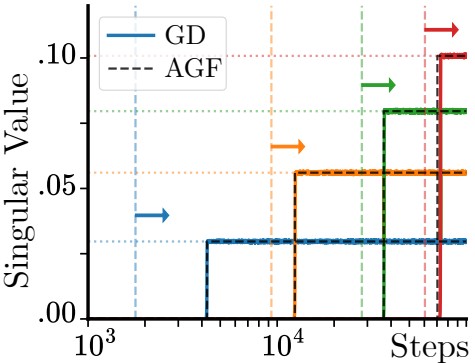

Figure 5: **Stepwise principal component regression.** Training a linear transformer to learn linear regression in context. We show the evolution of singular values of $\sum_{i=1}^H V_i K_i Q_i^\intercal$. Horizontal lines show theoretical $A_k$ and vertical dashed lines show lower bounds for the jump time from Equation (7) with $l = k - 1$. Dashed black lines are numerical AGF predictions.

We consider a attention-only linear transformer $f(X; \Theta) = X + \sum_{i=1}^H W_{V,i} X X^\intercal W_{K,i} W_{Q,i}^\intercal X$, where $X$ is the input sequence, $W_V \in \mathbb{R}^{H \times (d+1) \times (d+1)}$, $W_K, W_Q \in \mathbb{R}^{H \times (d+1) \times R}$ represent the value, key, and query matrices respectively, and $H$ denotes the number of attention heads. While this network uses linear activations, it is cubic in its input and parameters. Also, when the rank $R = 1$, each attention head behaves like a homogeneous neuron with $\kappa = 3$, making it compatible with the AGF framework.

We consider the linear regression task with input $X_i = \binom{x_i}{y_i}$ for $i \leq N$, where $x_i \sim \mathcal{N}(0, \Sigma_{xx})$, and $y_i = \beta^\intercal x_i$ with $\beta \sim \mathcal{N}(0, \mathbf{I}_d)$. The input covariance $\Sigma_{xx} \in \mathbb{R}^{d \times d}$ has eigenvalues $\lambda_1 \geq \cdots \geq \lambda_d > 0$ with corresponding eigenvectors $v_1, \ldots, v_d$. The final input token is $X_{N+1} = \binom{x_{N+1}}{0}$. The network is trained by MSE loss to predict $y_{N+1}$ given the entire sequence $X$, where the model prediction is taken to be $\hat{y}_{N+1} = f(X; \Theta)_{d+1, N+1}$. Following Zhang et al. [32], we initialize all

parameters to zero except for some slices, which we denote by $V \in \mathbb{R}^H, Q, K \in \mathbb{R}^{H \times d}$, which are initialized from $\mathcal{N}(0, \alpha)$. The parameters initialized at zero will remain at zero throughout training [32], and the model prediction reduces to $\hat{y}_{N+1} = \sum_{h=1}^{H} V_h \sum_{n=1}^{N} y_n x_n^\mathsf{T} K_h Q_h^\mathsf{T} x_{N+1}$. We defer derivations to Appendix E, and briefly outline how AGF predicts the saddle-to-saddle dynamics.

**Utility maximization.** At the $k^{\text{th}}$ iteration of AGF, the utility of a dormant attention head is $\mathcal{U}_i = N V_i Q_i^\mathsf{T} (\Sigma_{xx}^2 - \sum_{h=1}^{k-1} \lambda_h^2 v_h v_h^\mathsf{T}) K_i$ which is maximized over normalized parameters when $\bar{V}_i = \pm 1/\sqrt{3}$ and $\bar{Q}_i, \bar{K}_i = \pm v_k/\sqrt{3}$, where the sign is chosen such that the maximum utility is $\bar{\mathcal{U}}^* = N \lambda_k^2 / 3\sqrt{3}$. In other words, utility maximization encourages dormant attention heads to align their key and query vectors with the dominant principal component of the input covariance not yet captured by any active head. This creates a race condition among dormant heads, where the first to reach the activation threshold, measured by their accumulated utility, becomes active and learns the corresponding component. Assuming instantaneous alignment of the key and query vectors, we can lower bound the jump time for the next head to activate, as shown in Equation (7).

**Cost minimization.** During cost minimization, because $v_1, \ldots, v_d$ form an orthonormal basis, the updates for each attention head are decoupled. Thus, we only need to focus on how the magnitude of the newly active head changes. Specifically, we determine the magnitude $A_k$ of the newly learned function component, given by $f_k(X; \theta_k)_{d+1, N+1} = A_k \sum_{n=1}^{N} y_n x_n^\mathsf{T} v_k v_k^\mathsf{T} x_{N+1}$. Solving $\frac{\partial \mathcal{L}(A_k)}{\partial A_k} = 0$, we find that the optimal magnitude is $A_k = \frac{1}{\text{tr}\Sigma_{xx} + (N+1)\lambda_k}$, from which we can derive the expression for the prediction and loss level after the $k^{\text{th}}$ iteration of AGF, as shown in Equation (7).

**AGF in action: principal component regression.** At each iteration, the network projects the input onto a newly selected principal component of $\Sigma_{xx}$ and fits a linear regressor along that direction. Let $\mu^{(l)} = \max_{i \in \mathcal{D}} \|\theta_i(\tau^{(l)})\|$ for $l < k$ and $\eta_\alpha = \alpha^{-1}$. This process yields the sequence,

$$\hat{y}_{N+1}^{(k)} = \sum_{i,n=1}^{k,N} \frac{y_n x_n^\mathsf{T} v_i v_i^\mathsf{T} x_{N+1}}{\text{tr}\Sigma_{xx} + (N+1)\lambda_i}, \quad \ell^{(k)} = \frac{\text{tr}\Sigma_{xx}}{2} - \sum_{i=1}^{k} \frac{N\lambda_i/2}{\frac{\text{tr}\Sigma_{xx}}{\lambda_i} + N+1}, \quad \tau^{(k)} \gtrsim \tau^{(l)} + \frac{\eta_\alpha^{-1}}{\mu^{(l)}} \frac{\sqrt{3}}{N\lambda_k^2}, \quad (7)$$

which recovers the results derived in Zhang et al. [32] and provides an excellent approximation to the gradient flow dynamics (see Figure 5).

## 5 AGF Predicts the Emergence of Fourier Features in Modular Addition

In previous sections, we showed how AGF unifies prior analyses of feature learning in linear networks. We now consider a novel nonlinear setting: a two-layer quadratic network trained on modular addition. Originally proposed as a minimal setting to explore emergent behavior [89], modular addition has since become a foundational setup for mechanistic interpretability. Prior work has shown that networks trained on this task develop internal Fourier representations and use trigonometric identities to implement addition as rotations on the circle [6, 90, 91]. Similar Fourier features have been observed in networks trained on group composition tasks [9], and in large pre-trained language models performing arithmetic [92, 93]. Despite extensive empirical evidence for the universality of Fourier features in deep learning, a precise theoretical explanation of their emergence remains open. Recent work has linked this phenomenon to the average gradient outer product framework [94], the relationship between symmetry and irreducible representations [95], implicit maximum margin biases of gradient descent [96], and the algebraic structure of the solution space coupled with a simplicity bias [97]. Here, we leverage AGF to unveil saddle-to-saddle dynamics, where each saddle corresponds to the emergence of a Fourier feature in the network (see Figure 6).

We consider a setting similar to [90, 96, 97], but with more general input encodings. Given $p \in \mathbb{N}$, the ground-truth function is $y \colon \mathbb{Z}/p \times \mathbb{Z}/p \to \mathbb{Z}/p, (a, b) \mapsto a + b \bmod p$, where $\mathbb{Z}/p = \{0, \ldots, p-1\}$ is the (additive) Abelian group of integers modulo $p$. Given a vector $x \in \mathbb{R}^p$, we consider the encoding $\mathbb{Z}/p \to \mathbb{R}^p, a \mapsto a \cdot x$, where $\cdot$ denotes the action of the cyclic permutation group of order $p$ on $x$ (which cyclically permutes the components of $x$). The modular addition task with this encoding consists of mapping $(a \cdot x, b \cdot x)$ to $(a + b) \cdot x$. For $x = e_0$, our encoding coincides with the one-hot encoding $a \mapsto e_a$ studied in prior work. We consider a two-layer neural network with quadratic activation function $\sigma(z) = z^2$, where each neuron is parameterized by $\theta_i = (u_i, v_i, w_i) \in \mathbb{R}^{3p}$ and computes the function $f_i(a \cdot x, b \cdot x; \theta_i) = (\langle u_i, a \cdot x \rangle + \langle v_i, b \cdot x \rangle)^2 w_i$. The network is composed of the sum of $H$ neurons and is trained over the entire dataset of $p^2$ pairs $(a, b)$. We make some technical assumptions in order to simplify the analysis. First, since the network contains no bias term, we assume that the data is centered, i.e., we subtract $(\langle x, \mathbf{1} \rangle / p)\mathbf{1}$ from $x$. Second, in order to encourage

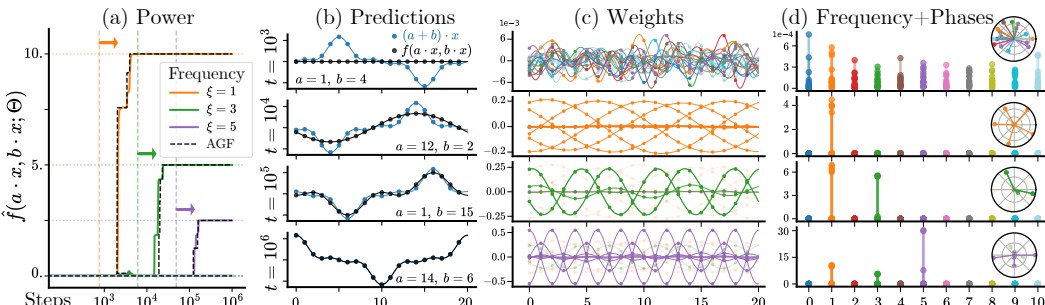

**Figure 6: Stepwise Fourier decomposition.** We train a two-layer quadratic network on a modular addition task with $p = 20$, using a template vector $x \in \mathbb{R}^p$ composed of three cosine waves: $\hat{x}[1] = 10$, $\hat{x}[3] = 5$, and $\hat{x}[5] = 2.5$. (a) Output power spectrum over time. The network learns the task by sequentially decomposing $x$ into its Fourier components, acquiring dominant frequencies first. Colored solid lines are gradient descent, black dashed line is AGF run numerically from the same initialization. (b) Model outputs on selected inputs at four training steps, showing progressively accurate reconstructions of the template. (c) Output weight vector $w_i$ for all $H = 18$ neurons and (d) their frequency spectra and dominant phase. Neurons are color-coded by dominant frequency. As predicted by the theory, the neurons group by frequency, while distributing their phase shifts.

sequential learning of frequencies, we assume that all the non-conjugate Fourier coefficients of $x$ have distinct magnitude: $|\hat{x}[k]| \neq |\hat{x}[k']|$ for $k \neq \pm k' \pmod{p}$. Third, in order to avoid dealing with the Nyquist frequency $k = p/2$, we assume that $p$ is odd. We now describe how AGF applies to this setting (see Appendix F for proofs). Here, $\hat{\cdot}$ denotes the Discrete Fourier Transform (DFT).

**Utility maximization.** First, we show how the initial utility can be expressed entirely in the frequency domain of the parameters $\hat{\Theta}$. For a dormant neuron parameterized by $\theta = (u, v, w)$, the initial utility is $\mathcal{U} = (2/p^3) \sum_{k \in [p] \setminus \{0\}} |\hat{x}[k]|^2 \hat{u}[k] \hat{v}[k] \overline{\hat{w}[k]} \hat{x}[k]$. We then argue that the unit vectors maximizing $\mathcal{U}$ align with the dominant harmonic of $\hat{x}$:

**Theorem 5.1.** *Let $\xi$ be the frequency that maximizes $|\hat{x}[k]|$, $k = 1, \ldots, p-1$, and denote by $s_x$ the phase of $\hat{x}[\xi]$. Then the unit vectors $\theta_* = (u_*, v_*, w_*)$ maximizing the utility function $\mathcal{U}$ are:*

$$u_*[a] = A_p \cos(\omega_\xi a + s_u), \quad v_*[b] = A_p \cos(\omega_\xi b + s_v), \quad w_*[c] = A_p \cos(\omega_\xi c + s_w), \quad (8)$$

*where $a, b, c \in [p]$ are indices, $s_u, s_v, s_w \in \mathbb{R}$ are phase shifts satisfying $s_u + s_v \equiv s_w + s_x \pmod{2\pi}$, $A_p = \sqrt{2/(3p)}$ is the amplitude, and $\omega_\xi = 2\pi\xi/p$ is the frequency. Moreover, $\mathcal{U}$ has no other local maxima and achieves a maximal value of $\bar{\mathcal{U}}^* = \sqrt{2/(27p^3)}|\hat{x}[\xi]|^3$.*

Therefore, after utility minimization, neurons specialize to unique frequencies. Now that we know the maximal utility, we can estimate the jump time by assuming instantaneous alignment as done in Section 4, resulting in the lower bound shown in Equation (9).

**Cost minimization.** To study cost minimization, we consider a regime in which a group $\mathcal{A}$ of $N \leq H$ neurons activates simultaneously, each aligned to the harmonic of frequency $\xi$. While this is a technical simplification of AGF, which activates a single neuron per iteration, it allows us to analyze the collective behavior more directly. We additionally assume that once aligned, the neurons in $\mathcal{A}$ remain aligned under the gradient flow (see Appendix F.2.1 for a discussion of possible "resonant" escape directions). We then analyze cost minimization for a configuration $\Theta_{\mathcal{A}} = (u^i, v^i, w^i)_{i=1}^N$ of aligned neurons with arbitrary amplitudes and phase shifts, and prove the following result.

**Theorem 5.2.** *The loss function satisfies the lower bound $\mathcal{L}(\Theta_{\mathcal{A}}) \geq \|x\|^2/2 - \langle x, \mathbf{1} \rangle^2/(2p) - |\hat{x}[\xi]|^2/p$, which is tight for $N \geq 6$. When the bound is achieved, the network learns the function $f(a \cdot x, b \cdot x; \Theta_{\mathcal{A}}) = (2|\hat{x}[\xi]|/p)(a+b) \cdot \chi_\xi$, where $\chi_\xi[c] = \cos(2\pi\xi c/p + s_x)$, which leads to the new utility function for the remaining dormant neurons: $\mathcal{U} = (2/p^3) \sum_{k \in [p] \setminus \{0, \pm\xi\}} |\hat{x}[k]|^2 \hat{u}[\xi] \hat{v}[\xi] \overline{\hat{w}[\xi]} \hat{x}[\xi]$.*

Put simply, the updated utility function after the first iteration of AGF has the same form as the old one, but with the dominant frequency $\xi$ of $x$ removed (together with its conjugate frequency $-\xi$). Therefore, at the second iteration of AGF, another group of neurons aligns with the harmonic of the second dominant frequency of $x$. Lastly, we argue via orthogonality that the groups of neurons aligned with different harmonics optimize their loss functions independently, implying that at each iteration of AGF, another group of neurons learns a new Fourier feature of $x$.

**AGF in action: greedy Fourier decomposition.** Taken together, we have shown that a two-layer quadratic network with hidden dimension $H \geq 3p$ trained to perform modular addition with a centered encoding vector $x \in \mathbb{R}^p$ sequentially learns frequencies $\xi_1, \xi_2, \dots$, ordered by decreasing $|\hat{x}[\xi]|$. After learning $k \geq 0$ frequencies, the function, level, and next jump time are:

$$f^{(k)}(a \cdot x, b \cdot x) = \sum_{i=1}^{k} \frac{2|\hat{x}[\xi_i]|}{p}(a+b) \cdot \chi_{\xi_i}, \ell^{(k)} = \sum_{i>k} \frac{|\hat{x}[\xi_i]|^2}{p}, \tau^{(k)} \gtrsim \tau^{(l)} + \frac{\eta_\alpha^{-1}}{\mu^{(l)}} \frac{\sqrt{3}p^{\frac{3}{2}}}{\sqrt{2}|\hat{x}[\xi_{k+1}]|^3}, \quad (9)$$

where $\mu^{(l)} = \max_{i \in \mathcal{D}} \|\theta_i(\tau^{(l)})\|$ and learning rate $\eta_\alpha = \alpha^{-1}$. Simulating AGF numerically, we find this sequence closely approximates the gradient flow dynamics (see Figure 6).

**Extensions to other algebraic tasks.** Our analysis of modular addition can naturally extend to a broader class of algebraic problems. First, one can consider modular addition over multiple summands, defined by the map $y: (\mathbb{Z}/p)^k \to \mathbb{Z}/p$, $(a_i)_{i=1}^k \mapsto a_1 + \cdots + a_k \pmod{p}$. Using a higher-degree activation function $\sigma(x) = x^k$, the arguments for utility maximization should carry over, while the cost minimization step might be more subtle. Second, one can replace modular integers with an arbitrary finite group and study a network trained to learn the group multiplication map. Non-commutative groups introduce technical challenges due to the involvement of higher-dimensional unitary representations in their Fourier analysis. We leave a detailed analysis of these extensions to future work.

## 6 Discussion, Limitations, and Future Work

In this work we introduced Alternating Gradient Flows (AGF), a framework modeling feature learning in two-layer neural networks as an alternating two-step process: maximizing a utility function over dormant neurons and minimizing a cost function over active ones. We showed how AGF converges to gradient flow in diagonal linear networks, recovers prior saddle-to-saddle analyses in linear networks, and extends to quadratic networks trained on modular addition. While these findings highlight AGF's utility as an *ansatz* for feature learning, it remains open whether its correspondence to gradient flow always holds in the vanishing initialization limit. Proving such a conjecture is theoretically challenging. Empirical validation is also difficult because it requires taking both the initialization scale and learning rate to zero. Moreover, this conjecture may simply fail to hold in more general settings. On natural data tasks, loss curves are often not visibly stepwise even at very small initialization scales (see Figure 7), suggesting there may be limitations to AGF. That said, if many dormant neurons reach their activation thresholds in close succession, their cost-minimization phases could interleave, causing the aggregate loss to appear smooth. Recent works reconciling the emergent capabilities of large language models with their neural scaling laws have made similar suggestions [98–102]. However, there are feature learning regimes in

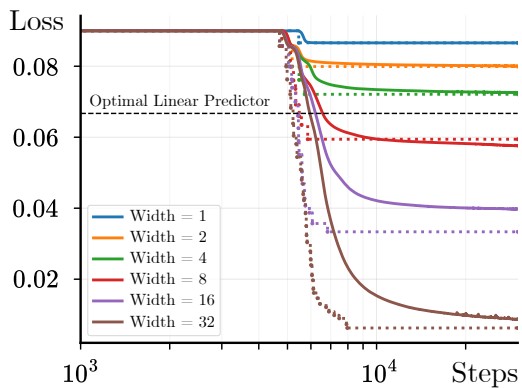

Figure 7: **From a staircase to a slide in a two-layer ReLU network.** Training loss for a two-layer ReLU network on a subset of CIFAR-10 under varying hidden widths from an extremely small initialization. Solid lines show gradient descent dynamics and dotted lines show the sequence produced by a numerical implementation of AGF from the *same* initialization. At small widths, gradient descent loss curves are stepwise and AGF tracks the jumps closely. As width increases, cost-minimization phases overlap and multiple dormant neurons activate in close succession, producing a smoother *slide*-like loss curve; accordingly, the correspondence with AGF weakens. All experimental details are available at a Github code base.

two-layer networks—such as those studied in Saad and Solla [46], Goldt et al. [47], Arnaboldi et al. [51]—that display saddle-to-saddle behavior due to population-level transitions not captured by AGF. Connecting AGF to these multi-index models and teacher–student settings (see Appendix A for a review) remains a key direction for future work. Lastly, the central limitation of the framework is its focus on two-layer networks, leaving open how it might generalize to deeper and more realistic architectures. Possible extensions include leveraging recent analyses of modularity in deep networks [103–105] and adapting insights from the early alignment dynamics of deep networks near the origin [59, 106]. All together, our results suggest that AGF offers a promising step towards a deeper understanding of *what* features neural networks learn and *how*.

## Acknowledgments and Disclosure of Funding

We thank Clémentine Dominé, Jim Halverson, Boris Hanin, Christopher Hillar, Alex Infanger, Arthur Jacot, Mason Kamb, David Klindt, Florent Krzakala, Zhiyuan Li, Sophia Sanborn, Nati Srebro, and Yedi Zhang for helpful discussions. Daniel thanks the Open Philanthropy AI Fellowship for support. Giovanni is partially supported by the Wallenberg AI, Autonomous Systems and Software Program (WASP) funded by the Knut and Alice Wallenberg Foundation. Surya and Feng are partially supported by NSF grant 1845166. Surya thanks the Simons Foundation, NTT Research, an NSF CAREER Award, and a Schmidt Science Polymath award for support. Nina is partially supported by NSF grant 2313150 and the NSF grant 240158. This work was supported in part by the U.S. Army Research Laboratory and the U.S. Army Research Office under Contract No. W911NF-20-1-0151.

## Author Contributions

Daniel, Nina, Giovanni, and James are primarily responsible for developing the AGF framework in Section 2. Daniel is primarily responsible for the analysis of diagonal and fully-connected linear networks in Sections 3 and 4. Feng is primarily responsible for the analysis of the attention-only linear transformer in Section 4. Daniel and Giovanni are primarily responsible for the analysis of the modular addition task in Section 5. Dhruva is primarily responsible for an implementation of AGF used in the empirics. All authors contributed to the writing of the manuscript.

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

# A  Additional Related Work

## A.1  What is a feature?

Here, we briefly review definitions of a *feature* used in the mechanistic interpretability and deep learning theory literature:

- In mechanistic interpretability, terms such as *feature*, *circuit*, and *motif* are used to describe network components—such as directions in activation space, subnetworks of weights, or recurring activation motifs—that correspond to human-interpretable concepts or functions within the model's computation [10]. While this literature has led to many insights into interpretability, these definitions generally lack precise mathematical formalization.

- In deep learning theory, features and feature learning are defined in terms of the *neural tangent kernel (NTK)*. For a network $f(x; \theta)$, the NTK is given by the kernel $K(x, x'; \theta) = \langle \nabla_\theta f(x; \theta), \nabla_\theta f(x'; \theta) \rangle$, formed by the Jacobian feature map $\nabla_\theta f(x; \theta)$. Feature learning—also referred to as *rich learning*—occurs when this kernel evolves during training. While this definition is mathematically precise, it offers limited interpretability.

In this work, we implicitly adopt the notion of features from the deep learning theory perspective, but consider learning settings often studied in mechanistic interpretability.

## A.2  Analysis of feature learning in two-layer networks

Here, we discuss related theoretical approaches to studying feature learning in two-layer networks. These approaches operate in distinct regimes—mean-field/infinite-width, teacher–student at high-dimensional (thermodynamic) limits, and single-step analyses—whereas AGF focuses on the *vanishing initialization* limit for two-layer networks. Our aim is to position AGF as *complementary to*, not a replacement for, these approaches.

**Mean-field analysis.** An early line of work in the study of feature learning analyzes the population dynamics of two-layer neural networks under the *mean-field* parameterization, yielding analytic characterizations of training dynamics in the infinite-width limit [40–43]. These ideas have been extended to deep networks via the *tensor program* framework, culminating in the *maximal update parametrization* ($\mu$P) [54, 107]. Related analyses have also been obtained through self-consistent dynamical mean-field theory [108]. Although feature learning occurs in this regime, these analyses primarily address how to preserve feature learning when changing network depth and width, rather than elucidating what specific features are learned or how they emerge.

**Teacher-student setup.** The two-layer teacher–student framework provides a precise setting for studying feature learning by analyzing how a student network trained on supervised examples generated from a fixed teacher recovers the structure of the teacher. Foundational work by Saad and Solla [44, 45, 46] provided closed-form dynamics for online gradient descent in soft committee machines trained on i.i.d. Gaussian inputs in the high-dimensional limit, deriving deterministic differential equations governing a set of order parameters. Goldt et al. [47], Veiga et al. [48] extended this dynamical theory to study the evolution of generalization error of one-pass SGD under an arbitrary learning rate, and a range of hidden layer widths. Further developments have analyzed conditions for the absence of spurious minima [49], extended the framework to capture more realistic data settings [50], and provided a unified perspective connecting to the infinite hidden width limit of the mean-field regime [51]. Much of this analysis use techniques from statistical physics—including the replica method and approximate message passing—to characterize the generalization error and uncover sharp phase transitions and statistical-to-computational gaps in teacher–student models, discussed further in Aubin et al. [109], Barbier et al. [110].

**Multi-index models.** Similar to the teacher–student setting, *multi-index models* provide a structured framework in which a neural network is trained on data whose labels depend on a nonlinear function of low-dimensional projections of high-dimensional inputs. Feature learning in this setting is governed by the network's ability to align with the relevant low-dimensional subspace. This setup often gives rise to characteristic staircase-like training dynamics [111, 36, 37, 60, 33–35]. Prior analyses have primarily focused on characterizing generalization and sample complexity, and recent work has established precise theoretical and computational thresholds for weak learnability in high-dimensional multi-index problems [112, 113]. Other studies have demonstrated that reusing batches in gradient descent can help networks overcome information bottlenecks, accelerating the learning of relevant

projections [38, 39]. Additionally, incorporating depth into these models has been shown to provide computational advantages for learning hierarchical multi-index functions [114].

**Single gradient step.** A recent line of work has focused on understanding feature learning in neural networks after just a single gradient step. Empirically, it has been observed that with large initial learning rates, models can undergo "catapult" dynamics, where sudden spikes in the training loss accelerate feature learning and improve generalization [115, 116]. Ba et al. [117] initiated the precise analysis of feature learning after a single gradient step by showing, in a high-dimensional single-index setting, how the first update to the first-layer weights can dramatically improve prediction performance beyond random features, depending on the learning rate scaling. This single-step analysis has been extended to characterize the asymptotic behavior of the generalization error [118], establish an equivalence between the updated features and an isotropic spiked random feature model [119], and investigate the effects of multiple gradient steps [120]. Collectively, these works highlight that even a single gradient step can induce meaningful feature learning in two-layer networks.

**Distinct phases of alignment and growth in ReLU networks.** Maennel et al. [56] first observed a striking phenomenon in two-layer ReLU networks trained from small initializations: the first-layer weights concentrate along fixed directions determined by the training data, regardless of network width. Subsequent studies of piecewise linear networks have refined this effect by imposing structural constraints on the training data, such as orthogonally separable [24, 26, 28], linearly separable and symmetric [25], pair-wise orthonormal [27], correlated [121], small angle [30], binary and sparse [29]. Across these analyses, a consistent observation is that the learning dynamics involve distinct learning phases separated by timescales: a fast alignment phase driven by directional movement and a slow fitting phase driven by radial movement [60, 61]. Concurrently, Bantzis et al. [122] use a closely related directional–radial decomposition to identify the optimal escape directions from the saddle at the origin in deep ReLU networks.

# B Derivations in Alternating Gradient Flows Framework

In this section, we proivde details and motivations for the general framework of AGF. All code and experimental details are available at `github.com/danielkunin/alternating-gradient-flows`.

## B.1 Deriving dormant dynamics near a saddle point in terms of the utility function

At any point in parameter space the gradient of the MSE loss with respect to the parameters of a neuron can be expressed as

$$-\nabla_{\theta_i}\mathcal{L}(\Theta) = \nabla_{\theta_i}\mathbb{E}_x\left[\langle f_i(x;\theta_i), r(x;\Theta)\rangle\right],\tag{10}$$

where $r(x;\Theta) = y(x) - \sum_i f_i(x;\theta_i)$ is a residual that depends on all neurons. Given a partition of neurons into two sets, an active $\mathcal{A}$ and dormant $\mathcal{D}$ set, then the gradient for a dormant neuron $i \in \mathcal{D}$ can be decomposed into two terms:

$$-\nabla_{\theta_i}\mathcal{L}(\Theta) = \nabla_{\theta_i}\mathbb{E}_x\left[\left\langle f_i(x;\theta_i), y(x) - \sum_{j \in \mathcal{A}} f_j(x;\theta_j)\right\rangle\right] - \nabla_{\theta_i}\mathbb{E}_x\left[\left\langle f_i(x;\theta_i), \sum_{j \in \mathcal{D}} f_j(x;\theta_j)\right\rangle\right].\tag{11}$$

The first term is the gradient of the utility $\nabla_\theta \mathcal{U}_i$ as defined in Equation (1). The second term only depends on the dormant neurons and is thus negligible when all the dormant neurons are small in norm. Taken together, in the vicinity of a saddle point of the loss defined by an active and dormant set, the dormant neurons are driven approximately by gradient flow maximizing their utility:

$$\frac{d}{dt}\theta_i \approx \eta\nabla_\theta\mathcal{U}_i.\tag{12}$$

This observation motivates the utility and our AGF ansatz for gradient flow. To formally prove that AGF converges to gradient flow in the limit of vanishing initialization one might consider using tools like *Hartman–Grobman theorem* to make this step rigorous.

**Directional and radial expansion of the dynamics.** Using this approximation, we can further decompose the dynamics of a dormant neuron near a saddle point into a directional and radial component:

$$\frac{d}{dt}\frac{\theta_i}{\|\theta_i\|} = \eta\frac{\mathbf{P}^\perp_{\theta_i}}{\|\theta_i\|}\nabla_\theta\mathcal{U}_i, \quad \frac{d}{dt}\|\theta_i\| = \eta\frac{1}{\|\theta_i\|}\langle\theta_i, \nabla_\theta\mathcal{U}_i\rangle,\tag{13}$$

where $\mathbf{P}^\perp_{\theta_i} = \left(\mathbf{I}_m - \frac{\theta_i\theta_i^\intercal}{\|\theta_i\|^2}\right)$ is a projection matrix. When $\sigma(\cdot)$ is a homogeneous function of degree $k$, then the utility is a homogeneous function of degree $\kappa = k + 1$. Using Euler's homogeneous function theorem, Equation (13) then simplifies to Equation (2). When $\sigma(\cdot)$ is not homogeneous, we can Taylor expand the utility around the origin, such that the utility coincides approximately with a homogeneous function of degree $\kappa$, where $\kappa$ is the leading order of the Taylor expansion.

**Dynamics of normalized utility.** An interesting observation, that we will use in Appendix C, is that the normalized utility function follows a Riccati-like differential equation

$$\frac{d}{dt}\bar{\mathcal{U}}_i = \eta\kappa^2\|\theta_i\|^{\kappa-2}\left(\frac{\|\nabla_\theta\bar{\mathcal{U}}_i\|^2}{\kappa^2} - \bar{\mathcal{U}}_i^2\right).\tag{14}$$

In general, this ODE is coupled with the dynamics for both the direction and norm, except when $\kappa = 2$ for which the dependency on the norm disappears.

## B.2 Deriving the jump time

To compute $\tau_{i_*}$, we use Equation (2) to obtain the time evolution of the norms of the dormant neurons. For $i \in \mathcal{D}$, we obtain:

$$\|\theta_i(t)\| = \begin{cases} \|\theta_i(0)\|\exp\left(\mathcal{S}_i(t)\right) & \text{if } \kappa = 2, \\ \left(\|\theta_i(0)\|^{2-\kappa} + (2-\kappa)\mathcal{S}_i(t)\right)^{\frac{1}{2-\kappa}} & \text{if } \kappa > 2, \end{cases}\tag{15}$$

where we have defined the *accumulated utility* as the path integral $\mathcal{S}_i(t) = \int_0^t \kappa\bar{\mathcal{U}}_i(s)\mathrm{d}s$ of the normalized utility. We find $\tau_i$ as the earliest time at which neuron $i$ satisfies $\|\theta_i(\tau_i)\| > 1$:

$$\tau_i = \inf\{t > 0 \mid \mathcal{S}_i(t) > c_i/\eta\}, \quad \text{where } c_i = \begin{cases} -\log\|\theta_i(0)\| & \text{if } \kappa = 2, \\ -\frac{\|\theta_i(0)\|^{2-\kappa}-1}{2-\kappa} & \text{if } \kappa > 2. \end{cases}\tag{16}$$

Note that the expression for $c_i$ is continuous at $\kappa = 2$.

**Lower bound on jump time.** When applying this framework to analyze dynamics, computing the exact jump time can be challenging due to the complexity of integrating $\bar{\mathcal{U}}$. In such cases, we resort on the following lower bound as a useful analytical approximation. Let $\mathcal{U}_i^*$ be the maximal value of the utility, or at least an upper bound on it. Since $\mathcal{S}_i(t) \leq \kappa \mathcal{U}_i^* t$, we deduce:

$$\tau_i \geq \begin{cases} -\frac{\log \|\theta_i(0)\|}{2\mathcal{U}_i^*} & \text{if } \kappa = 2, \\ -\frac{\|\theta_i(0)\|^{2-\kappa}}{\kappa(2-\kappa)\mathcal{U}_i^*} & \text{if } \kappa > 2. \end{cases} \tag{17}$$

## B.3 Active neurons can become dormant again

In the cost minimization phase of AGF, active neurons can become dormant. Here, we motivate and discuss this phenomenon. Intuitively, this happens when the GF trajectory brings an active neuron close to the origin. Due to the preserved quantities of GF for homogeneous activation functions, the trajectory of active neurons is constrained. Thus, this phenomenon can occur only in specific scenarios, as quantified by the following result.

**Lemma B.1.** *Suppose that $\Theta$ evolves via the GF of $\mathcal{L}$, and let $c = (\kappa - 1)\|a_i(0)\|^2 - \|w_i(0)\|^2$. The norm $\|\theta_i(t)\|^2$ satisfies the lower bound,*

$$\|\theta_i(t)\|^2 \geq \max\left(-c, \frac{c}{\kappa - 1}\right), \tag{18}$$

*which holds with equality when either $\|w_i(t)\|^2 = 0$ for $c \geq 0$ or $\|a_i(t)\|^2 = 0$ for $c < 0$.*

*Proof.* For homogeneous activation functions of degree $\kappa$, it is well known that the quantity $c = (\kappa - 1)\|a_i(t)\|^2 - \|w_i(t)\|^2$ is preserved along trajectories of gradient flow. Since $\|\theta_i(t)\|^2 = \|a_i(t)\|^2 + \|w_i(t)\|^2$, we have:

$$\begin{aligned} \|\theta_i(t)\|^2 &= -c + \kappa \|a_i(t)\|^2 \geq -c, \\ \|\theta_i(t)\|^2 &= \frac{c}{\kappa - 1} + \frac{\kappa}{\kappa - 1}\|w_i(t)\|^2 \geq \frac{c}{\kappa - 1}. \end{aligned} \tag{19}$$

The claim follows by combining the above inequalities. $\qquad\square$

Since at vanishing initialization $(\kappa - 1)\|a_i\|^2 \sim \|w_i\|^2$, we have $c \sim 0$. Therefore, an active neuron can approach the origin during the cost minimization phase. If this happens, the neuron becomes dormant. When a neuron becomes dormant again it does so with an accumulated utility $\mathcal{S}_i(t) = \frac{c_i}{\eta}$. See Figure 8 for an example of a diagonal linear network where this behavior is observable.

## B.4 Instantaneous alignment in the vanishing initialization limit

In this section, we revisit the separation of the dynamics dictated by revisit Equation (2). In particular, we wish to discuss the conditions under which the directional dynamics dominates the radial one, resulting, at the vanishing initialization limit, in instantaneous alignment during the utility maximization phase. To this end, we establish the following scaling symmetry with respect to the initialization scale factor $\alpha$.

**Theorem B.2.** *Suppose $\theta_i(t) = f_i(t)$ solves the initial value problem defined by Equation (2) with initial condition $\theta_i(0) = \theta_{0,i}$. Then for all $\alpha > 0$, the scaled solution $\theta_i(t) = \alpha f_i(\alpha^{\kappa-2}t)$ solves the initial value problem with initial condition $\theta_i(0) = \alpha\theta_{0,i}$.*

*Proof.* First, we consider the angular dynamics:

$$\begin{aligned} \frac{d}{dt}\bar{\theta}_i &= \frac{d}{dt}\frac{f_i(\alpha^{\kappa-2}t)}{\|f_i(\alpha^{\kappa-2}t)\|} \\ &= \alpha^{\kappa-2}\frac{d}{ds}\frac{f_i(s)}{\|f_i(s)\|}\bigg|_{s=\alpha^{\kappa-2}t} \\ &= \alpha^{\kappa-2}\|f_i(\alpha^{\kappa-2}t)\|^{\kappa-2}\mathbf{P}_{\bar{\theta}_i}^{\perp}\nabla_\theta\mathcal{U}_i(\bar{\theta}_i(\alpha^{\kappa-2}t); r) \\ &= \|\alpha f_i(\alpha^{\kappa-2}t)\|^{\kappa-2}\mathbf{P}_{\bar{\theta}_i}^{\perp}\nabla_\theta\mathcal{U}_i(\bar{\theta}_i(\alpha^{\kappa-2}t); r) \end{aligned}$$

Therefore, $\theta_i(t) = \alpha f_i(\alpha^{\kappa-2}t)$ satisfies the first identity in Equation (2) above. Next, consider the norm dynamics:

$$
\begin{aligned}
\frac{d}{dt}\|\theta_i\| &= \frac{d}{dt}\|\alpha f_i(\alpha^{\kappa-2}t)\| \\
&= \alpha^{\kappa-1}\frac{d}{ds}\|f_i(s)\|\bigg|_{s=\alpha^{\kappa-2}t} \\
&= \alpha^{\kappa-1}\kappa\|f_i(\alpha^{\kappa-2}t)\|^{\kappa-1}\mathcal{U}_i(\bar{\theta}_i(\alpha^{\kappa-2}t);r) \\
&= \kappa\|\alpha f_i(\alpha^{\kappa-2}t)\|^{\kappa-1}\mathcal{U}_i(\bar{\theta}_i(\alpha^{\kappa-2}t);r)
\end{aligned}
$$

Hence, $\theta_i(t) = \alpha f_i(t\alpha^{\kappa-2})$ also satisfies the second equation. Finally, verifying the initial conditions gives, $\theta_i(0) = \alpha f_i(0) = \alpha\theta_{0,i}$. Therefore, $\theta_i(t) = \alpha f_i(t\alpha^{\kappa-2})$ satisfies the given initial condition. $\square$

Theorem B.2 establishes a transformation among the angular dynamics of different initialization scales. Specifically, for a given point $(s, \frac{f_i(s)}{\|f_i(s)\|})$ in the solution trajectory of the initial value problem with initial condition $\theta_i(0) = \theta_{0,i}$, the corresponding point in the initial value problem with scaled initial condition $\theta_i(0) = \alpha\theta_{0,i}$ is $(s\alpha^{2-\kappa}, \frac{f_i(s)}{\|f_i(s)\|})$. This correspondence reveals that the angular alignment process is effectively slowed down by a factor of $\alpha^{\kappa-2}$ with initialization scale $\alpha$.

Next, we examine the alignment speed in accelerated time in the limit of $\alpha \to 0$. In this regime, the asymptotics of Equation (16) is given by,

$$
c_\kappa(\alpha) \sim \begin{cases} -\log\alpha & \text{if } \kappa = 2, \\ \alpha^{2-\kappa} & \text{if } \kappa > 2, \end{cases} \tag{20}
$$

Therefore, in the accelerated time, the alignment speed is effectively scaled by a scaling factor of:

$$
\gamma(\alpha) := \frac{c_\kappa(\alpha)}{\alpha^{\kappa-2}} = \begin{cases} -\log\alpha & \text{if } \kappa = 2, \\ 1 & \text{if } \kappa > 2, \end{cases} \tag{21}
$$

And,

$$
\lim_{\alpha\to 0}\gamma(\alpha) = \begin{cases} +\infty & \text{if } \kappa = 2, \\ 1 & \text{if } \kappa > 2, \end{cases} \tag{22}
$$

The limiting behavior implies that the alignment is instantaneous in accelerated time for almost all initialization $\theta_{0,i}$ if and only if $\kappa = 2$.

## B.5 A neuron-specific adaptive learning rate yields instantaneous alignment

As discussed above, instantaneous alignment of dormant neurons with their local utility-maximizing directions occurs only when $\kappa = 2$. For higher-order activations ($\kappa > 2$), the directional dynamics acquire a norm-dependent factor $\|\theta_i\|^{\kappa-2}$ that slows their angular evolution. Consequently, directional and radial dynamics no longer decouple, even in the vanishing initialization limit, and dormant neurons rotate gradually rather than aligning instantaneously.

This dependence can be removed by introducing a neuron-specific adaptive learning rate

$$
\eta_i = \|\theta_i\|^{2-\kappa}\eta, \tag{23}
$$

where $\eta$ is a global base rate. When $\kappa = 2$, this scaling has no effect and all neurons evolve at the same rate. For $\kappa > 2$, however, neurons with small norm ($\|\theta_i\| < 1$) are accelerated, while those with large norm ($\|\theta_i\| > 1$) are slowed down. This rescaling effectively reparametrizes time so that the directional dynamics become norm-independent while the radial dynamics remain norm-dependent. Substituting this adaptive rate into Equation (2) yields an evolution equivalent to that of the $\kappa = 2$ case, resulting in the decoupling between directional and radial dynamics in the vanishing-initialization limit for all $\kappa$. In practice, this scaling acts analogously to a form of neuron-wise adaptive optimization—resembling RMSProp or Adam—but derived directly from the analytical structure of the $\kappa$-homogeneous gradient flow.

## C  Complete Proofs for Diagonal Linear Networks

In this section, we derive the AGF dynamics for the two-layer diagonal linear network; see Section 3 for the problem setup and notation. Throughout this section, we assume the initialization $\theta_i(0) = (\sqrt{2}\alpha, 0)$, following the convention in Pesme and Flammarion [18].

### C.1  Utility maximization

**Lemma C.1.** *After the $k^{\text{th}}$ iteration of AGF, the utility for the $i^{\text{th}}$ neuron is*

$$\mathcal{U}_i\left(\theta_i; r^{(k)}\right) = -u_i v_i \nabla_{\beta_i} \mathcal{L}\left(\beta^{(k)}\right), \tag{24}$$

*which is maximized on the unit sphere $\|\theta_i\| = 1$ when $\text{sgn}(u_i v_i) = -\text{sgn}\left(\nabla_{\beta_i}\mathcal{L}\left(\beta^{(k)}\right)\right)$ and $|u_i| = |v_i|$ resulting in a maximal utility value of $\bar{\mathcal{U}}_i^* = \frac{1}{2}\left|\nabla_{\beta_i}\mathcal{L}\left(\beta^{(k)}\right)\right|.$*

*Proof.* Substituting the residual $r_j^{(k)} = y_j - \left(X^\mathsf{T}\beta^{(k)}\right)_j$ into the definition of the utility function:

$$\mathcal{U}_i(\theta_i; r^{(k)}) = \frac{1}{n}\sum_{j=1}^{n} u_i v_i X_{ij} r_j = -u_i v_i \nabla_{\beta_i}\mathcal{L}\left(\beta^{(k)}\right). \tag{25}$$

Observe that the expression for $\mathcal{U}_i$ is linear in $u_i v_i$. Therefore, under the normalization constraint $\|\theta_i\| = 1$, the utility is maximized when $|u_i| = |v_i|$ and $\text{sgn}(u_i v_i) = -\text{sgn}\left(\nabla_{\beta_i}\mathcal{L}\left(\beta^{(k)}\right)\right)$, yielding the maximal value $\bar{\mathcal{U}}_i^* = \frac{1}{2}\left|\nabla_{\beta_i}\mathcal{L}\left(\beta^{(k)}\right)\right|.$ $\qquad\square$

**Lemma C.2.** *At any time $t$ during AGF, the parameters of the $i^{\text{th}}$ neuron satisfy:*

$$u_i(t)^2 - v_i(t)^2 = 2\alpha, \qquad 2u_i(t)v_i(t) = \pm\sqrt{\|\theta_i(t)\|^2 - 4\alpha^4} \tag{26}$$

*Proof.* Both the utility and loss are invariant under the transformation $(u_i, v_i) \mapsto (g u_i, g^{-1} v_i)$ for any $g \neq 0$. This continuous symmetry, together with the fact that each AGF step consists of gradient flow in one of these functions, implies that the quantity $u_i(t)^2 - v_i(t)^2$ is conserved throughout AGF. Plugging the initialization into this expression gives the first identity $u_i(t)^2 - v_i(t)^2 = 2\alpha$. See Kunin et al. [77] for a general connection between continuous symmetries and conserved quantities in gradient flow. For the second identity, we solve for the intersection of the hyperbola $u_i(t)^2 - v_i(t)^2 = 2\alpha$ and the circle $\|\theta_i(t)\|^2 = u_i(t)^2 + v_i(t)^2$, which gives

$$u_i(t) = \pm\sqrt{\frac{\|\theta_i(t)\|^2 + 2\alpha^2}{2}}, \qquad v_i(t) = \pm\sqrt{\frac{\|\theta_i(t)\|^2 - 2\alpha^2}{2}}. \tag{27}$$

Multiplying these expressions together gives the identity for the product. $\qquad\square$

**Lemma C.3.** *After the $k^{\text{th}}$ iteration of AGF, the normalized utility for the $i^{\text{th}}$ neuron is driven by a Riccati ODE $\frac{d}{dt}\bar{\mathcal{U}}_i = 4\eta_\alpha\left(\left(\bar{\mathcal{U}}_i^*\right)^2 - \bar{\mathcal{U}}_i^2\right)$ with the unique solution,*

$$\bar{\mathcal{U}}_i(\tau^{(k)} + t) = \bar{\mathcal{U}}_i^* \tanh\left(\delta_i^{(k)} + 4\eta_\alpha\bar{\mathcal{U}}_i^* t\right), \quad \text{where } \delta_i^{(k)} = \tanh^{-1}\left(\frac{\bar{\mathcal{U}}_i\left(\tau^{(k)}\right)}{\bar{\mathcal{U}}_i^*}\right). \tag{28}$$

*Proof.* We begin by observing the gradient of the utility function takes the form

$$\nabla_\theta \mathcal{U}_i\left(\theta_i; r^{(k)}\right) = -\nabla_{\beta_i}\mathcal{L}\left(\beta^{(k)}\right)\begin{bmatrix} v_i \\ u_i \end{bmatrix}. \tag{29}$$

Evaluating this quantity at the normalized parameters $\bar{\theta}_i$, and recalling the expression for the maximal utility from Lemma C.1, we obtain,

$$\|\nabla_\theta \mathcal{U}_i(\bar{\theta}_i; r^{(k)})\|^2 = 4\left(\bar{\mathcal{U}}_i^*\right)^2. \tag{30}$$

Substituting this into the normalized utility dynamics derived previously (see Equation (14)), we obtain the Riccati equation. This is a standard ODE with a known solution, where the constant $\delta_i^{(k)}$ is determined by the initial condition $\bar{\mathcal{U}}_i\left(\tau^{(k)}\right)$. $\qquad\square$

**Theorem C.4.** *After the $k^{\text{th}}$ iteration of AGF, the accumulated utility for the $i^{\text{th}}$ neuron is*

$$\mathcal{S}_i\left(\tau^{(k)}+t\right) = \frac{1}{2\eta_\alpha}\log\cosh\left(2\eta_\alpha\left(2\mathcal{U}_i^*t + \frac{\zeta_i}{2\eta_\alpha}\cosh^{-1}\exp\left(2\eta_\alpha\mathcal{S}_i\left(\tau^{(k)}\right)\right)\right)\right), \quad (31)$$

*where $\eta_\alpha = -\log(\sqrt{2}\alpha)$ is the learning rate and $\zeta_i = \text{sgn}\left(-\nabla_{\beta_i}\mathcal{L}\left(\beta^{(k)}\right)\right)\rho_i\left(\tau^{(k)}\right)$. The quantity $\rho_i\left(\tau^{(k)}\right) = \text{sgn}\left(u_i\left(\tau^{(k)}\right)v_i\left(\tau^{(k)}\right)\right)$ is determined by the recursive formula*

$$\rho_i\left(\tau^{(k)}+t\right) = \text{sgn}\left(\frac{\rho_i\left(\tau^{(k)}\right)}{2\eta_\alpha}\cosh^{-1}\exp\left(2\eta_\alpha\mathcal{S}_i\left(\tau^{(k)}\right)\right) - \nabla_{\beta_i}\mathcal{L}\left(\beta^{(k)}\right)t\right), \quad (32)$$

*where $\mathcal{S}_i(0) = 0$ and $\rho_i(0) = 0$.*

*Proof.* Using the expression for the normalized utility derived in Lemma C.3, we can derive an exact expression for the integral of the normalized utility, i.e., the accumulated utility:

$$\mathcal{S}_i\left(\tau^{(k)}+t\right) = \mathcal{S}_i\left(\tau^{(k)}\right) + \int_0^t 2\bar{\mathcal{U}}_i(s)ds = \mathcal{S}_i\left(\tau^{(k)}\right) + \frac{1}{2\eta_\alpha}\log\left(\frac{\cosh\left(4\eta_\alpha\bar{\mathcal{U}}_i^*t + \delta_i^{(k)}\right)}{\cosh\left(\delta_i^{(k)}\right)}\right). \quad (33)$$

Using the hyperbolic identity $\tanh^{-1}(x) = \text{sgn}(x)\cosh^{-1}\left(\frac{1}{\sqrt{1-x^2}}\right)$, we can express the constant $\delta_i^{(k)}$ introduced in Lemma C.3:

$$\delta_i^{(k)} = \text{sgn}\left(-\nabla_{\beta_i}\mathcal{L}\left(\beta^{(k)}\right)\right)\tanh^{-1}\left(\frac{2u_i(\tau^{(k)})v_i(\tau^{(k)})}{\|\theta_i(\tau^{(k)})\|^2}\right) \qquad \text{Lemma C.1} \quad (34)$$

$$= \text{sgn}\left(-\nabla_{\beta_i}\mathcal{L}\left(\beta^{(k)}\right)\right)\rho_i\left(\tau^{(k)}\right)\cosh^{-1}\left(\frac{\|\theta(\tau^{(k)})\|^2}{2\alpha^2}\right) \qquad \text{Lemma C.2} \quad (35)$$

$$= \text{sgn}\left(-\nabla_{\beta_i}\mathcal{L}\left(\beta^{(k)}\right)\right)\rho_i\left(\tau^{(k)}\right)\cosh^{-1}\left(\exp\left(2\eta_\alpha\mathcal{S}_i\left(\tau^{(k)}\right)\right)\right), \quad (36)$$

where in the last equality we used the simplification $\|\theta(\tau^{(k)})\|^2 = 2\alpha^2\exp\left(2\eta_\alpha\mathcal{S}_i\left(\tau^{(k)}\right)\right)$. Substituting this expression into Equation (33), notice that the denominator inside the logarithm simplifies substantially, as the $\mathcal{S}_i(\tau^{(k)})$ term cancels out, yielding Equation (31). Finally, by Lemma C.2, $\rho_i$ changes sign only when $\mathcal{S}_i = 0$, yielding Equation (32). $\square$

**Corollary C.5.** *After the $k^{\text{th}}$ iteration of AGF, the next dormant neuron to activate is $i^* = \arg\min_{i\in\mathcal{D}}\Delta\tau_i$ where*

$$\Delta\tau_i = \frac{\cosh^{-1}\exp\left(2\eta_\alpha\right) - \zeta_i\cosh^{-1}\exp\left(2\eta_\alpha\mathcal{S}_i(\tau^{(k)})\right)}{2\eta_\alpha \cdot 2\bar{\mathcal{U}}_i^*}, \quad (37)$$

*$\zeta_i = \text{sgn}\left(-\nabla_{\beta_i}\mathcal{L}\left(\beta^{(k)}\right)\right)\rho_i\left(\tau^{(k)}\right)$, and the next jump time $\tau^{(k+1)} = \tau^{(k)} + \Delta\tau_{i^*}$.*

*Proof.* From Theorem C.4, we solve for the time $t$ such that $\mathcal{S}_i(\tau^{(k)} + t) = 1$ by inverting the expression for accumulated utility. This time is $\Delta\tau_i$. $\square$

## C.2 Cost minimization

Active neurons represents non-zero regression coefficients of $\beta$. During the cost minimization step, the active neurons work to collectively minimize the loss. When a neuron activates it does so with a certain sign $\text{sgn}(u_iv_i) = -\text{sgn}(\nabla_{\beta_i}\mathcal{L}(\beta_\mathcal{A}))$. If during the cost minimization phase a neuron changes sign, then it can do so only by returning to dormancy first. This is due to the fact that throughout the gradient flow dynamics the quantity $u_i^2 - v_i^2 = 2\alpha^2$ is conserved and thus in order to flip the sign of the product $u_iv_i$, the parameters must return to their initialization. As a result, the critical point reached during the cost minimization step is the unique solution to the constrained optimization problem,

$$\beta_\mathcal{A}^* = \arg\min_{\beta\in\mathbb{R}^d}\mathcal{L}(\beta) \quad \text{subject to} \quad \begin{cases} \beta_i = 0 & \text{if } i \notin \mathcal{A}, \\ \beta_i \cdot \text{sgn}(u_iv_i) \geq 0 & \text{if } i \in \mathcal{A}, \end{cases} \quad (38)$$

where uniqueness follows from the general position assumption. All coordinates where $(\beta_\mathcal{A}^*)_i = 0$ are dormant at the next step of AGF.

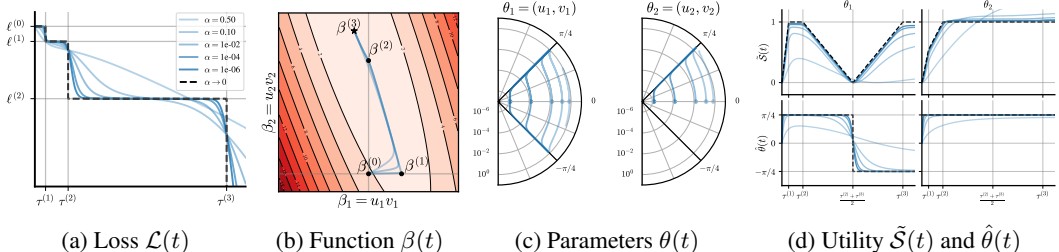

| (a) Loss $\mathcal{L}(t)$ | (b) Function $\beta(t)$ | (c) Parameters $\theta(t)$ | (d) Utility $\tilde{\mathcal{S}}(t)$ and $\hat{\theta}(t)$ |

Figure 8: **AGF = GF as $\alpha \to 0$ in diagonal linear networks.** We consider a diagonal linear network $\beta = u \odot v \in \mathbb{R}^2$, trained by gradient flow from initialization $u = \sqrt{2}\alpha\mathbf{1}$, $v = 0$, with varying $\alpha$, inspired by Figure 2 of Pesme and Flammarion [18]. This example is special as it demonstrates how an active neuron, in this case $\beta^{(1)}$, can return to dormancy during the cost minimization phase, as discussed theoretically in Appendix B. In (a) we plot the loss over accelerated time and in (b) the loss landscape over $\beta$. For small $\alpha$, trajectories evolve from $\beta^{(0)}$ to $\beta^{(3)}$, passing near intermediate saddle points $\beta^{(1)}$ and $\beta^{(2)}$. Each saddle point is associated with a plateau in the loss. As $\alpha \to 0$, gradient flow spends all its time at these saddles, jumping instantaneously between them at $\tau^{(1)}, \tau^{(2)}, \tau^{(3)}$, matching the stepwise loss drops. In (c) we plot trajectories in parameter space and in (d) the same dynamics visualized in terms of their accumulated utility and angle. The jump times between critical points correspond to when the accumulated utility satisfies $\tilde{\mathcal{S}}_i(\tau) = 1$. When the first coordinate returns to dormancy, the accumulated utility first touches zero, causing the angle to flip sign, before reactivating with the opposite sign.

## C.3 AGF in action

Combining the results of utility maximization (Appendix C.1) and cost minimization (Appendix C.2) yields an explicit recursive expression for the sequence generated by AGF. For each neuron we only need to track the accumulated utility $\mathcal{S}_i(t)$ and the sign $\rho_i(t)$, which at initialization are both zero. We can now consider the relationship between the sequence produced by AGF in the vanishing initialization limit $\alpha \to 0$ and the sequence for gradient flow in the same initialization limit introduced by Pesme and Flammarion [18].

In Pesme and Flammarion [18], they prove that in the vanishing initialization limit $\alpha \to 0$, the trajectory of gradient flow, under accelerated time $\tilde{t}_\alpha(t) = -\log(\alpha) \cdot t$, converges towards a piecewise constant limiting process corresponding to the sequence of saddles produced by Algorithm 2. This algorithm can be split into two steps, which match exactly with the two steps of AGF in the vanishing initialization limit.

---

**Algorithm 2:** Pesme and Flammarion [18]

**Initialize:** $t \leftarrow 0, \beta \leftarrow 0 \in \mathbb{R}^d, \mathcal{S} \leftarrow 0 \in \mathbb{R}^d$;
**while** $\nabla\mathcal{L}(\beta) \neq 0$ **do**

$\quad \mathcal{D} \leftarrow \{j \in [d] \mid \nabla\mathcal{L}(\beta)_j \neq 0\}$
$\quad \tau^* \leftarrow \inf\{\tau_i > 0 \mid \exists i \in \mathcal{D}, \mathcal{S}_i - \tau_i\nabla\mathcal{L}(\beta)_i = \pm 1\}$
$\quad t \leftarrow t + \tau^*, \quad \mathcal{S} \leftarrow \mathcal{S} - \tau^*\nabla\mathcal{L}(\beta)$

$\quad \beta = \arg\min \mathcal{L}(\beta)$ where $\beta \in \left\{ \beta \in \mathbb{R}^d \,\middle|\, \begin{matrix} \beta_i \geq 0 & \text{if } \mathcal{S}_i = +1, \\ \beta_i \leq 0 & \text{if } \mathcal{S}_i = -1, \\ \beta_i = 0 & \text{if } \mathcal{S}_i \in (-1,1) \end{matrix} \right\}$

**return** Sequence of $(\beta, t)$;

---

We adapt the original notation to fit our framework, highlighting utility maximization (blue) and cost minimization (red) steps.

As in AGF, at each iteration of their algorithm a coefficient that was zero becomes non-zero. The new coefficient comes from the set $\{i \in [d] : \nabla\mathcal{L}(\beta)_i \neq 0\}$, which is equivalent to the set of dormant neurons with non-zero utility, as all active neurons will be in equilibrium from the previous cost minimization step. The index for the new active coefficient is determined by the following expression,

$$i_* = \arg\min\{\tau_i > 0 : \exists i \in \mathcal{D}, s_i - \tau_i\nabla\mathcal{L}(\beta)_i = \pm 1\}. \tag{39}$$

Because $\tau_i > 0$ and $s_i \in (-1, 1)$ for all coefficients $i \in \mathcal{D}$, then this expression only makes sense if we choose the boundary associated with $\text{sgn}(\nabla\mathcal{L}(\beta)_i)$. Taking this into account, we get the following simplified expression for the jump times proposed by Pesme and Flammarion [18],

$$\tau_i = \frac{1 + \text{sgn}(\nabla\mathcal{L}(\beta)_i)s_i}{|\nabla\mathcal{L}(\beta)_i|}. \tag{40}$$

This expression is equivalent to the expression given in Corollary C.5 in the limit $\eta_\alpha \to \infty$. To see this, we use the asymptotic identity $\cosh^{-1}(\exp(x)) \sim x + \log 2$ as $x \to \infty$. Combined with the

simplification $2\bar{\mathcal{U}}_i^* = |\nabla_{\beta_i}\mathcal{L}(\beta)|$, this allows us to simplify the expression for the jump time and show that $\rho_i \mathcal{S}_i = s_i$ where $s_i$ is the integral from Algorithm 2.

The second step of Algorithm 2, is that the new $\beta$ is determined by the following constrained minimization problem,

$$\beta^* = \arg\min_{\beta \in \mathbb{R}^d} \mathcal{L}(\beta) \quad \text{subject to} \quad \begin{cases} \beta_i \geq 0, & \text{if } s_i = 1 \\ \beta_i \leq 0, & \text{if } s_i = -1 \\ \beta_i = 0, & \text{if } s_i \in (-1, 1) \end{cases} \tag{41}$$

Using the correspondence $\rho_i = \operatorname{sgn}(s_i)$, and noting that neurons with $s_i \in (-1, 1)$ correspond to the dormant set, this constrained optimization problem is equivalent to the cost minimization step of AGF, presented in Appendix C.2.

# D  Complete Proofs for Fully Connected Linear Networks

In this section, we provide the details behind the results around linear networks in Section 4.

We begin by clarifying the notion of 'neuron' in this context. As briefly explained in Section 4, due to the symmetry of the parametrization of fully-connected linear networks, the usual notion of a neuron does not define a canonical decomposition of $f$ into rank-1 maps. Instead, the singular value decomposition of the linear map $AW$ computed by the network, defines such a canonical decomposition, being the unique orthogonal one. Therefore, we will think of neurons as basis vectors $(\tilde{a}_i, \tilde{w}_i) \in \mathbb{R}^{c+d}$ under orthogonality constraints. These vectors align with the singular vectors and are partitioned into dormant and active sets, $\mathcal{D}$ and $\mathcal{A}$, based on whether their corresponding singular value is $\mathcal{O}(1)$.

## D.1  Utility maximization

Differently from the usual setting of AGF, the orthogonality constraint on the dormant basis vectors implies that the utility function is not decoupled. Instead, it is maximized over the space of $|\mathcal{D}|$ orthonormal basis vectors, i.e., the Stiefel manifold.

**Lemma D.1.** *After the $k^{\text{th}}$ iteration of AGF, the utility function of the $i^{\text{th}}$ dormant basis vector with parameters $\theta_i = (\tilde{a}_i, \tilde{w}_i)$ is:*

$$\mathcal{U}_i\left(\theta_i; r^{(k)}\right) = -\tilde{a}_i^{\intercal} \nabla_\beta \mathcal{L}\left(\beta_{\mathcal{A}}^{(k)}\right) \tilde{w}_i. \tag{42}$$

*The total utility $\sum_{i \in \mathcal{D}} \mathcal{U}_i\left(\theta_i; r^{(k)}\right)$ is maximized on the Stiefel manifold when $\{\tilde{a}_i, \tilde{w}_i\}_{i=1}^{|\mathcal{D}|}$ coincides with the set of the top $|\mathcal{D}| = H - k$ left and right singular vectors of $\nabla_\beta \mathcal{L}(\beta_{\mathcal{A}}^{(k)})$, resulting in a maximal utility value for the $i^{\text{th}}$ dormant basis vector of $\bar{\mathcal{U}}_i^* = \sigma_i^{(k)}/2$, where $\sigma_i^{(k)}$ is the corresponding singular value. Moreover, $\mathcal{U}_i$ has no other local maxima.*

*Proof.* After substituting $r^{(k)} = y - \beta_{\mathcal{A}}^{(k)} x$, the computation of the utility is straightforward. Everything else follows from the standard theory of Reyleigh quotients over Stiefel manifolds [123]. □

This means that the gradient flow of the utility function aligns the dormant basis vectors with the singular vectors of $-\nabla_\beta \mathcal{L}(\beta_{\mathcal{A}}^{(k)})$.

## D.2  Cost minimization

As discussed in Section 4, the cost minimization phase is governed by the well-known Eckart-Young theory of reduced-rank regression [78]. According to the latter, during cost minimization, the active basis vectors converge to

$$\beta_{\mathcal{A}}^{(k)} = \mathbf{P}_{U_k} \Sigma_{yx} \Sigma_{xx}^{-1}, \tag{43}$$

where $\mathbf{P}_{U_k}$ is the projection onto the top $k$ eigenvectors of $\Sigma_{yx} \Sigma_{xx}^{-1} \Sigma_{yx}^{\intercal}$. Plugging this solution into the expression for the gradient, we find that the utility at the next iteration will be computed with the projection $\mathbf{P}_{U_k}^{\perp} \Sigma_{yx}$:

$$-\nabla_\beta \mathcal{L}(\beta_{\mathcal{A}}^{(k)}) = \Sigma_{yx} - \beta_{\mathcal{A}}^{(k)} \Sigma_{xx} = (\mathbf{I}_c - \mathbf{P}_{U_k})\Sigma_{yx}. \tag{44}$$

## D.3  AGF in action

Putting together the utility maximization and cost minimization steps, AGF progressively learns the projected OLS solution (Equation (43)), coinciding with the GLRL algorithm by Li et al. [20], shown in Algorithm 3, with notation from the original work. At each stage, the GLRL algorithm selects a top eigenvector (blue: utility maximization) and performs gradient descent in the span of selected directions (red: cost minimization).

---

**Algorithm 3:** Greedy Low-Rank Learning Li et al. [20]

**Initialize:** $r \leftarrow 0$, $W_0 \leftarrow 0 \in \mathbb{R}^{d \times d}$, $U_0(\infty) \in \mathbb{R}^{d \times 0}$
**while** $\lambda_1(-\nabla\mathcal{L}(W_r)) > 0$ **do**

$r \leftarrow r + 1$;
$u_r \leftarrow$ unit top eigenvector of $-\nabla\mathcal{L}(W_{r-1})$;
$U_r(0) \leftarrow [U_{r-1}(\infty) \quad \sqrt{\varepsilon}u_r] \in \mathbb{R}^{d \times r}$;

**for** $t = 0, 1, \ldots, T$ **do**
$\quad \lfloor \; U_r(t+1) \leftarrow U_r(t) - \eta\nabla\mathcal{L}(U_r(t))$;
$W_r \leftarrow U_r(\infty)U_r^{\top}(\infty)$

**return** Sequence of $W_r$

---

We now discuss jump times, and motivate Conjecture 4.1. Recall from Section B.4 that in this setting, because $\kappa = 2$, in the limit of vanishing initialization, the alignment in the utility maximization phase occurs instantaneously. As a consequence, the individual utility functions remain, essentially, constantly equal to their corresponding maximal value throughout each utility maximization phase. From the definition of accumulated utility with $\eta_\alpha = -\log(\alpha)$, we immediately deduce the recursive relation:

$$\mathcal{S}_i \left( \tau^{(k)} + t \right) = \mathcal{S}_i \left( \tau^{(k-1)} \right) + \sigma_i^{(k)} t. \tag{45}$$

However, since $-\nabla_\beta \mathcal{L}(\beta_{\mathcal{A}}^{(k)}) = \mathbf{P}_{U_k}^\perp \Sigma_{yx}$ and $-\nabla_\beta \mathcal{L}(\beta_{\mathcal{A}}^{(k-1)}) = \mathbf{P}_{U_{k-1}}^\perp \Sigma_{yx}$ have different singular values and vectors, the relation between the values of Equation (45) as $i \in \mathcal{D}$ varies is unclear, and it is hard to determine the first dormant basis vector to accumulate a utility value of $1$. Yet, suppose that the dormant basis vectors align in such a way that the ordering of the corresponding singular values is preserved across the utility maximization phases. In this case, the ordering of the cumulated utilities in Equation (45) is also preserved for all iterations $k$ and all times $t$. In particular, it follows by induction that the basis vector with index $i_*$ corresponding to the largest eigenvalue $\sigma_{i_*}^{(k)}$ jumps after a time of:

$$\Delta\tau^{(k)} = \frac{1}{\sigma_{i_*}^{(k)}} \left( 1 - \mathcal{S}_{i_*} \left( \tau^{(k-1)} \right) \right). \tag{46}$$

Once unrolled, the above recursion is equivalent to the statement on jump times in Conjecture 4.1.

When $\Sigma_{xx}$ commutes with $\Sigma_{yx}^\intercal \Sigma_{yx}$, the left singular vectors of $\Sigma_{yx}$ simultaneously diagonalize $\Sigma_{xx}$ and $\Sigma_{yx} \Sigma_{xx}^{-1} \Sigma_{yx}^\intercal$. In this case, the singular values of $\mathbf{P}_{U_k}^\perp \Sigma_{yx}$ correspond to the bottom ones of $\Sigma_{yx}$, with the same associated singular vectors. Therefore, the dormant basis vectors are correctly aligned already from the first iteration of AGF. By the discussion above, this confirms Conjecture 4.1 in this specific scenario. Moreover, the recursive expression from Equation (46) reduces to $\Delta\tau^{(k)} = 1/\sigma_k - 1/\sigma_{k-1}$, where $\sigma_k$ is the $k^{\text{th}}$ largest singular value of $\Sigma_{yx}$. Therefore, the jump times are simply:

$$\tau^{(k)} = \frac{1}{\sigma_k}. \tag{47}$$

These values, together with the loss values in Conjecture 4.1, coincide with the ones derived by Gidel et al. [19] for GF at vanishing initialization. This means that under the commutativity assumption, AGF and GF converge to the same limit, similarly to the setting of diagonal linear networks.

When $\Sigma_{xx}$ and $\Sigma_{yx}^\intercal \Sigma_{yx}$ do not commute, in order to prove Conjecture 4.1, it is necessary to understand the relation between the SVD of $\mathbf{P}_{U_k}^\perp \Sigma_{yx}$ and of $\Sigma_{yx}$, as $k$ varies. The Poincaré separation theorem describes this relation for the singular values, stating that they are interlaced, i.e., they alternate between each other when ordered. This is not sufficient, since the geometry of the singular vectors plays a crucial role in the alignment phase. Even though there exist classical results from linear algebra in this direction [124], we believe that Conjecture 4.1 remains open.

# E  Complete Proofs for Attention-only Linear Transformer

Here, we provide additional details and derivations for the results around transformers (see Section 4 for the problem setup and notation). This analysis is based on the setting presented by Zhang et al. [32], to which we refer the reader for further details.

Note that this scenario does not exactly fit the formalism of Section 2, since we are considering a neural architecture different from a fully-connected network. Yet, due to our assumption on the rank of the attention heads, each head behaves as a bottleneck, resembling a 'cubical version' of a neuron. Therefore, we will interpret attention heads as 'neurons', and apply AGF to this context (with $\kappa = 3$).

We will show in the following sections that at the end of the $k^{\text{th}}$ iteration of AGF, the $i^{\text{th}}$ attention head, $h = 1, \ldots, k$, learns the function:

$$f_h(X; \theta_h^*)_{D+1,N+1} = \frac{1}{\text{tr}\Sigma_{xx} + (N+1)\lambda_h} \sum_{n=1}^{N} y_n x_n^{\mathsf{T}} v_h v_h^{\mathsf{T}} x_{N+1}, \tag{48}$$

where $\lambda_h$ is the $h^{\text{th}}$ largest eigenvalue of $\Sigma_{xx}$, and $v_h$ is the corresponding eigenvector. We will proceed by induction on $k$.

## E.1  Utility maximization

We prove the following lemma, establishing, inductively, the utility function for the $k^{\text{th}}$ iteration of AGF.

**Lemma E.1.** *At the $k^{\text{th}}$ iteration of AGF, assume that the $(k-1)$ active attention heads have the function form as in Equation* (48). *Then, the utility function of a dormant head with parameters $\theta_i = (V_i, Q_i, K_i)$ is given by:*

$$\mathcal{U}_i\left(\theta_i; r^{(k)}\right) = N V_i\, Q_i^{\mathsf{T}} \left(\Sigma_{xx}^2 - \sum_{h=1}^{k-1} \lambda_h^2 v_h v_h^{\mathsf{T}}\right) K_i \tag{49}$$

*Moreover, the utility is maximized over normalized parameters by $(\bar{V}_i^*, \bar{Q}_i^*, \bar{K}_i^*) = (\pm 1/\sqrt{3}, \pm v_k/\sqrt{3}, \pm v_k/\sqrt{3})$, where the sign is determined such that $\bar{\mathcal{U}}_i^* = N\lambda_k^2/(3\sqrt{3})$. Moreover, $\mathcal{U}_i$ has no other local maxima.*

*Proof.* By assumption, we have:

$$\mathcal{U}_i\left(\theta_i; r^{(k)}\right) = \mathbb{E}_{x_1,\ldots,x_{N+1},\beta}\left[\left(y_{N+1} - \sum_{h=1}^{k-1} f_h(X; \theta_h^*)_{D+1,N+1}\right) f_i(X, \theta_i)_{D+1,N+1}\right]. \tag{50}$$

The first term, which coincides with the initial utility, can be computed as:

$$\mathcal{U}_i\left(\theta_i; y\right) = \mathbb{E}_{x_1,\ldots,x_{N+1},\beta}\left[y_{N+1} f_i(X, \theta_i)_{D+1,N+1}\right]$$

$$= \mathbb{E}_{x_1,\ldots,x_{N+1},\beta}\left[\beta^{\mathsf{T}} x_{N+1} V_i \sum_{n=1}^{N} y_n x_n^{\mathsf{T}} K_i Q_i^{\mathsf{T}} x_{N+1}\right]$$

$$= V_i\, Q_i^{\mathsf{T}} \mathbb{E}_{x_{N+1}}[x_{N+1} x_{N+1}^{\mathsf{T}}]\mathbb{E}_{\beta}[\beta\beta^{\mathsf{T}}] \sum_{n=1}^{N} \mathbb{E}_{x_n}[x_n x_n^{\mathsf{T}}] K_i$$

$$= N V_i\, Q_i^{\mathsf{T}} \Sigma_{xx}^2 K_i. \tag{51}$$

Denote $A_h = (\text{tr}\Sigma_{xx} + (N+1)\lambda_k)^{-1}$. Then, the summand can be computed as:

$$\mathbb{E}_{x_1,\ldots,x_{N+1},\beta}\left[f_h^*(X, \theta_h)_{D+1,N+1} f_i(X, \theta_i)_{D+1,N+1}\right]$$

$$= A_h V_i \mathbb{E}_{x_1,\ldots,x_{N+1},\beta}\text{tr}\left(\beta\beta^{\mathsf{T}} \sum_{n=1}^{N} x_n x_n^{\mathsf{T}} K_i Q_i^{\mathsf{T}} x_{N+1} x_{N+1}^{\mathsf{T}} v_h v_h^{\mathsf{T}} \sum_{n=1}^{N} x_n x_n^{\mathsf{T}}\right)$$

$$= A_h V_i \text{tr}\left(K_i Q_i^{\mathsf{T}} \Sigma_{xx} v_h v_h^{\mathsf{T}} \mathbb{E}_{x_1,\ldots,x_n}\left(\sum_{n=1}^{N} x_n x_n^{\mathsf{T}}\right)^2\right) \tag{52}$$

$$= A_h V_i \text{tr}\left(K_i Q_i^{\mathsf{T}} \lambda_h v_h v_h^{\mathsf{T}} \left(N\Sigma_{xx}\text{tr}\Sigma_{xx} + N(N+1)\Sigma_{xx}^2\right)\right)$$

$$= N V_i Q_i^{\mathsf{T}} \lambda_h^2 v_h v_h^{\mathsf{T}} K_i,$$

where in the second equality, we have used the fact that

$$
\mathbb{E}_{x_1,\ldots,x_N}\left[\left(\sum_{n=1}^{N}x_n x_n^\mathsf{T}\right)^2\right] = \mathbb{E}_{x_1,\ldots,x_N}\left[\sum_{n=1}^{N}(x_n x_n^\mathsf{T})^2 + 2\sum_{1\le i<j\le N}x_i x_i^\mathsf{T} x_j x_j^\mathsf{T}\right] \tag{53}
$$
$$
= N\Sigma_{xx}\mathrm{tr}\Sigma_{xx} + N(N+1)\Sigma_{xx}^2.
$$

This provides the desired expression for the utility, which corresponds to a Rayleigh quotient. Since the largest eigenvalue of $(\Sigma_{xx}^2 - \sum_{h=1}^{k-1}\lambda_h^2 v_h v_h^\mathsf{T})$ is $\lambda_k^2$, the rest of the statement follows from the standard theory of Rayleigh quotients. $\qquad\square$

### E.2  Cost minimization

After utility maximization, a new head aligns to the corresponding eigenvector and becomes active. This implies that at the beginning of the cost minimization phase of the $k^{\text{th}}$ iteration of AGF, the $k$ active attention heads have parameters $\theta_h = (V_h, Q_h, K_h)$, such that $Q_h$ and $K_h$ coincide up to a multiplicative scalar to $v_h$ for all $1 \le h \le k$. Moreover, Equation (48) holds for $1 \le h < k$.

We wish to show that during the cost minimization phase, the newly-activated head with parameters $\theta_k$ learns a function in the same form. First, we prove that the loss function is decoupled for each active attention head.

**Lemma E.2.** *Assume that the $k$ active attention heads have the same function form, up to multiplicative scalar, as in Equation* (48)*, Then, the loss over active heads decoupled into sum of individual losses, that is*

$$
\mathcal{L}(\Theta_\mathcal{A}) = \sum_{i=1}^{k}\mathcal{L}(\theta_i). \tag{54}
$$

*Moreover, for all $h = 1,\ldots,k$, the directional derivatives $\frac{\partial\mathcal{L}}{\partial Q_h}\mathcal{L}(\Theta_\mathcal{A})$ and $\frac{\partial\mathcal{L}}{\partial K_h}\mathcal{L}(\Theta_\mathcal{A})$ are proportional to $v_h$ throughout the cost minimization phase.*

*Proof.* The first statement follows from a straightforward calculation using the orthogonality of eigenvectors. As a result,

$$
\frac{\partial\mathcal{L}}{\partial Q_h}(\Theta_\mathcal{A}) = \frac{\partial\mathcal{L}}{\partial Q_h}(\theta_h) = \mathbb{E}_{x_1,\ldots,x_N,\beta}\left(V_h\sum_{n=1}^{N}y_n x_n^\mathsf{T}K_h\right)^2 \Sigma_{xx}Q_h, \tag{55}
$$

From this expression, we see that if $Q_h$ starts as an eigenvector of $\Sigma_{xx}$, then subsequent gradient updates will be in the same direction. As a result, the direction of $Q_h$ remains unchanged in cost minimization. A similar argument holds for $K_h$ as well. $\qquad\square$

Since the active heads follow the GF of $\mathcal{L}$, the decoupling of the losses implies that during the cost minimization phase, each head actually follows the gradient of the loss of its own parameters, disregarding the other heads. In particular, $\theta_h$ remains at equilibrium for $h = 1,\ldots,k-1$, since its loss has been optimized during the previous iterations of AGF. By Lemmas E.1 and E.2, the newly-activated head remains aligned to its eigenvector $v_k$, learning a function of the form

$$
f_k(X;\theta_k)_{D+1,N+1} = A_k\sum_{n=1}^{N}y_n x_n^\mathsf{T}v_{k+1}v_{k+1}^\mathsf{T}x_{N+1}, \tag{56}
$$

where $A_k \in \mathbb{R}$ is a parameter that is optimized in cost minimization. $\mathcal{L}(\theta_k)$ is convex with respect to $A_k$, and the corresponding minimization problem is easily seen to be solved by $A_k = 1/(\mathrm{tr}\Sigma_{xx} + (N+1)\lambda_k)$. This concludes our inductive argument.

### E.3 AGF in action

As we have seen above, AGF describes a recursive procedure, where each head sequentially learns a principal component. We can easily compute the loss value at the end of each iteration:

$$
\begin{aligned}
\ell^{(k)} &\equiv \mathbb{E}_{x_1,\ldots,x_{N+1},\beta} \frac{1}{2} \left( y_{N+1} - \sum_{h=1}^{k} A_h \sum_{n=1}^{N} y_n x_n^\mathsf{T} v_h v_h^\mathsf{T} x_{N+1} \right)^2 \\
&= \frac{1}{2} \left( \sum_{i=1}^{D} \lambda_i - \sum_{h=1}^{k} \frac{N \lambda_h^2}{\mathrm{tr}\Sigma_{xx} + (N+1)\lambda_h} \right).
\end{aligned}
\tag{57}
$$

Furthermore, as an immediate consequence of Lemma E.1, we can lower bound the jump times. The accumulated utility at the $k^{\text{th}}$ iteration of AGF is upper-bounded by $\mathcal{S}_i(t) \leq 3t\bar{\mathcal{U}}_i^* = tN\lambda_k^2/\sqrt{3}$. The jump time is given by the first head that reaches $\mathcal{S}_i(\tau^{(k)} - \tau^{(l)}) = 1/\|\theta_i(\tau^{(l)})\|$, which yields

$$
\tau^{(k)} - \tau^{(l)} \geq \frac{\sqrt{3}}{N\lambda_k^2 \mu_l},
\tag{58}
$$

where $\mu_k = \max_i \|\theta_i(\tau^{(l)})\|$ and $\tau^{(0)} = 0$.

# F  Complete Proofs for Generalized Modular Addition

In this section, we provide the proofs for the results summarized in Section 5. Instead of reasoning inductively as in Section E, for simplicity of explanation we will derive in detail the steps of the first iteration of AGF. As we will explain in Section F.3, all the derivations can be straightforwardly extended to the successive iterations, completing the inductive argument.

## F.1  Utility maximization

In order to compute the utility for a single neuron, we will heavily rely on the Discrete Fourier Transform (DFT) and its properties. To this end, recall that for $u \in \mathbb{R}^p$, its DFT $\hat{u} \in \mathbb{R}^p$ is defined as:

$$\hat{u}[k] = \sum_{a=0}^{p-1} u[a] e^{-2\pi i k a / p}, \tag{59}$$

where $i = \sqrt{-1}$ is the imaginary unit. We start by proving a technical result.

**Lemma F.1.** *Let $x, u, v, w \in \mathbb{R}^p$. Then:*

$$\sum_{a,b=0}^{p-1} \langle u, a \cdot x \rangle \langle v, b \cdot x \rangle \langle w, (a+b) \cdot x \rangle = \frac{1}{p} \sum_{k=0}^{p-1} |\hat{x}[k]|^2 \, \hat{u}[k] \hat{v}[k] \overline{\hat{w}[k] \hat{x}[k]}. \tag{60}$$

*Proof.* Notice that the inner product $\langle u, a \cdot x \rangle$ can be expressed as

$$\langle u, a \cdot x \rangle = \sum_{k=0}^{p} x[k] u[k+a] = (x \star u)[a], \tag{61}$$

where $x \star u$ is the cross-correlation between $x$ and $u$. Thus, the left-hand side of Equation (60) can be rewritten as:

$$\sum_{a,b=0}^{p-1} (x \star u)[a] \, (x \star v)[b] \, (x \star w)[a+b] = \langle x \star u, x \star v \star x \star w \rangle$$

By using Plancharel's theorem $\langle x, y \rangle = \frac{1}{p} \langle \hat{x}, \hat{y} \rangle$ and the cross-correlation property $\widehat{x \star y}[k] = \overline{\hat{x}[k]} \hat{y}[k]$, the above expression equals

$$\frac{1}{p} \sum_{k=0}^{p-1} \hat{x}[k] \hat{u}[k] \hat{v}[k] \overline{\hat{w}[k] \hat{x}[k]^2}, \tag{62}$$

which corresponds to the desired result via the simplification $\overline{\hat{x}[k]} \hat{x}[k] = |\hat{x}[k]|^2$. $\square$

**Lemma F.2.** *Under the modular addition task with embedding vector $x$ and mean centered labels, the initial utility function for a single neuron parameterized by $\theta = (u, v, w)$ can be expressed as*

$$\mathcal{U}(\theta; y) = \frac{2}{p^3} \sum_{k=1}^{p-1} |\hat{x}[k]|^2 \, \hat{u}[k] \hat{v}[k] \overline{\hat{w}[k] \hat{x}[k]}. \tag{63}$$

*Proof.* By definition, the utility is:

$$\mathcal{U}(\theta; y) = \frac{1}{p^2} \sum_{a,b=0}^{p-1} \left\langle \left( \langle u, a \cdot x \rangle + \langle v, b \cdot x \rangle \right)^2 w, (a+b) \cdot x - \frac{\langle x, \mathbf{1} \rangle}{p} \mathbf{1} \right\rangle$$

$$= \frac{1}{p^2} \sum_{a,b=0}^{p-1} \left( \langle u, a \cdot x \rangle^2 + \langle v, b \cdot x \rangle^2 + 2\langle u, a \cdot x \rangle \langle v, b \cdot x \rangle \right) \left( \langle w, (a+b) \cdot x \rangle - \frac{\langle w, \mathbf{1} \rangle \langle x, \mathbf{1} \rangle}{p} \right). \tag{64}$$

Due to the cyclic structure of $\langle w, (a+b) \cdot x \rangle$, the contributions from the terms $\langle u, a \cdot x \rangle^2$ and $\langle v, b \cdot x \rangle^2$ terms vanish. This reduces the utility to

$$\frac{2}{p^2} \sum_{a,b=0}^{p-1} \langle u, a \cdot x \rangle \langle v, b \cdot x \rangle \langle w, (a+b) \cdot x \rangle - \frac{2}{p^3} \left( \sum_{a=0}^{p-1} \langle u, a \cdot x \rangle \right) \left( \sum_{b=0}^{p-1} \langle v, b \cdot x \rangle \right) \left( \sum_{c=0}^{p-1} \langle w, c \cdot x \rangle \right).$$

(65)

The first summand in the above expression is provided by Lemma F.1. Moreover, since the mean of a vector corresponds to the zero-frequency component $k = 0$ of its DFT, the second summand reduces to:

$$\frac{2}{p^3} \left( \sum_{a=0}^{p-1} \langle u, a \cdot x \rangle \right) \left( \sum_{b=0}^{p-1} \langle v, b \cdot x \rangle \right) \left( \sum_{c=0}^{p-1} \langle w, c \cdot x \rangle \right) = \frac{2}{p^3} \widehat{x \star u}[0] \; \widehat{x \star v}[0] \; \widehat{x \star w}[0]$$

$$= \frac{2}{p^3} |\hat{x}[0]|^3 \hat{u}[0]\hat{v}[0]\hat{w}[0].$$

(66)

Since the latter coincides with the term $k = 0$ in Equation (61), we obtain the desired expression. $\qquad\square$

We now solve the constrained optimization problem involved in the utility maximization step of our framework.

**Theorem F.3.** *Let $\xi$ be a frequency that maximizes $|\hat{x}[k]|$, $k = 1, \ldots, p-1$, and denote by $s_x$ the phase of $\hat{x}[\xi]$. Then the unit vectors $\theta_* = (u_*, v_*, w_*)$ that maximize the initial utility function $\mathcal{U}(\theta; y)$ take the form*

$$u_*[a] = \sqrt{\frac{2}{3p}} \cos\left( 2\pi \frac{\xi}{p} a + s_u \right)$$

$$v_*[b] = \sqrt{\frac{2}{3p}} \cos\left( 2\pi \frac{\xi}{p} b + s_v \right)$$

(67)

$$w_*[c] = \sqrt{\frac{2}{3p}} \cos\left( 2\pi \frac{\xi}{p} c + s_w \right),$$

*where $a, b, c \in \{0, \ldots, p-1\}$ are indices and $s_u, s_v, s_w \in \mathbb{R}$ are phase shifts satisfying $s_u + s_v \equiv s_w + s_x \pmod{2\pi}$. They achieve a maximal value of $\mathcal{U}^* = \sqrt{2/(27p^3)}|\hat{x}[\xi]|^3$. Moreover, the utility function has no local maxima other than the ones described above.*

*Proof.* By Plancharel's theorem, the constraint $\|\theta\|^2 = \|u\|^2 + \|v\|^2 + \|w\|^2 = 1$ is equivalent to a constraint on the squared norm of its DFT, up to a scaling. Together with Lemma F.2, this allows us to reformulate the original optimization problem in the frequency domain as:

$$\text{maximize} \quad \frac{2}{p^3} \sum_{k=1}^{p-1} |\hat{x}[k]|^2 \, \hat{u}[k]\hat{v}[k]\overline{\hat{w}[k]\hat{x}[k]}$$

(68)

$$\text{subject to} \quad \|\hat{u}\|^2 + \|\hat{v}\|^2 + \|\hat{w}\|^2 = p.$$

To solve this optimization problem, consider the magnitudes $\mu_x, \mu_u, \mu_v, \mu_w \in \mathbb{R}_+^p$ and phases $S_x, S_u, S_v, S_w \in [0, 2\pi)^p$ of the DFT coefficient of $x, u, v, w$, respectively. This means that $\hat{x}[k] = \mu_x[k] \exp(\mathrm{i}S_x[k])$ for every $k$, and similarly for $u, v, w$.

Since $u$, $v$, and $w$ are real-valued, their DFTs satisfy conjugate symmetry (i.e., $\hat{u}[-k] = \overline{\hat{u}[k]}$). Since $p$ is odd, the periodicity of the DFT (i.e., $\hat{u}[k] = \hat{u}[k + p]$) allows us to simplify the objective by only considering the first half of the frequencies. Substituting the magnitude and phase expressions we obtain the equivalent optimization,

$$\text{maximize} \quad \frac{4}{p^3} \sum_{k=1}^{\frac{p-1}{2}} \mu_x[k]^3 \mu_u[k]\mu_v[k]\mu_w[k] \cos(S_u[k] + S_v[k] - S_w[k] - S_x[k])$$

(69)

$$\text{subject to} \quad \sum_{k=0}^{\frac{p-1}{2}} \mu_u[k]^2 + \sum_{k=0}^{\frac{p-1}{2}} \mu_v[a]^2 + \sum_{k=0}^{\frac{p-1}{2}} \mu_w[a]^2 = \frac{p}{2}.$$

The phase terms can be chosen independent of the constraints to maximize the cosine term. Since the only local maximum of the cosine is $0 \pmod{2\pi}$, the only local optimum of that term is achieved by setting $S_u[k] + S_v[k] - S_w[k] - S_x[k] \equiv 0 \pmod{2\pi}$. But then the optimization problem reduces to maximizing $\sum_{k \in [(p-1)/2]} \mu_x[k]^3 \mu_u[k] \mu_v[k] \mu_w[k]$, subject to the constraints. The only local maximum of this problem is easily seen to be achieved by concentrating all the magnitude at the dominant frequency $\xi$ of $x$ (and its conjugate), i.e.:

$$\mu_u[k] = \mu_v[k] = \mu_w[k] = \begin{cases} \sqrt{\frac{p}{6}} & \text{if } k = \pm\xi \pmod{p} \\ 0 & \text{otherwise.} \end{cases} \tag{70}$$

This gives a maximal objective value of $\bar{\mathcal{U}}^* = (4/p^3)\mu_x[\xi]^3(p/6)^{3/2} = \sqrt{2/(27p^3)}|\hat{x}[\xi]|^3$. By applying the inverse DFT with these choices for magnitudes and phases, while accounting for conjugate symmetry, yields the desired result. Note that in the statement, $s_x, s_u, s_v, s_w$ denote $S_x[\xi], S_u[\xi], S_v[\xi], S_w[\xi]$, respectively.

$\square$

We highlight that Theorem F.3 and its proof is very similar to Theorem 7 and its proof in Morwani et al. [96]. They derived this optimization problem by looking for a maximum margin solution, suggesting there may be an interesting connection between utility maximization and maximum margin biases of gradient descent.

### F.2 Cost minimization

We now discuss cost minimization. As demonstrated in the previous section, during the utility maximization step, the parameters of each neuron align with the harmonic of the dominant frequency $\xi$ of $x$, albeit with phase shifts $s_u^i, s_v^i, s_w^i$. The goal of this section is two-fold. First, in Appendix F.2.1 we discuss our assumption that during the cost minimization phase, the neurons remain aligned with the same harmonic, possibly varying the amplitudes and phase shifts of their parameters. Second, in Appendix F.2.2 we solve the cost minimization problem over such aligned neurons. Therefore, throughout the section, we consider $N$ neurons parametrized by $\Theta = (u^i, v^i, w^i)_{i=1}^N$, where:

$$u^i[a] = A_u^i \sqrt{\frac{2}{3p}} \cos\left(2\pi\frac{\xi}{p}a + s_u^i\right),$$

$$v^i[b] = A_v^i \sqrt{\frac{2}{3p}} \cos\left(2\pi\frac{\xi}{p}b + s_v^i\right), \tag{71}$$

$$w^i[c] = A_w^i \sqrt{\frac{2}{3p}} \cos\left(2\pi\frac{\xi}{p}c + s_w^i\right),$$

for some amplitudes $A_u^i, A_v^i, A_w^i \in \mathbb{R}_{\geq 0}$ and some phase shifts $s_u^i, s_v^i, s_w^i \in \mathbb{R}$.

To begin with, note that loss splits as:

$$\mathcal{L}(\Theta) = \frac{1}{2p^2} \sum_{a,b=0}^{p-1} \left\| \sum_{i=1}^N f_i(a \cdot x, b \cdot x; \theta_i) - (a+b) \cdot x + \frac{\langle x, \mathbf{1} \rangle}{p}\mathbf{1} \right\|^2$$

$$= \mathcal{C}(\Theta) - \mathcal{U}(\Theta; y) + \frac{1}{2}\left( \|x\|^2 - \frac{\langle x, \mathbf{1} \rangle^2}{p} \right), \tag{72}$$

where

$$\mathcal{C}(\Theta) = \frac{1}{2p^2} \sum_{i,j=1}^N \langle w^i, w^j \rangle \sum_{a,b=0}^{p-1} \left( \langle u^i, a \cdot x \rangle + \langle v^i, b \cdot x \rangle \right)^2 \left( \langle u^j, a \cdot x \rangle + \langle v^j, b \cdot x \rangle \right)^2 \tag{73}$$

and $\mathcal{U}(\Theta; y) = \sum_{i=1}^N \mathcal{U}(\theta_i; y)$ is the cumulated utility function of the $N$ neurons. From Lemma F.2, we know that:

$$\mathcal{U}(\theta_i; y) = \sqrt{\frac{2}{27p^3}}|\hat{x}[\xi]|^3 A_u^i A_v^i A_w^i \cos(s_u^i + s_v^i - s_w^i - s_x). \tag{74}$$

Throughout this section, we repeatedly use the following identity: for any $p \in \mathbb{N}$, $k \in \mathbb{Z}$, and $s \in \mathbb{R}$,

$$\sum_{a=0}^{p-1} \cos\left(s - \tfrac{2\pi ka}{p}\right) = \begin{cases} p\cos(s), & \text{if } k = 0 \pmod{p}, \\ 0, & \text{otherwise.} \end{cases} \tag{75}$$

This follows by writing $\cos(\theta) = \mathrm{Re}(e^{i\theta})$ and noting that the geometric sum $\sum_{a=0}^{p-1} e^{-2\pi i ka/p}$ vanishes unless $k = 0 \pmod{p}$.

### F.2.1 Preservation of harmonic alignment

To analyze cost minimization, we restrict attention to a regime in which the newly activated neurons do not change their aligned harmonic during this phase.

**Assumption F.4.** During cost minimization, the $N$ newly activated neurons remain aligned to the harmonic $\xi$.

This restriction allows us to solve cost minimization within the subspace spanned by these $N$ neurons, in close analogy with prior sections. However, first we characterize the conditions under which this assumption is valid and clarify in what sense it provides a faithful description of the dynamics.

**Theorem F.5.** *Let $h \in \mathbb{R}^p$ be a vector in the form:*

$$h[a] = A\cos\left(2\pi\frac{\xi'}{p}a + s\right), \tag{76}$$

*for some $A, s \in \mathbb{R}$ and $\xi' \neq 0, \pm\xi$. If for all $a, b \in [p]$*

$$\sum_{i=1}^{N} w^i \langle u^i, a \cdot x\rangle^2 = \sum_{i=1}^{N} w^i \langle v^i, b \cdot x\rangle^2 = 0, \tag{77}$$

*then for all $i = 1, \ldots, N$:*

$$\left\langle h, \frac{\partial \mathcal{L}}{\partial u^i}(\Theta)\right\rangle = \left\langle h, \frac{\partial \mathcal{L}}{\partial v^i}(\Theta)\right\rangle = \left\langle h, \frac{\partial \mathcal{L}}{\partial w^i}(\Theta)\right\rangle = 0. \tag{78}$$

*Proof.* A direct calculation leads to the following expressions for the derivatives:

$$\left\langle h, \frac{\partial \mathcal{L}}{\partial u^i}(\Theta)\right\rangle = 4\sum_{a,b=0}^{p-1} \langle h, a \cdot x\rangle \left(\langle u^i, a \cdot x\rangle + \langle v^i, b \cdot x\rangle\right) \langle w^i,\ f(a \cdot x, b \cdot x; \Theta) - (a+b)\cdot x\rangle,$$

$$\left\langle h, \frac{\partial \mathcal{L}}{\partial v^i}(\Theta)\right\rangle = 4\sum_{a,b=0}^{p-1} \langle h, b \cdot x\rangle \left(\langle u^i, a \cdot x\rangle + \langle v^i, b \cdot x\rangle\right) \langle w^i,\ f(a \cdot x, b \cdot x; \Theta) - (a+b)\cdot x\rangle,$$

$$\left\langle h, \frac{\partial \mathcal{L}}{\partial w^i}(\Theta)\right\rangle = 2\sum_{a,b=0}^{p-1} \left(\langle u^i, a \cdot x\rangle + \langle v^i, b \cdot x\rangle\right)^2 \langle h,\ f(a \cdot x, b \cdot x; \Theta) - (a+b)\cdot x\rangle.$$

$$\tag{79}$$

Via a tedious goniometric computation, the three expressions above can be expanded into a sum of cosine terms. Since $\xi \neq \xi'$, each of these terms depends on $a$ or $b$, and therefore averages out. We do not report the details of the computation here, but refer to the proof of Lemma F.6 for more details. $\qquad\square$

The condition Equation (77) rules out square terms that could otherwise generate gradient components at *resonant frequencies* $\xi' \neq \pm\xi$. If the squared activations $\langle u^i, a \cdot x\rangle^2$ or $\langle v^i, b \cdot x\rangle^2$ have a nonzero aggregate contribution, their trigonometric expansion produces cosine terms at doubled or mixed frequencies that may survive the averaging over $a$ or $b$ via the cosine-sum identity Equation (75). In this case, the gradient can acquire components outside the span of the aligned harmonic, allowing neurons to escape alignment.

However, pairs of aligned neurons can easily satisfy Equation (77) by coordinating their amplitudes and phases. For example, using the identity $z_1 z_2 = \frac{1}{4}(z_1 + z_2)^2 - \frac{1}{4}(z_1 - z_2)^2$, neurons can arrange for square terms to cancel in aggregate while preserving their linear contributions. Such cancellations correspond to directional adjustments that can occur rapidly during cost minimization.

### F.2.2 Cost minimization over aligned neurons

Next, in order to solve the cost minimization problem over aligned neurons, we explicitly compute the term $\mathcal{C}(\Theta)$ in the loss function.

**Lemma F.6.** *We have:*

$$\mathcal{C}(\Theta) = \frac{|\hat{x}[\xi]|^4}{54p^2} \sum_{i,j=1}^{N} A_w^i A_w^j \cos(s_w^i - s_w^j) \Big( \big((A_u^i)^2 + (A_v^i)^2\big)\big((A_u^j)^2 + (A_v^j)^2\big)$$
$$+ \frac{(A_u^i A_u^j)^2}{2} \cos(2(s_u^i - s_u^j)) + \frac{(A_v^i A_v^j)^2}{2} \cos(2(s_v^i - s_v^j)) \tag{80}$$
$$+ 2A_u^i A_v^i A_u^j A_v^j \big(\cos(s_u^i + s_v^i - s_u^j - s_v^j) + \cos(s_u^i - s_v^i - s_u^j + s_v^j)\big) \Big).$$

*Proof.* This follows from a rather tedious computation, which we summarize here. First, note that

$$\langle w^i, w^j \rangle = \frac{2 A_w^i A_w^j}{3p} \sum_{c=0}^{p-1} \cos\left(s_w^i + 2\pi\frac{\xi}{p}c\right) \cos\left(s_w^j + 2\pi\frac{\xi}{p}c\right) = \frac{A_w^i A_w^j}{3} \cos(s_w^i - s_w^j). \tag{81}$$

Next, the shift-equivariance property of the Fourier transform and Plancharel's theorem imply that for all $i$ and $a$ we have:

$$\langle u^i, a \cdot x \rangle = \frac{1}{p}\langle \widehat{u^i}, \widehat{a \cdot x} \rangle = \sqrt{\frac{2}{3p}}|\hat{x}[\xi]| A_u^i \cos\left(s_x - s_u^i - 2\pi\xi\frac{a}{p}\right), \tag{82}$$

and similarly for $v^i$. We can plug the above expression into the quadratic term $\big(\langle u^i, a \cdot x \rangle + \langle v^i, b \cdot x \rangle\big)^2$ from Equation (73) and expand it via the goniometric identity:

$$(A\cos(\alpha) + B\cos(\beta))^2 = \frac{1}{2}(A^2 + B^2 + A^2\cos(2\alpha) + B^2\cos(2\beta)) + AB(\cos(\alpha+\beta) + \cos(\alpha-\beta)). \tag{83}$$

We similarly expand the term $\big(\langle u^j, a \cdot x \rangle + \langle v^j, b \cdot x \rangle\big)^2$. By leveraging on the goniometric identity $2\cos(\alpha)\cos(\beta) = \cos(\alpha - \beta) + \cos(\alpha + \beta)$, the product of these quadratic terms for the $i^{th}$ and $j^{th}$ neurons expands into 16 unique terms. Since $p$ is odd, $2\xi, 4\xi \neq 0 \pmod{p}$, and thus, by Equation (75), only the terms independent from both $a$ and $b$ do not average out. This leaves four terms, providing the desired result. □

We now provide a lower bound for the (meaningful terms of the) loss function.

**Theorem F.7.** *We have the following lower bound:*

$$\mathcal{C}(\Theta) - \mathcal{U}(\Theta; y) \geq -\frac{|\hat{x}[\xi]|^2}{p}. \tag{84}$$

*Moreover, equality holds if, and only if, we have that $\sum_{i=1}^{N} C_i \cos(\alpha_i) = \sum_{i=1}^{N} C_i \sin(\alpha_i) = 0$ for any choice of $(C_i, \alpha_i)$ among*

$$(A_w^i((A_u^i)^2 + (A_v^i)^2), s_w^i),$$
$$(A_w^i(A_u^i)^2, s_w^i \pm 2s_u^i),$$
$$(A_w^i(A_v^i)^2, s_w^i \pm 2s_v^i), \tag{85}$$
$$(A_w^i A_u^i A_v^i, s_w^i \pm (s_u^i - s_v^i)),$$
$$(A_w^i A_u^i A_v^i, s_w^i + s_u^i + s_v^i),$$

*and, moreover, $\sum_{i=1}^{N} A_w^i A_u^i A_v^i \sin(s_w^i + s_x - s_u^i - s_v^i) = 0$ and $\sum_{i=1}^{N} A_w^i A_u^i A_v^i \cos(s_w^i + s_x - s_u^i - s_v^i) = \sqrt{54p}/|\hat{x}[\xi]|$.*

*Proof.* Given amplitudes $C_1, \ldots, C_N$ and angles $\alpha_1, \ldots, \alpha_N$, consider the goniometric identity

$$\sum_{i,j=1}^{N} C_i C_j \cos(\alpha_i - \alpha_j) = \left(\sum_{i=1}^{N} C_i \cos(\alpha_i)\right)^2 + \left(\sum_{i=1}^{N} C_i \sin(\alpha_i)\right)^2. \tag{86}$$

By expanding each product of two cosines in Equation (80) into a sum of two cosines, we can apply the above identity to each summand by choosing $\alpha_i \in \{s_w^i, s_w^i \pm 2s_u^i, s_w^i \pm 2s_v^i, s_w^i \pm (s_u^i - s_v^i), s_w^i + s_u^i + s_v^i, s_w^i + s_x - s_u^i - s_v^i\}$. Since both the summands in the right-hand side of Equation (86) are positive, we conclude that:

$$\mathcal{C}(\Theta) \geq \frac{|\hat{x}[\xi]|^4}{54p^2} \left( \sum_{i=1}^{N} A_w^i A_u^i A_v^i \cos(s_w^i + s_x - s_u^i - s_v^i) \right)^2, \tag{87}$$

with equality holding only if Equation (85) is satisfied. Therefore, using Equation (74), we deduce:

$$
\begin{aligned}
\mathcal{C}(\Theta) - \mathcal{U}(\Theta; y) &\geq \frac{|\hat{x}[\xi]|^4}{54p^2} \left( \sum_{i=1}^{N} A_w^i A_u^i A_v^i \cos(s_w^i + s_x - s_u^i - s_v^i) \right)^2 \\
&\quad - \sqrt{\frac{2}{27p^3}} \, |\hat{x}[\xi]|^3 \sum_{i=1}^{N} A_w^i A_u^i A_v^i \cos(s_w^i + s_x - s_u^i - s_v^i) \\
&= \left( \frac{|\hat{x}[\xi]|^2}{\sqrt{54p}} \sum_{i=1}^{N} A_w^i A_u^i A_v^i \cos(s_w^i + s_x - s_u^i - s_v^i) - \frac{|\hat{x}[\xi]|}{\sqrt{p}} \right)^2 - \frac{|\hat{x}[\xi]|^2}{p} \\
&\geq -\frac{|\hat{x}[\xi]|^2}{p},
\end{aligned}
\tag{88}
$$

with equality only when $(|\hat{x}[\xi]|^2/(\sqrt{54}p)) \sum_{i=1}^{N} A_w^i A_u^i A_v^i \cos(s_w^i + s_x - s_u^i - s_v^i) = |\hat{x}[\xi]|/\sqrt{p}$, which concludes the proof. $\qquad\square$

For $N \geq 6$, the lower bound from Equation (84) is tight. For even $N$, minimizers can be constructed by setting $s_w^i + s_x = s_u^i + s_v^i = \frac{2\pi i}{N}$, $s_u^i - s_v^i \in \{0, \pi\}$ alternating (depending on $N$), and $A_w^i A_u^i A_v^i = \sqrt{54p}/(N|\hat{x}[\xi]|)$. For $N$ odd, a similar construction holds.

By combining Theorem F.7 and Equation (72), we deduce the following tight lower bound on the loss function:

$$\mathcal{L}(\Theta) \geq \frac{1}{2} \left( \|x\|^2 - \frac{\langle x, \mathbf{1} \rangle^2}{p} - \frac{2|\hat{x}[\xi]|^2}{p} \right). \tag{89}$$

### F.3 Utility update

Next, we consider the utility for a new neuron after some group of $N \geq 6$ neurons has undergone the previous steps AGF. Suppose that $\Theta_* = (u_*^i, v_*^i, w_*^i)_{i=1}^{N}$ are parameters minimizing the loss as in Section F.2. We start by describing the function computed by the network with parameters $\Theta_*$.

**Lemma F.8.** *For all $0 \leq a, b < p$, the network satisfies:*

$$f(a \cdot x, b \cdot x; \Theta_*) = \frac{2|\hat{x}[\xi]|}{p} (a + b) \cdot \chi_\xi, \tag{90}$$

*where $\chi_\xi$ is defined as $\chi_\xi[c] = \cos\left(2\pi \frac{\xi}{p} c + s_x\right)$.*

*Proof.* The $N$ neurons compute the function

$$f(a \cdot x, b \cdot x; \Theta_*)[c] = \sum_{i=1}^{N} \left( \langle u_*^i, a \cdot x \rangle + \langle v_*^i, b \cdot x \rangle \right)^2 w_*^i[c]. \tag{91}$$

The equation above can be expanded via a computation analogous to the one in the proof of Lemma F.6. The conditions on minimizers from Theorem F.7 imply that the only non-vanishing term is of the

form:

$$\sqrt{\frac{2}{27p^3}}|\hat{x}[\xi]|^2 \sum_{i=1}^{N} A_w^i A_u^i A_v^i \cos\left(2s_x + s_w^i - s_u^i - s_v^i + 2\pi\frac{\xi}{p}(c - a - b)\right)$$

$$=\sqrt{\frac{2}{27p^3}}\cos\left(s_x + 2\pi\frac{\xi}{p}(c - a - b)\right)\underbrace{\sum_{i=1}^{N} A_w^i A_u^i A_v^i \cos(s_w^i + s_x - s_u^i - s_v^i)}_{\sqrt{54p}/|\hat{x}[\xi]|} \tag{92}$$

$$=\frac{2|\hat{x}[\xi]|}{p}\cos\left(s_x + 2\pi\frac{\xi}{p}(c - a - b)\right),$$

where in the first identity we used the fact that $\sum_{i=1}^{N} A_w^i A_u^i A_v^i \sin(s_w^i + s_x - s_u^i - s_v^i) = 0$. This concludes the proof. $\square$

Finally, we compute the utility function with the updated residual $r(a \cdot x, b \cdot x) = (a + b) \cdot x - (1/p)\langle x, \mathbf{1}\rangle\mathbf{1} - f(a \cdot x, b \cdot x; \Theta_*)$, which is maximized by new neurons in the second iteration of AGF.

**Theorem F.9.** *The utility function with the updated residual for a neuron with parameters $\theta = (u, v, w)$ is:*

$$\mathcal{U}(\theta; r) = \frac{2}{p^3} \sum_{k \in [p]\setminus\{0, \pm\xi\}} |\hat{x}[k]|^2 \hat{u}[\xi]\hat{v}[\xi]\overline{\hat{w}[\xi]\hat{x}[\xi]}. \tag{93}$$

*Proof.* The new utility function is

$$\mathcal{U}(\theta; r) = \mathcal{U}(\theta; y) - \frac{1}{p^2} \underbrace{\sum_{a,b=0}^{p-1} \left\langle (\langle u, a \cdot x\rangle + \langle v, b \cdot x\rangle)^2 w, f(a \cdot x, b \cdot x; \Theta_*)\right\rangle}_{\Delta(\theta)} \tag{94}$$

We wish to express $\Delta$ in the frequency domain. By plugging in Equation (90), and due to the cyclic structure of $\chi_\xi$, the squared terms average out:

$$\Delta(\theta) = \sum_{a,b=0}^{p-1} \left(\langle u, a \cdot x\rangle^2 + 2\langle u, a \cdot x\rangle\langle v, b \cdot x\rangle + \langle v, b \cdot x\rangle^2\right)\langle w, f(a \cdot x, b \cdot x; \Theta_*)\rangle \tag{95}$$

$$= 2\sum_{a,b=0}^{p-1} \langle u, a \cdot x\rangle\langle v, b \cdot x\rangle\langle w, f(a \cdot x, b \cdot x; \Theta_*)\rangle. \tag{96}$$

Now, by reasoning analogously to the proof of Lemma F.1, we obtain:

$$\Delta(\theta) = \frac{4|\hat{x}[\xi]|}{p} \sum_{a,b=0}^{p-1} (x \star u)[a](x \star y)[b](\chi_\xi \star w)[a + b]$$

$$= \frac{4|\hat{x}[\xi]|}{p^2} \sum_{k=0}^{p-1} \widehat{(x \star u)}[k]\,\widehat{(x \star v)}[k]\,\overline{\widehat{(\chi_\xi \star w)}[k]} \tag{97}$$

$$= \frac{2|\hat{x}[\xi]|}{p}\left(\hat{u}[\xi]\hat{v}[\xi]\overline{\hat{w}[\xi]\hat{x}[\xi]} + \overline{\hat{u}[\xi]\hat{v}[\xi]}\hat{w}[\xi]\hat{x}[\xi]\right),$$

where in the last equality we used the fact that $\widehat{\chi_\xi}[k] = \frac{p}{2}e^{\pm is_x}$ if $k = \pm\xi$ and $0$ otherwise. By subtracting the above expression to the expression for the initial utility from Lemma F.2, we obtain the desired result. $\square$

In summary, after an iteration of AGF, the utility for a new neuron has the same form as the initial one, but with the summand corresponding to the dominant (conjugate) frequencies $\pm\xi$ of $x$ removed. Consequently, by the same reasoning as in Section F.1, we conclude that during the utility

maximization phase of the second iteration of AGF, some new group of dormant neurons aligns with the harmonic whose frequency has the second largest magnitude in $\hat{x}$.

A subtlety arises during the cost minimization phase of the second iteration, as the neurons that aligned during the first phase are still involved in the optimization process. However, note that in the loss component $\mathcal{C}(\Theta)$ (Equation (73)), the terms of the form $\langle w_i, w_j \rangle$ vanish when $i$ and $j$ are indices of neurons aligned with harmonics of different frequencies. Therefore, during the second cost minimization phase, the loss $\mathcal{L}(\Theta)$ splits into the sum of two losses, corresponding to the neurons aligned during the first and second iteration of AGF, respectively. The neurons from the first phase are already at a critical point of their respective loss term, thus the second group of neurons is optimized independently, via the same arguments as in Section F.1. This scenario is analogous to the one discussed in Section E.2 for linear transformers (cf. Lemma E.2). In conclusion, after the second iteration of AGF, the new utility will again have the same form as the initial one, but with the two (conjugate pairs of) frequencies removed. This argument iterates recursively, until either all the frequencies are exhausted, or all the $H$ neurons have become active. The quantities in Equation (9) can be derived for the successive iterations of AGF analogously to the previous sections. Lastly, the estimate on jump times is obtained by the same argument as in Section E.3.

