# OpenReview forum: "Alternating Gradient Flows: A Theory of Feature Learning in Two-layer Neural Networks"
_NeurIPS.cc/2025/Conference — NeurIPS 2025 poster_

### Official Review · Reviewer_vhww · 2025-06-30

**Clarity:** 3
**Significance:** 3
**Originality:** 3
**Rating:** 5
**Confidence:** 4

**Summary:**

This paper introduces the concept of alternating gradient flows to explain saddle-to-saddle or stepwise dynamics during training, providing a unified framework across four distinct settings. The central insight is that, despite differing representations (e.g., neurons, SVD modes, Fourier modes), the dynamics exhibit a common mechanism: an initial utility maximization phase over dormant neurons, followed by a cost minimization phase over active neurons.

**Questions:**

There are no major technical questions. The methodology and arguments are well-presented. The main concerns lie in the assumptions and scope of the setup, as noted in the weaknesses.

**Ethical Concerns:**

["NO or VERY MINOR ethics concerns only"]

**Final Justification:**

The rebuttal has adequately addressed my concerns. Considering the paper's novelty and insight, I am raising my overall evaluation to a 5.

**Limitations:**

Yes

**Quality:**

3

**Strengths And Weaknesses:**

**Strengths**

The proposed unification offers novel insights into the training dynamics of two-layer neural networks with small initialization. The paper is clearly written, and the key ideas are well-articulated and supported by analysis.


**Weaknesses**

1. Small initialization leads to slow dynamics in the early stages of training and may result in neuron alignment along only a few directions. In such cases, the network effectively utilizes only a small subset of neurons. This behavior may diverge from typical training regimes used in practice, raising questions about the framework's applicability to real-world scenarios and data.

2. Extending the proposed framework to deeper architectures is non-trivial, potentially limiting its generalizability and broader impact.

---

> ### Author Rebuttal · Authors · 2025-07-30
>
> Thank you for your review. We will address the two weaknesses you brought up individually below.
>
> **Studying small initializations.** The study of learning dynamics from small initialization—and its connection to sparsity and simplicity biases—has a rich history in the theory of deep learning literature. As we discuss and cite (e.g., Woodworth et al., 2020; Li et al., 2020; Jacot et al., 2021), this regime has been instrumental in developing theoretical insights into feature learning. You are correct that saddle-to-saddle learning dynamics of neural networks from small initialization differs from the maximal update parameterization used in most large-scale models, where weights move $\mathcal{O}(1)$ in time. We explicitly acknowledge this distinction in lines 72-75. However, recent work (e.g., Atanasov et al., 2025) has shown that while the scale of updates differs, the qualitative behavior of feature learning—such as alignment and progressive recruitment of neurons—remains similar across regimes. We study small initialization precisely because the slower dynamics and separation of timescales make the underlying feature learning process more analyzable. This structure is what enables the development of AGF, which decomposes training into a sequence of simpler optimization problems. In this sense, the very property that makes small initialization less practical for realistic training is what makes it valuable for theory. In our updated manuscript, we will expand Section 6 to include a more explicit discussion of the limitations of the small initialization regime, clarifying how it serves as a tractable proxy for studying feature learning in more realistic settings.
>
> **Extending AGF to deeper networks.** We agree, as we state on line 386, that the main limitation of AGF is its restriction to two-layer networks. We believe that an extension to deeper networks is possible. In our revised manuscript, we have added a longer discussion in Section 6 on how future work could build off our framework to extend to deeper networks. One promising direction is to extend our numerical implementation of AGF for two-layer networks (discussed in our response to Reviewer D2un) to ResNet-style architectures by treating each residual block as a two-layer network and the encoder and decoder layers as another two-layer network. Other possibilities would be to consider generalizing the definition of a “neuron” to an input-output path as in the path-framework proposed in Saxe et al., 2022 analysis of deep gated ReLU networks.
>
> We understand that your main concern is in the scope of our work's applicability to real-world scenarios and data. We would like to emphasize that the analysis of feature learning from small-scale initialization and in shallow neural networks is an active area of research, with several papers published in the space every year. While we agree that our work will not directly impact researchers training SOTA models, we believe that it can have a significant impact on this community and our understanding of feature learning in neural networks from first principles. We hope you will consider raising your score to reflect this.

---

> > ### Comment · Reviewer_vhww · 2025-08-01
> >
> > I appreciate the authors’ responses to my questions. I agree that training SOTA models should not be the evaluation metric for theoretical papers. My concern lies more in the gap between theoretical analysis of simplified models and practical feature learning in realistic settings, as well as the potential for such theoretical insights to extend to the latter.
> > While there is indeed a substantial body of work on two-layer networks, it is often unclear to what extent these results capture the dynamics in practical feature learning region. That said, the paper presents a unifying perspective on different types of saddle-to-saddle dynamics, which I find insightful. The authors also cite supporting work suggesting that small initialization can serve as a meaningful proxy for studying feature learning in more realistic settings, and that such insights may extend to deeper architectures.
> >
> > If the authors are willing to elaborate further on these two points, I would be inclined to raise my evaluation. In particular, I was unable to locate the referenced paper by Atanasov et al. (2025), and a more detailed discussion on how the analysis could extend to deeper networks would be appreciated.

---

> > > ### Author Response · Authors · 2025-08-04
> > >
> > > Thank you for the thoughtful follow-up and for clarifying your concerns. Below, we expand on two points you raised: (1) how our results relate to practical feature-learning dynamics, and (2) how the analysis can extend to deeper networks.
> > >
> > >
> > > ### (1) “to what extent these results capture the dynamics in practical feature learning region”
> > > As you noted, small initialization can lead to dynamics where only a small subset of neurons are active—an effect that might appear to limit the practical relevance of our analysis. However, numerous empirical and theoretical works suggest that this behavior is not just an artifact of toy models, but may in fact reflect broader principles of feature learning in realistic settings. We expand here on a few empirical works we cited on lines 34–36:
> > > - *A Closer Look at Memorization in Deep Networks* (Arpit et al., 2017) was among the first to highlight that deep networks, despite having the capacity to memorize arbitrary labels, tend to learn simpler patterns first when trained on natural data. This "simplicity bias" helps explain generalization in overparameterized models and is consistent with the stepwise, low-rank learning we observe in AGF.
> > > - *SGD on Neural Networks Learns Functions of Increasing Complexity* (Kalimeris et al., 2019) provides direct empirical evidence of a simplicity bias in SGD-trained networks. Their experiments demonstrate that networks trained on real and synthetic data learn progressively more complex functions, supporting the notion of gradual feature acquisition.
> > > - *Hidden Progress in Deep Learning: SGD Learns Parities Near the Computational Limit* (Barak et al., 2022) shows that feature learning can proceed in a discontinuous manner, with sudden jumps in performance once certain features emerge—an observation qualitatively aligned with the saddle-to-saddle transitions modeled in AGF.
> > > - *The Optimization Landscape of SGD Across the Feature Learning Strength* (Atanasov et al., 2024) is the work we meant to cite in our rebuttal (apologies for the incorrect year). This paper explores how varying initialization scale interpolates between kernel regimes and strong feature learning. Crucially, it finds that alignment and recruitment behaviors persist even when the initialization is not vanishingly small, suggesting that small-initialization analyses can capture qualitatively relevant phenomena in broader settings.
> > > We will revise Section 6 of our paper to include a more explicit discussion of these connections. Our aim is to clarify that while our framework focuses on small initialization for analytical tractability, the mechanisms it captures—such as progressive recruitment and alignment—have been observed empirically in more realistic settings as well.
> > >
> > > ### (2) “a more detailed discussion on how the analysis could extend to deeper networks”
> > > We emphasize that extending our analysis to deep networks is non-trivial, and we do not claim to have resolved this challenge in the present work. Rather, our aim is to provide a clear and general framework in the two-layer setting that can serve as a foundation for future developments. Below, we outline several directions we find promising for extending AGF to deeper architectures. While none are guaranteed to succeed, we believe they offer potential avenues for exploration:
> > > - **Layer-wise AGF dynamics:** A first idea is to extend AGF to deeper networks in a layer-wise fashion. Specifically, this would consist in applying AGF to all consecutive two-layer blocks, simultaneously; when a block is considered, all the other ones are treated as “frozen”. In other words, the utility function of a hidden neuron would depend on the parameters of the current active neurons in the other layers, and would be obtained by Taylor-expand the loss w.r.t. the weights adjacent to that neuron. This would capture interleaved feature learning at different depths. Moreover, such an extension is especially suitable for ResNet-style architectures, where each residual block consists, by design, of two layers.
> > > - **Path-based generalizations:** Another, perhaps more refined, direction is to replace the notion of a neuron in AGF with a path. Specifically, a “neuron” would correspond to an input-output path through a deep network. Similarly to the two-layer version of AGF, each such path would accumulate utility, and jump from dormant to active. This perspective is inspired by Saxe et al. (2022), where a similar path-based perspective is exploited for an analysis of deep gated ReLU networks. We believe that this direction might be more mathematically sound than the previous one, but probably involves further subtleties, due to interactions between intersecting paths.
> > >
> > >
> > > We sincerely appreciate your thoughtful engagement with our paper. We hope this addresses your concerns and that you will consider raising your evaluation.

---

> > > > ### Comment · Reviewer_vhww · 2025-08-07
> > > >
> > > > Thank you for the response. I believe it has adequately addressed my concerns, and I will raise my evaluation accordingly.

---

### Official Review · Reviewer_qgue · 2025-07-01

**Clarity:** 3
**Significance:** 2
**Originality:** 2
**Rating:** 4
**Confidence:** 2

**Summary:**

This work introduces an *Alternating Gradient Flow* (AGF) algorithm for approximating the gradient flow dynamics of two-layer layer neural networks. Inspired by the saddle-to-saddle dynamics observed accross different ML tasks, AGF mimicks by an iterative, two-step procedure: (1) Utility maximization (plateau): at a saddle-point, a subset of *dormant* neurons (with small weights) evolve independently to maximize their correlation with the rest, *active* neurons; (2) Cost minimization (drop): Once a dormant neuron develops meaningful correlation, it becomes active, leading to a drop of loss until they reach a new plateau.

THe paper then investigates AGF in different settings. First, they show that AGF is exact when the initialization scale $\alpha\to 0^{+}$ for diagonal neural networks. Second, they show that in linear two-layer networks AGF recovers the step-wise singular value recovery previously observed in the context of commuting covariate-covariate and label-covariate covariances, and conjecture that this remains approximately true in the non-commuting case. Third, they show that AGF implements stepwise principal component regression for linear transformers. Finally, they show that two-layer quadratic networks trained on AGF learn Fourier features a modular addition task.

**Questions:**

- The notation is confusing. In lines 64-65, the 2LNN is written as $f(x;\Theta) = \sum_{i=1}^{H}a_{i}\sigma(w_{i}^{\top}g_{i}(x))$ with $a_{i}\in\mathbb{R}^{c}$. Do you actually mean $c=1$ or this is understood as a vector $\times$ scalar multiplication? In L72 you write $a_{i}\sim\mathcal{N}(0,\alpha/\sqrt{2c})$ suggesting $a_{i}$ is a scalar. Or should I understand this as i.i.d. component-wise?

- How do you define a *smooth dynamical system* in L77-78? Is it clear that GF is a smooth dynamical system for any activation function $\sigma$?

- L43-45: "*However, these analyses often rely on specific simplifying assumptions about the architecture (e.g., linear activations), data (e.g., orthogonal inputs), and optimizer (e.g., layer-wise learning)*". This is a strange way of contrasting the paper to the cited literature, since all cases analyzed here are for simplified architectures (2 layer, linear or quadratic) and a simplified optimizer (GF, AGF), and from the results it is not really clear to what extent they generalize beyond it. Moreover, most of the cited papers only use one of these simplifying assumptions. I suggest tuning this down.

- What is the role of the parametrization of the quadratic network in L337-338 in the dynamics? For instance, given the data $z=(a\dot x,b\dot x)\in\mathbb{R}^{2d}$, one could fit a standard quadratic network $f_{i}(x;\Theta) = \langle \beta_{i}, z\rangle^{2}w_{i}$ with double number of neurons $\beta_{i}\in\mathbb{R}^{2d}$, how does the AGF dynamics look like? Does it spontaneously learn to split the neurons?

- The population dynamics of non-linear 2 layer neural networks has been widely studied, specially in the context of multi-index models (Saad, Solla 1995; Goldt et al., 2018; Arnaboldi et al., 2023a). A picture arising from these works is that the saddle-to-saddle dynamics in these models correspond to a *specialization transition* (Saad, Solla 1995) from a regime where the correlation with the target directions is homogeneous (all neurons attempt to fit the linear part of the target independently) to a regime where neurons specialize to directions of the target (see for instance fig. 1 of Goldt et al., 2018 or fig. 2 of Arnaboldi et al., 2023a). In particular, the exact details of the specialized regime depends on the target architecture. For instance, if the target is itself a 2LNN and the model has a number of neurons which is a multiple of the target the dynamics can prioritize groups of neurons specializing at some target weights, while if the model width is not a multiple of the target, this can induce neurons to "shut down" instead of grouping (Goldt et al., 2018). Moreover, (Arnaboldi et al., 2023a) has proven that in this context the model neurons are only approximately independent in the large-width (mean-field) limit, with the correlation between neurons being non-negligeable for a finite width network. Note that all these works preceed [36], which confirm and prove part of this analysis under some additional assumptions. Together, this seems to paint a quite different picture from AGF, where individual neurons specialize sequentially and independently, not collectively. How do the authors see these works in the light of AGF?

- On a note similar to the above, the SGD dynamics of quadratic neural networks have been widely studied as a simple non-linear setting in the literature (Sarao Mannelli et al., 2020; Arnaboldi et al., 2023b; Martin et al., 2024). The picture that emerge from these works is that every target direction is learned simultaneously, although at different speeds (depending on the corresponding eigenvalue). This seems to lack the separation of time implied by the AGF where directions are fully learned sequentially. Am I missing something?

**References**:

- (Saad, Solla 1995) David Saad, Sara A. Solla. "*Dynamics of On-Line Gradient Descent Learning for Multilayer Neural Networks*". NeurIPS 1995

- (Ben Arous et al., 2021) Gerard Ben Arous, Reza Gheissari, Aukosh Jagannath. "*Online stochastic gradient descent on non-convex losses from high-dimensional inference*". JMLR 2021

- (Goldt et al., 2018) Sebastian Goldt, Madhu S. Advani, Andrew M. Saxe, Florent Krzakala, Lenka Zdeborová. "*Dynamics of stochastic gradient descent for two-layer neural networks in the teacher-student setup*". NeurIPS 2019.

- (Arnaboldi et al., 2023a) Luca Arnaboldi, Ludovic Stephan, Florent Krzakala, Bruno Loureiro. "*From high-dimensional & mean-field dynamics to dimensionless ODEs: A unifying approach to SGD in two-layers networks*". COLT 2023.

- (Sarao Mannelli et al., 2020) Stefano Sarao Mannelli, Eric Vanden-Eijnden, Lenka Zdeborová. "*Optimization and Generalization of Shallow Neural Networks with Quadratic Activation Functions*". NeurIPS 2020

- (Arnaboldi et al., 2023b) Luca Arnaboldi, Florent Krzakala, Bruno Loureiro, Ludovic Stephan. "*Escaping mediocrity: how two-layer networks learn hard generalized linear models with SGD*". arXiv:2305.18502

- (Martin et al., 2024) Simon Martin, Francis Bach, Giulio Biroli. "*On the Impact of Overparameterization on the Training of a Shallow Neural Network in High Dimensions*". ICML 2024

**Ethical Concerns:**

["NO or VERY MINOR ethics concerns only"]

**Final Justification:**

The concerns I have raised have been properly addressed by the authors during the rebuttal period.

I remain on the opinion that the paper has a tendency to oversell the centrality of AGF for understanding feature learning, specially given the evidence provided, which is limited to retrieving known phenomenology from a new perspective. Nevertheless, I think the promised changes will help to better place the paper within other approaches to study feature learning on 2LNN, as well as to make some of the claims more sober.

Considering this, as well as a careful reading of the other reviews, I am happy to reconsider my score towards a weak accept.

**Limitations:**

Limitations are discussed in the last section.

**Paper Formatting Concerns:**

N/A.

**Quality:**

3

**Strengths And Weaknesses:**

**Strengths**: The paper is well-written and the idea of unifying the dynamics of different settings studied in the literature under a single framework is appealing.

**Weaknesses**: The principal weakness of the paper is that the scope and relevance of AGF beyond the simple cases analyzed remain unclear. Equivalence with GF is proved or conjectured only in the linear setting, and the single non-linear example - a quadratic network - relies on an additional assumption that, while reasonable, falls outside the stated scope of AGF.

On a bigger picture, it is not clear to what extent the "broad picture" offered here is useful or insightful. Because gradient flow is a strictly descent method, once the critical points are characterized, its trajectory is fully determined by the initial conditions: it proceeds from initialization toward a minimum, potentially passing through successive saddles. Hence, the substantive challenge lies in calculating these critical points (and their indices) for concrete problems of interest. In this light, the principal contribution I see in this work is the identification of a utility function that potentially simplifies part of this analysis - estimating the time scale for escaping saddles — under certain conditions whose generality is unclear (see questions below). The framework does not, however, appear to simplify other essential aspects of the dynamics, such as determining the eventual point of convergence or whether that point is a local or global minimum for a given initialization.

On a minor note, the paper also ignores a large body of literature that has been dedicated to similar questions on quadratic nets and single-/multi-index models (see comment below).

---

> ### Author Rebuttal · Authors · 2025-07-30
>
> We thank the reviewer for the thoughtful and constructive feedback. We’ll respond in two parts: first addressing your specific questions, then the three main weaknesses. While we appreciate your engagement, we were surprised by the low score. We hope our clarifications help and welcome further discussion to better understand your concerns.
>
> ## Questions
>
> *1. Confusing notation on L64-65.*
>
> In our notation, $a_i$ is a vector in $\mathbb{R}^c$, without any assumptions on $c$ in general. In the diagonal linear network and linear transformer settings, $c$ is equal to $1$, while in the fully-connected linear network and modular addition settings, we have $c > 1$. In line 64-65,  $a_i$ appears in a vector-times-scalar product. In line 72, the Gaussian is an isotropic one, with diagonal covariance. We agree that the expression in line 72 could be confusing, because $i$ indexes the neuron, and not the element. We will rewrite the expressions to be $a_i \sim \mathcal{N}(0, \alpha / \sqrt{2c} \mathbf{I}_c),  w_i \sim \mathcal{N}(0, \alpha / \sqrt{2d} \mathbf{I}_d)$.
>
> *2. What is a smooth dynamical system in L77-78?*
>
> In lines 77-78, we are simply paraphrasing the main result by Bakhtin. In their work, a “smooth” dynamical system is an ODE (or SDE) where the vector field (and diffusion covariance) is $C^2$. Under this definition, GF is smooth when the activation function $\sigma$ is of class $C^3$, which holds for all the settings we study.
>
> *3. Tuning down statement on L43-45*
>
> Our intention with this sentence was to convey that prior analyses each rely on *different* simplifying assumptions—be it about architecture, data, or optimizer—which makes it difficult to extract general principles. We are not trying to imply that our work does not also consider simplifying assumptions. We agree that this sentence could sound too dismissive; we will rephrase it in order to tune it down.
>
> *4. What is the role of the parametrization on L337-338?*
>
> We believe that the parametrization you described is identical to the one we use, but with a different notation: the  $\beta_i$ in your parameterization corresponds to the pair $(u_i, v_i)$ in ours. Under this change of notation, $\langle \beta_i, z \rangle = \langle u_i, a \cdot x \rangle +   \langle v_i, b \cdot x \rangle$. We use this notation for two reasons. First, it matches the notation used in prior works such as Gromov 2023 or Morwani et al. 2023, and second, the vector $u_i$ and $v_i$ will have different behaviors in the forthcoming analysis. While they will both be aligned to the same frequency, they can have different phase shifts. Therefore, it is notationally convenient to split the vector $\beta_i \in \mathbb{R}^{2p}$ in these two components.
>
> *5. & 6. What is the distinction with prior analysis, and why do we see different feature learning behavior?*
>
> We wish to highlight that the key distinction between our analysis and the prior works you cite lies in the initialization regime. In those prior analyses, the network is not initialized near the saddle point at the origin. In contrast, our analysis focuses on the *vanishing initialization* limit ($\alpha \rightarrow 0$), placing the network infinitesimally close to this critical point. This regime is central to AGF (and to related frameworks such as those in Jacot et al., Li et al., and Pesme & Flammarion); it is the primary mechanism inducing a separation of timescales, allowing the gradient flow dynamics to decouple. This motivates AGF, where such a separation is enforced by design. Instead, the line of research you mention focuses on the large architecture (or large input dimension) limit. Crucially, this might cause a population-distributed behavior, rather than a neuron-specialized one. As a result, the dynamics captured by AGF and the “specialization transition” studied in the prior works appear to arise from fundamentally different mechanisms—though both may reflect transitions between saddle points in the loss landscape. The difference between the picture that emerges from the quadratic analyses you cite and our results with AGF is rooted in this distinction as well.
>
> Moreover, there are other differences in assumptions, analysis techniques, and theoretical goals. We summarize them in the table below, and will discuss them in our updated related work section:
>
>
> ||**Prior Analyses (e.g., Saad, Goldt, Arnaboldi)**|**AGF Framework (our work)**|
> |---|---|---|
> |**Output dimension**|Scalar output only ($c=1$)|Handles both scalar ($c=1$) and vector-valued outputs ($c>1$)|
> |**Optimization**|Often train only the first layer with online SGD and keep the second layer fixed|Always train both first and second layers jointly under GF|
> |**Initialization regime**|Finite initialization variance ($\alpha \in O(1)$)|Vanishing initialization ($\alpha \to 0$)|
> |**Input dimension**|Thermodynamic limit ($d \to \infty$)|Finite input dimension ($d \in O(1)$)|
> |**Analytical approach**|Track a small number of macroscopic order parameters with deterministic ODEs that can be solved numerically|Track individual neuron dynamics via a timescale separation between direction and norm|
> |**Task and theoretical focus**|Teacher–student setups; generalization under online SGD; each work is specific to a particular architecture or activation|Feature acquisition under gradient flow; general framework that unifies saddle-to-saddle dynamics across multiple settings, can extend to new settings (modular arithmetic)|
>
> ## Weaknesses
> *1. The relevance of AGF beyond simple cases*
>
> We agree that there are valid reasons to question the broader applicability of AGF, particularly given its current formulation is limited to two-layer networks. However, we believe that this criticism applies to many analysis in theoretical deep learning. While proving a general equivalence between AGF and GF in the vanishing initialization limit is likely to be extremely challenging, we view our convergence result in the diagonal linear case as strong support that such an equivalence may hold under the right conditions. Moreover, the precise match between theory and experiment across the four very different settings we study, along with the recovery of earlier analyses as special cases, gives us reason to believe that AGF captures a fundamental phenomenon. It is possible that the four settings are the only ones where AGF applies, but given their diversity, we think it is far more plausible that AGF provides a broadly useful lens for analyzing feature learning in two-layer networks.
>
> Regarding the quadratic network example, we are, unfortunately, not sure which additional assumption the reviewer is referring to. The analysis in Section 5 closely follows the setup used in Gromov (2023) and Morwani et al. (2023), and in fact generalizes those works by allowing an arbitrary mean-centered input vector $x$. We would appreciate clarification so we can address this concern more precisely.
>
> *2. The “broad picture” offered by AGF*
>
> We believe the greatest value of AGF lies in mathematically formalizing an intuition that has been present in the saddle-to-saddle literature for a long time: that learning, at small initialization scales, unfolds in distinct phases. Our goal was to show that a wide variety of prior analyses—previously treated as separate—can be unified under a single framework, and that this framework can explain empirical observations that had lacked a precise theoretical account. We agree that precisely predicting the order, timing, and magnitude of loss drops in simple examples is not valuable on its own. But showing that such predictions can be made from a unified mechanism that captures these transitions analytically is, we believe, a meaningful contribution. As you note, the formulation of the utility function may also have broader implications, particularly for estimating the timescale of escaping from saddles. We see AGF as a starting point, and we can imagine future work applying it to more general two-layer settings, using it to analyze the dynamics of generalization, or extending its principles to deeper networks.
>
> *3. Missing references*
>
> You are absolutely correct, this was an oversight on our part.  We were primarily focused on works studying saddle-to-saddle dynamics and overlooked the extensive body of works studying feature learning in two-layer networks using different techniques such as mean-field, teacher–student, multi-index models, and single-step analyses. In our updated manuscript, we have included a new appendix section devoted to related works on feature learning in two-layer neural networks. We cite all the works you referenced, and a number of other works using similar techniques.
>
> Please let us know if you have any other questions. We hope you will consider raising your score and look forward to further discussion. Thank you again for your review.

---

> > ### Comment · Reviewer_qgue · 2025-08-03
> >
> > I thank the authors for their rebuttal to my review and for welcoming some suggestions. Most of my questions have been cleared, but two points remain.
> >
> > - *Point 4. on the parametrization of quadratic nets for modular arithmetics*:
> >
> > I understand the choice of parametrization matching previous work to illustrate AGF in this particular problem. However, my questions was not about notation but about the phenomenology. Due to the non-linearity, GF for a "standard" 2LNN with parameter $\beta_{i}$ and for a 2LNN parametrized by a sum $\beta_i=(u_i,v_i)$ has a different behavior.
> >
> > My question was: does AGF hold for the standard 2LNN, without this neuron-splitting?
> >
> > - *Points 5. and 6. on the relation to previous work.*.
> >
> > Thank you for clarifying and highlighting the difference between the saddle-to-saddle phenomenology on the aforementoned related literature and your work. I agree that although both are driven by saddle-points in the dynamics, they are of a different nature, the former not being captured by the AGF picture offered by this work.
> >
> > Given the fundamental differences between two co-existing mechanisms governing feature learning on two-layer neural networks, what bothers me, besides the lack of reference to this complementary literature, is that this work fails to acknowledge these differences, often referring to AGF as "*the*" rather than "*a*" mechanism for feature learning in 2LNN. From the overly ambitious title to some sentences in the work, e.g. L46-49, L60-61, L378-379, etc.
> >
> > Note I am not questioning the correctness or usefulness of AGF as a theoretical framework to understand feature learning in 2LNN, but just drawing your attention to its limitations. I would be happy to reconsider my score if the authors can propose changes that acknowledge this.
> >
> > **Minor remark**: Not all works employing the ODE description in your table require the "thermodynamic limit" $d\to\infty$. (Arnaboldi et al., 2023a) shows that in some regimes, the analysis extends to $d=O(1)$, see e.g. Fig. 2 (right) in (Arnaboldi et al., 2023a).

---

> > > ### Author Response · Authors · 2025-08-04
> > >
> > > Thank you for the thoughtful follow-up. We are happy to hear that our reply has clarified most of your concerns. Below, we further discuss your two points that remain:
> > >
> > >
> > > *4. On the parametrization of quadratic nets.*
> > >
> > >
> > > We apologize, but we do not understand your concern. As mentioned in our rebuttal, a standard 2LNN with parameters $\beta_i \in \mathbb{R}^{2p}, w_i \in \mathbb{R}^p$ for each neuron, and the 2LNN with parameters $u_i  \in \mathbb{R}^p, v_i  \in \mathbb{R}^p, w_i  \in \mathbb{R}^p$ for each neuron, are identical networks with identical dynamics. They only differ in a change in notation. If we denote by $u_i := \beta_i[:p]$ the vector consisting of the first $p$ entries of $\beta_i$ (in Python indexing convention), and by $v_i := \beta_i[p:]$ the one consisting of the last $p$ entries, then for all inputs $x \in \mathbb{R}^{2p}$,
> > >
> > > $$\sum_{i = 1}^H w_i(\beta_i^\intercal x)^2 = \sum_{i = 1}^H w_i(u_i^\intercal x[:p] + v_i^\intercal x[p:])^2.$$
> > > The LHS is how we understand what you call a "standard 2LNN” with quadratic non-linearity, and the RHS is what appears in our paper (line 338). They are parametrized identically, albeit with a different notation. Please let us know if we are misunderstanding your perspective.
> > >
> > >
> > > *5. & 6. On the relation to previous work*
> > >
> > >
> > > We are happy to hear that we are in agreement about the differences in learning regime between AGF and the papers you mention. We agree with you that we should have cited that literature in our original submission. That was an oversight, and as we have explained in our rebuttal, we have added citations to this literature throughout our work. Moreover, we have added a whole related work section titled “Analysis of feature learning in two-layer networks”, where we discuss in detail other theories of feature learning in two-layer networks, and how they relate to our work. We have also extended our limitations section (Section 6) – please see our conversation with Reviewer vhww, where we discussed limitations of studying feature learning in the vanishing initialization regime. In this section, we will also discuss regimes in two-layer networks that exhibit saddle-to-saddle dynamics (identified in prior works) that are **not** captured by our analysis.
> > >
> > > We acknowledge that some of our sentences might sound dismissive. In addition to the changes we proposed in our rebuttal, we have changed the language in the specific lines you referenced accordingly:
> > > - L46-49: Now reads, “In this work, we introduce a theoretical framework that unifies existing analyses of saddle-to-saddle dynamics in two-layer networks **under the vanishing initialization limit**...” to make it clear the regime we are studying.
> > >
> > > - L60-61: We have added after this sentence, “See Appendix A for further **discussion of related theoretical approaches** to studying feature learning and saddle-to-saddle dynamics in two-layer networks, including mean-field analysis and teacher–student frameworks.” This directly links to our extended related work section.
> > >
> > > - L378-379: Now reads, “Our results position AGF as a unifying framework for understanding feature learning dynamics in two-layer neural networks **under vanishing initialization**.”
> > >
> > > We have searched for other sentences in the document where either our tone was too dismissive of prior work or over stating our own work, and rephrased them accordingly.
> > >
> > >
> > > We are happy to hear you are willing to reconsider your evaluation of our paper. We hope that our response has clarified your concerns, and that the changes we have proposed satisfy your reservations with our work.

---

> > > > ### Comment · Reviewer_qgue · 2025-08-05
> > > >
> > > > I thank the authors for their answer. The question on the parametrization of the quadratic 2LNN is clear now - it arised from a confusion on my side. As for the proposed changes, they address my concern. I am happy to update my score accordingly.

---

> > > > > ### Author Response · Authors · 2025-08-05
> > > > >
> > > > > Thank you for your careful review, and for the time and effort you put into engaging with us during this discussion period. Your feedback helped us strengthen the paper significantly, and we truly appreciate it.

---

### Official Review · Reviewer_D2un · 2025-07-01

**Clarity:** 3
**Significance:** 3
**Originality:** 3
**Rating:** 4
**Confidence:** 4

**Summary:**

In this paper, the authors study the training dynamics of shallow neural networks, with a focus on saddle-to-saddle dynamics that emerge under small initialization. They propose a framework called Alternating Gradient Flows (AGF) to describe the behavior in this regime. AGF consists of two alternating steps: maximizing a utility function over dormant (i.e., small) neurons, and minimizing a cost function over active (i.e., large) neurons. This framework unifies several existing saddle-to-saddle dynamics, including those found in diagonal linear networks, linear networks, and attention-only linear transformers. It also provides a new perspective on the dynamics involved in learning modular addition. Experimental results are included to support the proposed claims.

**Questions:**

1.	Is it possible for an active neuron to become dormant again? In all the plots, it seems that once a neuron becomes active, it remains active. Could the authors comment on whether reversion to dormancy is theoretically possible or observed in practice?
2.	Does the AGF framework accurately capture the dynamics of general (non-orthogonal) two-layer neural networks, such as the one shown on the left side of Figure 1? If not, are there specific conditions under which the agreement breaks down?
3.	Do the authors have an estimate for how small the initialization scale $\alpha$ needs to be in order for the actual dynamics to closely follow the AGF dynamics? Specifically, is there any known rate of convergence with respect to $\alpha$?

**Ethical Concerns:**

["NO or VERY MINOR ethics concerns only"]

**Final Justification:**

The proposed AGF framework unifies several prior analyses of saddle-to-saddle dynamics and offers intuitive explanations for the behavior observed in these settings. The response from authors addressed my concerns. I believe this is a valuable contribution and therefore recommend acceptance.

**Limitations:**

Yes

**Quality:**

3

**Strengths And Weaknesses:**

Strengths:

1.	Understanding the training dynamics of neural networks, particularly in the feature learning regime and in settings like the saddle-to-saddle dynamics studied in this paper, is an important and active area of research.
2.	The proposed Alternating Gradient Flows (AGF) framework unifies several prior analyses of saddle-to-saddle dynamics and offers intuitive explanations for the behavior observed in these settings.
3.	The framework also sheds light on the training dynamics involved in learning modular addition, which appears to be a novel contribution.

Weaknesses:

1.	While the AGF framework offers useful insights into saddle-to-saddle dynamics, it does not provide a full end-to-end analysis of the training process. Many of the examples rely on heuristics or assume that each step in the alternation (e.g., utility maximization and cost minimization) is perfectly executed. While these assumptions are helpful for building intuition, they limit the framework's predictive or theoretical completeness.

    Additionally, the utility maximization step may not be tractable in general. In some scenarios not covered by the paper, solving this maximization could be NP-hard.

---

> ### Author Rebuttal · Authors · 2025-07-30
>
> We thank the reviewer for thoroughly reviewing our paper, and for highlighting areas needing further clarification. We appreciate the positive feedback on our work. We will address each of the weaknesses and questions mentioned individually.
>
> **Full end-to-end analysis.** We agree that the AGF framework, like most theoretical frameworks, requires simplifying assumptions to enable tractable analysis. However, we would like to clarify that in the diagonal linear network setting, AGF does in fact yield a full end-to-end analysis of training dynamics, and we prove convergence of AGF to gradient flow (see Theorem 3.1). In the fully-connected linear network case, we provide a complete analysis under the assumption that the input covariance commutes with the input-output cross-covariance; in the more general non-commuting case, we state a conjecture supported by empirical evidence (see Conjecture 4.1). As correctly pointed out in the review, in the linear transformer and modular addition settings, we assume perfect execution of the utility maximization step, in order  to keep the analysis tractable. This is precisely why our the predicted jump times in Figures 5 and 6 are presented as lower bounds. Overall, while we acknowledge that the full generality of AGF remains analytically intractable in some settings, our goal is to isolate and analyze regimes where rigorous statements can be made, and the resulting predictions remain consistent with observed behavior.
>
> **Active to Dormant.** Yes, it is possible in the cost minimization step for active neurons to become dormant again, if the optimization trajectory brings an active neuron near the origin. This is mentioned on lines 142-143, and discussed more thoroughly in the Appendix, Section A.3 (unfortunately, the link on line 144 incorrectly points to Section B). This describes how reversion to dormancy is theoretically possible. While we rarely observe this empirically, it happens for diagonal linear networks – see figure 7 in the Appendix. Here, the first coordinate $\beta^{(1)}$ activates, followed by $\beta^{(2)}$. During the second step of cost minimization, $\beta^{(1)}$ deactivates as it most change sign and thus the optimization takes the neuron near the origin. At the next step $\beta^{(1)}$ activates with the new sign to reach the global minimum. We will highlight this example in the main text.
>
> **Numerical implementation of AGF.** To explore the applicability of AGF beyond analytically-tractable settings, we have developed a numerical implementation of AGF for arbitrary two-layer neural networks trained in PyTorch (shared via GitHub, though we are unable to link it here due to NeurIPS rebuttal policies). Using this tool, we have compared loss curves between gradient descent (at small initializations) and AGF in general non-orthogonal settings, where theoretical analysis is currently infeasible (e.g., a two-layer ReLU network trained on CIFAR-10). Our findings suggest that AGF can closely match the behavior of gradient descent in these settings. However, this requires careful tuning of hyperparameters, particularly those controlling the termination of the cost minimization step (e.g., number of iterations or gradient norm threshold). We will include an additional plot in Section 6 showing representative examples from these experiments, along with a discussion of when and why numerical AGF may diverge from standard training dynamics. We agree that identifying the precise conditions under which AGF faithfully approximates full gradient descent is a crucial direction for future work, and are actively exploring this question.
>
> **Convergence rate.** We believe that this is an interesting question, and something we have not spent time thinking about. Likely, the best place to start answering this question is the setting of diagonal linear networks, where we can prove that AGF and GF do converge to each other. Extracting convergence rates would require a careful analysis of the proof by Pesme and Flammarion, who first formally connected gradient flow with a step-wise process. We believe that this might be possible, but it would be subtle and technical. Yet, it is an important point to bring up for future work. We will add to the appendix a short discussion on this.
>
> Please let us know if you have any other questions regarding our work. We hope you will consider raising your score and continue to consider our paper an important contribution to the NeurIPS community.

---

> > ### Comment · Reviewer_D2un · 2025-08-04
> >
> > Thank you for the response to address my concerns. I will keep my score and recommend acceptance.

---

### Official Review · Reviewer_hRND · 2025-07-03

**Clarity:** 1
**Significance:** 3
**Originality:** 4
**Rating:** 5
**Confidence:** 5

**Summary:**

This paper introduces Alternating Gradient Flows (AGF), an algorithm designed to approximate the saddle-to-saddle dynamics observed when training two-layer neural networks with vanishing initialization. AGF alternates between a utility maximization phase, during which dormant neurons evolve independently (corresponding to plateaus in the loss), and a cost minimization phase, where active neurons collectively reduce the loss. The authors prove that AGF converges to the saddle-to-saddle dynamics of gradient flow in the limit of vanishing initialization for diagonal linear networks, and they conjecture that it captures the dynamics more broadly across other architectures.

**Questions:**

- In the definition of AGF, it is implicitly assumed that a single neuron $i_*$ activates at each iteration. However, in practice—particularly in fully connected networks or ReLU networks—multiple neurons often activate simultaneously in clusters. This phenomenon is not captured by the current formulation.
- Line 148 : "Through this process, we have generated a precise sequence of saddle points and jump times to describe how gradient flow leaves the origin through a saddle-to-saddle process."Can you clearly justify why the intermediate points in this sequence correspond to actual saddle points of the loss function.
- Intuitively, during the first phase of AGF (utility maximization by dormant neurons), one would expect the normalized weights $\bar{\theta}_i$ to quickly align with the direction that maximizes the utility function and then remain stationary. However, to my knowledge, this is not what happens in practice—for example, in ReLU networks (and even in certain linear settings), the alignment direction can change over time, even without any neuron becoming active. Could you comment on this?

**Ethical Concerns:**

["NO or VERY MINOR ethics concerns only"]

**Final Justification:**

My questions have been clearly addressed by the authors. I stand by my original review and recommend acceptance of the paper, with the hope that the fuzzy definitions will be clarified in the revised version.

**Quality:**

3

**Strengths And Weaknesses:**

**Strengths**
The paper addresses an important problem: understanding how features emerge during training. The proposed framework, Alternating Gradient Flows (AGF), offers an original perspective on this question. While the derivation involves many heuristic arguments, the approach is conceptually clear and quite compelling. A notable strength is that the framework applies broadly to two-layer architectures, rather than being tailored to a specific model. The introduction of new quantities—such as the utility function guiding dormant neuron dynamics—is (to the best of my knowledge) novel and insightful. Finally, the paper is supported by clear and informative illustrations (e.g., Figures 1 and 2), which help convey the underlying intuition.

**Weaknesses**
The paper relies on heuristics and informal arguments, and some of the mathematics lacks rigour and clarity. For example:
- the key equation in (2) is introduced rather abruptly. Since they underpin the AGF framework, it would be important to explain more carefully in what sense they validly approximate the gradient flow dynamics.
- The use of Euler’s homogeneous function theorem (e.g., line 643 in the appendix) appears questionable. While the utility function  $U_i$ may be locally $\kappa$-homogeneous near the origin, the theorem is applied in regions where $\Vert \theta_i \Vert$ is not necessarily small, which undermines its validity. Several steps in the derivations in the appendix are skipped; more detailed computations would be helpful.
- Another issue lies in the definition of AGF: the stopping conditions for the two while loops are ill-defined. Neither $\nabla L(\Theta_D)$ nor $\nabla L(\Theta_A)$ will reach exactly zero in finite time, so the algorithm as written is not well-posed. It is unclear how this should be interpreted. Moreover, the second loop (cost minimization) is said to run until convergence, or let's say "nearly convergence", which in practice would take some nonzero time. Yet, in the illustrations (e.g., Figure 2, right), the step is depicted as instantaneous. How come?
- In the definition of AGF, if a neuron collapses during the cost minimization phase and becomes dormant again, it is unclear how its associated accumulated utility $S_i$ should be initialized in the next utility maximization phase. This detail is not specified in the current version of the algorithm, yet it is essential for the consistency of the method.
- The current definition of AGF does not appear to align with the algorithm in Pesme and Flammarion. Consider their example in Figure 2 of their paper: initially, the active set is empty. At time $t_1$, coordinate 1 is added to the active set, and phase 2 proceeds until the gradient vanishes over this active set. We then return to phase 1, and at time $t_2$, the second coordinate is added to the active set. We then enter phase 2 again, and according to AGF,  this phase is meant to continue until the gradient is zero over both active coordinates. However, this is not what occurs in the dynamics: the iterates pass through an intermediate saddle point $\beta_{c,2}$ due to the collapse of the first coordinate. Therefore, in its current form, the definition of AGF does not capture this behaviour, and as such, Theorem 3.1—which claims equivalence in the $\alpha \to 0$ limit—cannot be correct without modification of the AGF algorithm.


**Minor comments**
-In the definition of AGF, the quantity $\eta_\alpha$ is used without prior definition.
- In the def of AGF: the expression $\frac{d \theta_j}{d t} = -\nabla L(\Theta_A)$ is problematic: the left-hand side is a vector corresponding to neuron $j$, while the right-hand side is the full gradient over all active neurons.
- In my opinion, the use of random initializations $a_i \sim \mathcal{N}(0, \frac{\alpha}{2c})$, $w_i \sim \mathcal{N}(0, \frac{\alpha}{\sqrt{2d}})$ makes things unnecessarily complicated. Why not simply consider deterministic initializations of the form $a_i^{\alpha} = \alpha a_i$ for fixed vectors $a_i$? This setup strictly generalizes the random case.

---

> ### Author Rebuttal · Authors · 2025-07-30
>
> We are grateful for the thorough review, the constructive suggestions, and the positive feedback. We will address the weaknesses and questions individually.
>
> **Validity of the AGF approximation.** We acknowledge that Equation 2 is introduced abruptly, and that the derivations in Appendix A.1 are rather compressed. In the updated version of the manuscript, we will expand the derivations in Section A.1, and summarize them in the main body of the paper (Section 2). We wish to point out that Equation 2, together with its related derivations, is meant to approximate the GF dynamics *around the origin*. Indeed, that equation is used only in the utility maximization step of AGF. This step only involves dormant neurons, which, by definition, lie close to the origin. This justifies treating the utility (of dormant neurons) as a $\kappa$-homogeneous function, and applying Euler’s theorem. In the updated version, we will stress the fact that Equation 2, and the derivations in Section A.1, assume that $\theta_i \sim \mathbf{0}$. Of course, during the utility maximization phase of AGF, the dormant neurons could, technically, leave the region where the approximation is valid. Indeed, AGF is at its core an *ansatz*; its derivation is not completely formal, and its equivalence with GF at the vanishing initialization limit ($\alpha \rightarrow 0$) is conjectural in the general case (see lines 150-152).
>
> **Definition of AGF.**  We agree that the pseudocode in Algorithm 1 is not completely precise. Our intention was to balance between mathematical rigor and intuition. Below, we address your main concerns raised regarding the formalism of Algorithm 1.
>
> - Our intention with the expression $\frac{d \theta_j}{d t} = - \nabla L (\Theta_{\mathcal{A}})$ was to imply that the gradient is taken w.r.t. $\theta_j$. We will change the notation for the RHS to $\nabla_{\theta_j} L (\Theta_{\mathcal{A}})$, hoping to resolve the confusion.
>
> - It is true that the equilibria during the cost minimization phase can take infinite time to be reached. In fact, the cost minimization step should be interpreted as taking infinite time, akin to how Greedy Low-Rank Learning (GLRL) is expressed in pseudocode in Li et al. 2020 (Algorithm 1 in their work). Like in their work, in practice, we approximate the infinite time limit by running sufficiently many steps. In fact, we find that the amount of time we run cost minimization becomes a crucial hyperparameter when running AGF numerically – see our reply to Reviewer D2un. Additionally, note that in AGF, time can only be accumulated in the utility maximization phase. That is why in Figure 2 (right) – which is meant to represent GF at the limit $\alpha \rightarrow 0$, where the loss becomes piecewise constant – the cost minimization phase is depicted as instantaneous.
>
> - The halting condition $\nabla\mathcal{L}(\Theta_\mathcal{D}) \neq 0$ is a way to express two conditions: (1) the dormant set is empty and (2) if all remaining dormant neurons have zero utility, i.e. their gradient is zero, then the next step of utility maximization will take infinite time and thus the algorithm should be terminated. In this paper we only consider settings where all neurons will eventually activate, however, if for example using ReLU, one could imagine a setting where all remaining dormant neurons are dead and thus no more neurons can activate.
>
> - If the $i$-th neuron becomes dormant again, its accumulated utility $\mathcal{S}_i$ should be initialized at the maximal allowed value, i.e., $c_i$. Such utility will then start decreasing at the next utility maximization phase. In order for this to be consistent with the current halting condition for the utility maximization step, the inequality $\mathcal{S}_i < c_i$ in Algorithm 1 should be corrected into $\mathcal{S}_i \leq c_i$ (otherwise, that neuron will immediately jump and become active again). We thank the reviewer for bringing this up, and will elaborate on this in the updated version, including the correction to the inequality.
>
> **Apparent inconsistency with Pesme and Flammarion.** We believe that AGF is actually coherent with Pesme and Flammarion’s algorithm – please see Figure 7 in our appendix. This figure actually recreates Figure 2 in Pesme and Flammarion, and visualizes empirical gradient flow dynamics for the parameters, accumulated utility, parameter directions and the predictions for these quantities under AGF. The subtlety is that while the cost minimization phase in AGF continues until convergence, the active neurons are removed immediately when they collapse, i.e., when they approach their initialization again (see Appendix A.3 for a formal discussion on the condition to become dormant again). This is the reason the sentence “Remove collapsed neurons” appears *inside* the loop of the cost minimization phase in Algorithm 1. We acknowledge that the instantaneous removal of collapsed active neurons is not clear in Algorithm 1; we will elaborate on this in the updated version of the paper. For the example discussed (using the notation in Figure 7 of our work), when the first coordinate $\beta_1$ collapses at time $\tau^{(2)}$, in AGF it would be removed immediately from the active set, and the cost minimization phase would continue for the other active neuron $\beta_2$ until it reaches a value corresponding to the second saddle. Then utility maximization would proceed with neuron $\beta_1$, which is dormant again, starting with an accumulated utility $\mathcal{S}_i = c_i$. It would first lose utility until $\mathcal{S}_i = 0$ at $(\tau^{(2)}+ \tau^{(3)}) / 2$, switching signs, then gaining utility until it activates at $\tau^{(3)}$ with the correct sign. Then during cost minimization $\beta_1$ and $\beta_2$ now both active with the correct sign would converge to the values associated with the minimum.
>
> **Questions**
> - We agree that it is possible for multiple dormant neurons to activate simultaneously. This is an edge case of AGF, where the accumulated utilities of multiple neurons reach their corresponding maximal value $c_i$ at the same exact time. AGF can still be applied in this case; the only nuance is that $i_*$ will not be a single index, but rather a subset of $\mathcal{D}$, which will be subtracted from $\mathcal{D}$ at the end of utility maximization. We will correct this inaccuracy in line 117 by mentioning that, as an edge case, $i_*$ could contain multiple indices, and is not necessarily a singleton.
>
> - If dormant neurons are treated as lying at the origin (which is exact at the limit $\alpha \rightarrow 0$, and only approximate away from it), the intermediate points in AGF are critical points (for the loss) of the type described in lines 85-94. Moreover, the trajectory of the dynamics passes in their proximity, approaching and then departing (excluding the last point, where the dynamics converges). At the limit $\alpha \rightarrow 0$, this implies that such points possess a stable (infinitesimal) direction, and an unstable one, i.e., they are saddles of the loss.
>
> - We agree that one could intuitively expect the dormant neurons to align instantaneously (in the limit $\alpha \rightarrow 0$), and to stay aligned throughout utility maximization. However, as stated in the review, this is not what happens, even for scenarios considered in our work. We discuss this question in Appendix A.4, where we argue that instantaneous alignment is expected for $\kappa = 2$, but not otherwise. This is why the jump times for linear transformers and quadratic networks, for which $\kappa = 3$, are stated as (approximate) lower bounds (Equations 7 and 9). When the alignment is not instantaneous, the direction $\bar{\theta}_i$ of a dormant neuron can change during a single utility maximization phase (i.e., without any neuron becoming active). Additionally, even for $\kappa = 2$ one could see a change in direction for dormant neurons, but this change would be instantaneous and associated with the accumulated utility reaching zero. In fact that is exactly what we see in Figure 7 for the neuron associated with $\beta_1$.
>
> We hope that the above discussions will clarify your concerns, and that this will increase your confidence in the importance of our work. Please let us know if you have any other questions. Thank you again for your very detailed and constructive review!

---

> ### Comment · Reviewer_hRND · 2025-08-06
>
> Thank you for your detailed responses. My questions and concerns have been addressed.
>
> I will keep my score as it is. However, I would strongly encourage that the revised version takes these concerns into account. I am fine with the idea that AGF is an ansatz—but it should be crystal clear to the reader which parts of the construction are heuristic or approximate, and which are formal.
>
> Moreover, in my view, there are too many opaque or ill-defined components in the current formulation of the AGF algorithm. I don’t mind having an intuitive or informal version in the main text, but I believe there should also be a fully precise and rigorous definition somewhere in the paper, one that eliminates ambiguities such as those I raised. This is particularly important given that the paper states formal theorems, for which one expects a corresponding level of rigour in both definitions and proofs.

---

> > ### Author Response · Authors · 2025-08-07
> >
> > Thank you again for your thoughtful review and your support of our work. We really appreciate both of your points. We'll revise the introduction, Section 2, and discussion to make it crystal clear that AGF is presented as an **ansatz**, and that parts of the motivation are approximate. We'll also include a more rigorous definition of AGF in the appendix and add a reference to it from the pseudocode in Figure 2. Thanks again for your helpful feedback!

---

### Official Review · Reviewer_wsYE · 2025-07-06

**Clarity:** 3
**Significance:** 3
**Originality:** 3
**Rating:** 4
**Confidence:** 4

**Summary:**

This paper introduces Alternating Gradient Flows (AGF), a framework that models how features emerge in two-layer networks trained from small initialization. AGF captures the staircase-shaped loss dynamics as an alternating process: dormant neurons activate one by one, each time acquiring a new feature and causing a sharp drop in loss. It predicts the timing and magnitude of these drops, aligning with experiments.

The authors claim that AGF unifies prior analyses across architectures (e.g., linear networks, transformers), and in some cases provably matches gradient flow. Applied to modular addition tasks, it shows that networks learn Fourier features in decreasing order. Overall, AGF offers a structured view of feature learning dynamics.

**Questions:**

No

**Ethical Concerns:**

["NO or VERY MINOR ethics concerns only"]

**Final Justification:**

I thank the authors for the answer that they give. It confirms my overall impression on the paper. I am not very familiar with the modular addition framework, its intrinsic difficulty, etc, but the authors made a point that their algorithm can illustrate well (and not explain!) some mechanisms that appear during the learning of neural networks.

I am willing to keep my score.

**Quality:**

3

**Strengths And Weaknesses:**

Strengths :

- The idea of this type of algorithmic/sequential dynamics that mimics the gradient flow has been in the air since 4-5 years and this attempt to unify these works and put some clear mathematical content is a good thing
- The identification of the independent dynamics of *dormant neurons* and the correlated dynamics of *active ones*
- The theorem linking AGF and diagonal linear networks
- The modular addition example

Weaknesses :
- The other side of the coin is that this article only seems to put words in a phenomenon that people understood (at least since Maennel, 2018). The mathematical content, beyond the equivalence with diagonal linear networks, is not very rich, and the work is mostly a phenomenology.
- I don't believe that a general conjecture between AGF and gradient flow is plausible : in the case of orthogonal inputs, for two layer relu networks, there might be several neurons that contribute to the loss, while it appears to be that with the AGF only two will be active (one for positive labels one for negative ones). Yet the loss dynamics shape should match (but not parameters dynamics).
- Sometimes the authors play a bit with the notion of what is *dormant*: one time it corresponds to small in magnitude, another time it comes from the fact that it is orthogonal to the signal etc... I feel that the precise definition of it has to be polished.

Minor typos :
- line 87 : exponentially many saddle points --> I believe in some settings, the set of saddle points to be connected
- critical points

---

> ### Author Rebuttal · Authors · 2025-07-30
>
> We appreciate the thoughtful review, and we are happy to hear you think our work merits acceptance. We will respond individually to the weaknesses and questions raised regarding our paper.
>
> **The mathematical content.** We agree that many of the ideas behind AGF (i.e., two phases of optimization separated by timescales) existed in prior literature, albeit not described in a unified framework. However, we disagree that the mathematical content in this work, beyond the equivalence with DLNs, is not rich. To this end, we wish to point out that nearly all the analysis in Section 5 (“AGF Predicts the Emergence of Fourier Features in Modular Addition”) is extremely novel and non-trivial. In particular, we highlight the following two theorems in this section:
>
> - Theorem 5.1 proves that the utility maximizing directions for a quadratic neuron in the modular addition task are cosine waves aligned with the dominant frequency of the encoding vector. Proving this involves recasting a nonlinear constrained optimization problem into a solvable optimization problem in the Fourier domain. Here we make use of the shift-equivariance property of the Fourier transform and Plancharel’s theorem.
>
> - Theorem 5.2 proves that there exists a lower bound on the loss within the subspace spanned by neurons aligned to the same frequency, and provides precise conditions on the relationship between these neurons in order to achieve this lower bound. The proof again involves tools from harmonic analysis and several subtle trigonometric identities.
>
> We do not consider our work as “phenomenological”, but rather as a precise mathematical description of intuition that unifies existing analysis, and that can be used to derive novel analysis leading to predictions matching empirical observations.
>
> **General conjecture.** Thank you for your thoughts here. We agree that proving the general conjecture between AGF and GF is likely to be extremely challenging. Yet, given that convergence of AGF towards GF is established in the setting of diagonal linear networks, we believe that there is substantial evidence to support a general conjecture, under the right constraints. For example, a constraint requiring that all neurons are needed to achieve zero training error, something that is true in the cases we studied and would prevent the situation you suggested in your example, might be needed. In order to clarify this, we will add a further discussion in Section 6 around when a general conjecture might be plausible, together with steps to be considered for future work.
>
> **Definition of dormant.** In the context of AGF, a neuron is defined to be dormant if its accumulated utility is less than its initialization-dependent threshold – see Section 2.2 or Figure 2. We are always consistent with this definition throughout the work. When motivating AGF, we introduce the concept of dormant and active neurons to describe the structure of saddle points in the loss landscape. Here, dormant and active are consistent with the definitions used in AGF. However, when describing GF near saddle points, we use dormant and active as adjectives to describe neurons that are either small in norm or not. In a sense, since GF is continuous, at finite scale a neuron is never dormant, but rather active to a degree. We understand that this language could cause confusion, and we will work to improve the writing here. Is there another section in the paper where you think we could be more formal with the definition of dormant?
>
> **Minor typos.** We agree that, in some settings, the saddle points described in lines 85-94 can be connected, in the sense that some of them can coincide. Note, on line 87 we state “one of *potentially* an exponential number of critical points”. For example, in diagonal linear networks, with generic data, the number of critical points is exponential. However, it is possible to construct data where the number of saddle points is less. We use the term "critical points" instead of "saddle points" because these points are not necessarily saddles; they could be (local) minima.
>
> Thank you again for your constructive feedback. We hope we addressed some of your concerns regarding our work. If we have, we would appreciate it if you could raise your score to reflect this.

---

> > ### Comment · Reviewer_wsYE · 2025-08-05
> > **Answer of the authors**
> >
> > I thank the authors for the answer that they give. It confirms my overall impression on the paper. I am not very familiar with the modular addition framework, its intrinsic difficulty, etc, but the authors made a point that their algorithm can illustrate well (and not explain!) some mechanisms that appear during the learning of neural networks.
> >
> > I am willing to keep my score.

---

### Decision · Program_Chairs · 2025-09-17

**Decision:**

Accept (poster)

**Comment:**

This work uses an Alternating Gradient Flow (AGF) algorithm for approximating the gradient flow dynamics of two-layer layer neural networks.
The motivation is from the fact that convergence of AGF towards GF is established for diagonal linear networks, but I'm still questionable about such a general conjecture (also mentioned by Reviewer wsYE). More discussion are needed, also committed by the authors.

The spirit of AGF follows the saddle-to-saddle dynamics: i) at a saddle-point, a subset of dormant neurons (with small weights) evolve independently to maximize their correlation with the rest, active neurons; ii) the alignment for a neuron leads to a drop of loss until they reach a new plateau. The results show that i) AGF is exact when the initialization scale for diagonal neural networks. ii) linear two-layer networks make AGF recovers the certain subspace. iii) AGF implements stepwise principal component regression for linear transformers. iv) two-layer quadratic networks trained on AGF learn Fourier features a modular addition task.

The results are comprehensive and all of the reviewers vote for acceptance after the rebuttal. Hence I recommend to accept this paper and suggest the authors to incoporate the reviewers' feedback in the final version.